# HIVEX: A High-Impact Environment Suite for Multi-Agent Research

## Abstract

Games have been vital test beds for the rapid development of Agent-based research. Remarkable progress has been achieved in the past, but it is unclear if the findings equip for real-world problems. While pressure grows, some of the most critical ecological challenges can find mitigation and prevention solutions through technology and its applications. Most real-world domains include multi-agent scenarios and require machine-machine and human-machine collaboration. Open-source environments have not advanced and are often toy scenarios, too abstract or not suitable for multi-agent research. By mimicking real-world problems and increasing the complexity of environments, we hope to advance state-of-the-art multi-agent research and inspire researchers to work on immediate real-world problems.

Here, we present HIVEX, an environment suite to benchmark multi-agent research focusing on ecological challenges. HIVEX includes the following environments: Wind Farm Control, Wildfire Resource Management, Drone-Based Reforestation, Ocean Plastic Collection, and Aerial Wildfire Suppression. We provide environments, training examples, and baselines for the main and sub-tasks. [1]

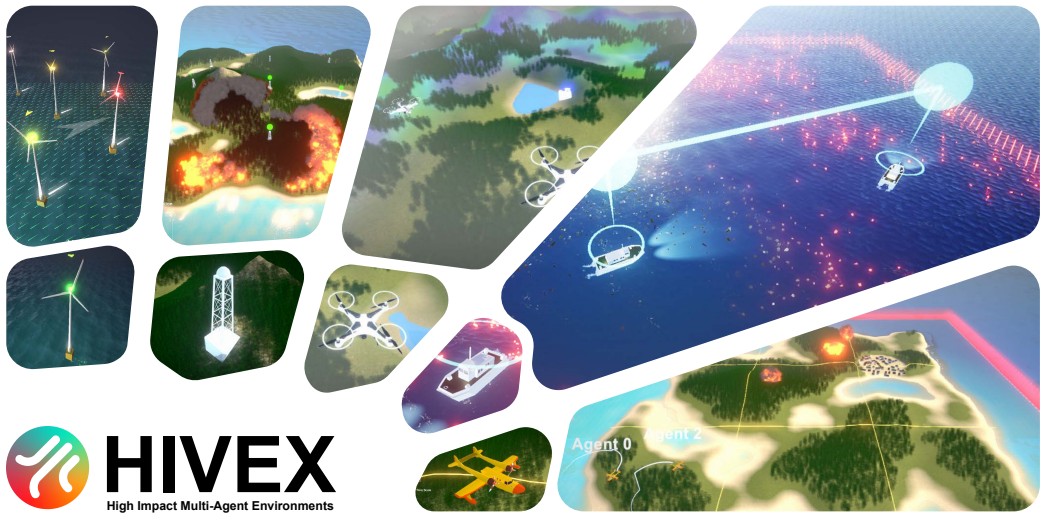

## 1 Introduction

Currently, no open-source benchmark for multi-agent reinforcement learning (MARL) closely mimics real-world scenarios focused on critical ecological challenges, offering sub-tasks, fine-grained terrain elevation or various layout patterns, supporting open-ended learning through procedurally generated environments and providing visual richness. Most common benchmarks with direct real-world applications are in the following domains: 1. intelligent machines and devices, 2. chemical

---

[1]GitHub Organisation: ANONYMIZED

engineering, biotechnology, and medical treatment, 3. human and society, and 4. social dilemmas Ning & Xie (2024).

The main HIVEX environment features are either procedurally generated or sampled from a random distribution. Therefore, training and evaluation are differentiated by seed values, ensuring testing scenarios are not seen during training. We aim to assess and compare MARL algorithms, focusing on test-time evaluation with zero-shot test scenarios. If applicable, a scenario consists of an environment and a task-pattern or terrain elevation combination. Each environment has a main end-to-end task and isolated subtasks that are independent or part of the main task. Environments have between two and nine tasks, various layout patterns, or terrain elevation levels. The environments described are ordered by increasing complexity in observation size and type, action count and type, and reward granularity, including individual and collective rewards. We introduce combinations of vector and visual observations and discrete and continuous actions.

Climate change is manifesting more visibly and urgently than ever Archer & Rahmstorf (2010); Romm (2022). We are witnessing an increase in frequent and intense weather phenomena, such as storms, droughts, fires, and floods UCLouvain (2023). Figure 8 shows the aforementioned disaster types triple in frequency between 1980 and 2020. These events reshape ecosystems and critically impact agriculture and natural resources vital to human survival Change (2012). A concerning report by the Intergovernmental Panel on Climate Change (IPCC) in 2022 highlights the dire consequences of continued greenhouse gas emissions, warning that significant curbing measures are needed within the next three decades to avert catastrophic impacts. Suppose the 1.5 ℃ degree increase in global warming cannot be negated. In that case, some impacts may be long-lasting or irreversible, such as the loss of ecosystems potentially fundamental to our existence Ipcc (2022). For further background and motivation behind this work, please refer to the Motivation: Critical Ecological Challenges section in the Appendix A.2.

## 2 THE HIVEX ENVIRONMENT SUITE

HIVEX addresses ecological challenges, developed in Unity using the ML-Agents Toolkit Juliani et al. (2020). Each environment mimics a real-world scenario where multiple agents interact, collaborate, and compete, providing rich settings for multi-agent research. Scenarios include:

1. **Wind Farm Control**: Agents adjust turbine orientations based on wind conditions.
2. **Wildfire Resource Management**: Agents allocate firefighting resources during wildfires.
3. **Drone-Based Reforestation**: Drones collaborate to plant trees in deforested areas.
4. **Ocean Plastic Collection**: Cleanup vessels locate and retrieve plastic waste from oceans.
5. **Aerial Wildfire Suppression**: Firefighting planes work together to extinguish wildfires and protect the village.

Agents receive vector and visual observations from their environment and perform multi-faceted actions such as adjusting turbines, shifting resources, planting seeds, and collecting ocean plastic. Real-world constraints are imposed, such as drone battery life limitations, requiring strategic recharging to maximize efficiency.

### 2.1 WIND FARM CONTROL

### 2.1.1 ENVIRONMENT SPECIFICATION

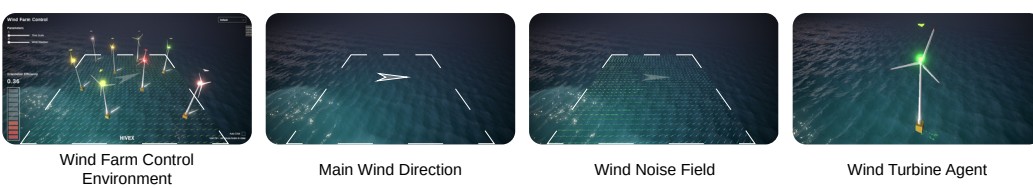

| Wind Farm Control Environment | Main Wind Direction | Wind Noise Field | Wind Turbine Agent |

Figure 1: Wind Farm Control main environment features. Details in the Appendix 11.

Table 1: Environment Specifications: Wind Farm Control

| Category | Parameter | Description/Value |
|---|---|---|
| General | Episode Length | 5000 |
| | Agent Count | 8 |
| | Neighbour Count | 0 |
| Vector Observations (6) | Stacks | 1 |
| | Normalized | True |
| | Turbine Location (2) | $\vec{p}(x, y)$ |
| | Turbine Direction (2) | $\vec{dir}(x, y)$ |
| | Wind Direction (2) | $\vec{wdir}(x, y)$ |
| Visual Observations (0) | - | - |
| Continuous Actions (0) | - | - |
| Discrete Actions (1) | Rotate Turbine | {0: Do Nothing, 1: Rotate Left, 2: Rotate Right} |

### 2.1.2 MAIN TASK AND REWARDS

Generate Energy - The agent's goal is to rotate the wind turbine to be oriented against the wind direction and generate energy. The agent receives a positive reward in the range of $[0, 1]$ at each time step. This reward corresponds to the performance of each wind turbine and is being calculated as described in equation 4. Orienting the wind turbine against the wind yields a high reward.

A comprehensive task list and description for the Wind Farm Control environment can be found in the Appendix A.9.1. We also provide extensive reward description and calculation in the Appendix A.8.1.

## 2.2 WILDFIRE RESOURCE MANAGEMENT

### 2.2.1 ENVIRONMENT SPECIFICATION

Table 2: Environment Specifications: Wildfire Resource Management

| Category | Parameter | Description/Value |
|---|---|---|
| General | Episode Length | 500 |
| | Agent Count | 9 |
| | Neighbour Count | 3 |
| Vector Observations (16) | Stacks | 2 |
| | Normalized | True |
| | Closest Fire Location (3) | $\vec{p}(x, y, z)$ |
| | Temperature (1) | $t$ |
| | Humidity (1) | $h$ |
| | Overcast (1) | $o$ |
| | Total Support (1) | $ts$ |
| Visual Observations (0) | - | - |
| Continuous Actions (0) | - | - |
| Discrete Actions (4) | Add/Sub Resource: Self | {0: No Action, 1: Add, 2: Sub} |
| | Add/Sub Resource: Neighbour 1 | {0: No Action, 1: Add, 2: Sub} |
| | Add/Sub Resource: Neighbour 2 | {0: No Action, 1: Add, 2: Sub} |
| | Add/Sub Resource: Neighbour 3 | {0: No Action, 1: Add, 2: Sub} |

### 2.2.2 MAIN TASK AND REWARDS

Resource Distribution - At each time step, the agent distributes a total of 1.0 resource units, in increments of 0.1, to either itself or neighbouring watchtowers. If the agent runs out of resources, it must first reallocate resources from itself or neighbouring watchtowers before redistributing. The agent's

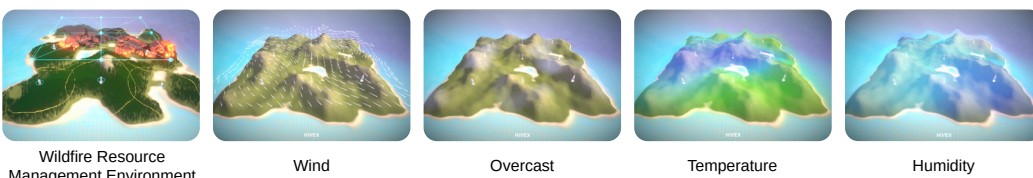

| Wildfire Resource Management Environment | Wind | Overcast | Temperature | Humidity |
|---|---|---|---|---|

Figure 2: Wildfire Resource Management main environment features. Details in the Appendix 13.

priority is to allocate resources to the watchtowers closest to and most threatened by incoming fires. The agent earns rewards based on three factors. First, it receives a positive reward corresponding to the performance of the watchtower it controls, weighted by the amount of resources allocated to itself, as described in Equation 9. Second, the agent also gains a reward based on the performance of neighbouring watchtowers, which is weighted by the resources allocated to them, as outlined in Equation 10. Additionally, extra rewards are given for distributing resources effectively to neighbouring watchtowers. Finally, the agent's overall reward includes a component that reflects the sum of the performance of all agent-controlled watchtowers, detailed in Equation 12.

For more detailed information on the task descriptions and reward calculations, please refer to the Appendix (A.9.2) and (A.8.2).

## 2.3 OCEAN PLASTIC COLLECTION

### 2.3.1 ENVIRONMENT SPECIFICATIONS

Table 3: Environment Specifications: Ocean Plastic Collection

| Category | Parameter | Description/Value |
|---|---|---|
| General | Episode Length | 5000 |
| | Agent Count | 3 |
| | Neighbour Count | 1 |
| Vector Observations (12) | Stacks | 2 |
| | Normalized | True |
| | Local Position (2) | $\vec{p}(x, y)$ |
| | Direction (2) | $\vec{dir}(x, y)$ |
| | Closest Neighbouring Vessel (2) | $\vec{np}(x, y)$ |
| Visual Observations (1250) | Resolution | 25x25x1 |
| | Stacks | 2 |
| | Normalized | True |
| | Trash | $t = [0, 1]$ |
| Continuous Actions (0) | - | - |
| Discrete Actions (2) | Throttle | {0: Do Nothing, 1: Accelerate} |
| | Steer | {0: Do Nothing, 1: Turn Right, 2: Turn Left} |

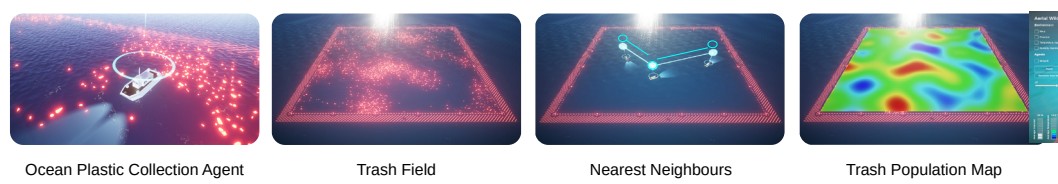

| Ocean Plastic Collection Agent | Trash Field | Nearest Neighbours | Trash Population Map |
|---|---|---|---|

Figure 3: Ocean Plastic Collection main environment features. Details in the Appendix 17.

### 2.3.2 MAIN TASK AND REWARDS

Plastic Collection - The agent aims to accelerate and steer the plastic collection vessel to collect as many floating plastic pebbles as possible while avoiding crashing into other vessels and crossing the environment's border. The agent receives a positive reward of 1 for each floating plastic pebble collected. Furthermore, the agent receives a positive reward for the lowest collected trash count amongst all agents at each time step. The lowest trash count is scaled by 0.01. The steps to calculate the lowest collected trash count reward can be found in Equation 15. Finally, the agent receives a negative reward of $-100$ when the border is crossed.

A comprehensive task list and description for the Ocean Plastic Collection environment can be found in the Appendix A.9.3. We also provide extensive reward description and calculation in the Appendix A.8.3.

## 2.4 DRONE-BASED REFORESTATION

### 2.4.1 ENVIRONMENT SPECIFICATIONS

Table 4: Environment Specifications: Drone-Based Reforestation

| Category | Parameter | Description/Value |
|---|---|---|
| General | Episode Length | 2000 |
| | Agent Count | 3 |
| | Neighbour Count | 0 |
| Vector Observations (20) | Stacks | 2 |
| | Normalized | True |
| | Distance to Ground (1) | $dg$ |
| | Local Position (3) | $\vec{p}(x, y, z)$ |
| | Direction (3) | $\vec{dir}(x, y, z)$ |
| | Drone Station Height (1) | $dsh$ |
| | Holding Seed (1) | $hs = [0, 1]$ |
| | Energy Level (1) | $el$ |
| Visual Observations (256) | Resolution | 16x16x1 |
| | Stacks | 1 |
| | Normalized | True |
| | Downward Pointing Camera | Grayscale (256), $t = [0, 1]$ |
| Continuous Actions (3) | Throttle | $[-1, 1]$ |
| | Steer | $[-1, 1]$ |
| | Up/Down | $[-1, 1]$ |
| Discrete Actions (1) | Drop Seed | {0: Do Nothing, 1: Drop Seed} |

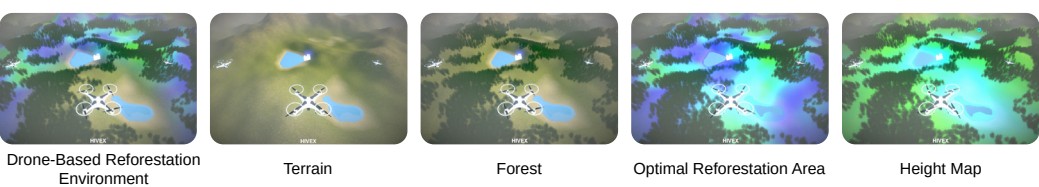

| Drone-Based Reforestation Environment | Terrain | Forest | Optimal Reforestation Area | Height Map |

Figure 4: Drone-Based Reforestation main environment features. Details in the Appendix 15.

### 2.4.2 MAIN TASK AND REWARDS

Maximizing Collective Tree Count - The agent's primary objective is to pick up seeds and recharge at the drone station, explore fertile ground near existing trees, and drop seeds while ensuring sufficient battery charge to return to the station. For each successful seed drop, the agent receives a reward based on two components: the quality of the drop location and its proximity to other seeds and trees. The seed quality reward ranges from 0 to 20, while the distance reward ranges from 0 to

10, giving a total possible reward of 0 to 30 for each drop. These calculations are detailed in Equation 32. When carrying a seed, the agent incurs a time-step penalty of $-1/(episode * length/2)$, with energy depletion penalties being higher when a seed is carried. If the drone is not carrying a seed, the penalty is $-1/episode * length$. The episode length is 2000 time steps. Additionally, the agent can receive a bonus for returning to the drone station. After a seed drop, the agent is also rewarded incrementally for reducing the distance to the station, with steps of 2.5. The incremental return reward ranges from 0 to 20 and is adjusted by a multiplier based on the seed drop quality. For example, if a seed is dropped 50 meters from the station, up to 20 incremental rewards may be received. The calculation of this reward is described in Equation 40.

Detailed descriptions of tasks and rewards for the Drone-Based Reforestation environment are available in the Appendix A.9.4 and A.8.4.

## 2.5 AERIAL WILDFIRE SUPPRESSION

### 2.5.1 ENVIRONMENT SPECIFICATIONS

Table 5: Environment Specifications: Aerial Wildfire Suppression

| Category | Parameter | Description/Value |
|---|---|---|
| General | Episode Length | 3000 |
| | Agent Count | 3 |
| | Neighbour Count | 0 |
| Vector Observations (8) | Stacks | 1 |
| | Normalized | True |
| | Local Position (2) | $\vec{p}(x, y)$ |
| | Direction (2) | $\vec{dir}(x, y)$ |
| | Holding Water (1) | $hw = [0, 1]$ |
| | Closest Tree Location (2) | $\vec{ct}(x, y)$ |
| | Closest Tree Burning (1) | $ctb = [0, 1]$ |
| Visual Observations (1764) | Resolution | 42x42x3 |
| | Stacks | 1 |
| | Normalized | True |
| | Downward Pointing Camera | RGB, $[r, g, b] = [[0, 1], [0, 1], [0, 1]]$ |
| Continuous Actions (1) | Steer Left/Right | $[-1, 1]$ |
| Discrete Actions (1) | Drop Water | {0: Do Nothing, 1: Drop Water} |

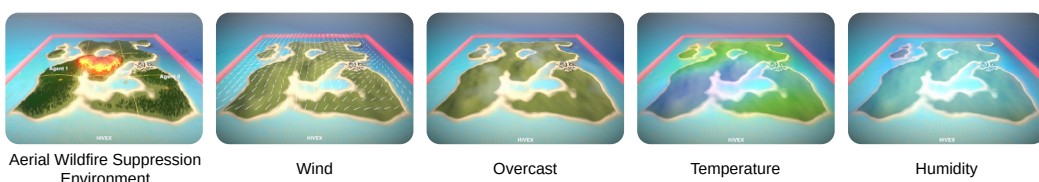

| Aerial Wildfire Suppression Environment | Wind | Overcast | Temperature | Humidity |

Figure 5: Aerial Wildfire Suppression main environment features. Details in the Appendix 19.

### 2.5.2 MAIN TASK AND REWARDS

Minimize Fire Duration and Protect the Village - The agent's primary goal is to pick up water and extinguish as many burning trees as possible or prepare unburned forest areas to prevent the spread of fire. A secondary goal is to protect the village by preventing fire from getting too close, either by extinguishing burning trees or redirecting the fire through tree preparation. Crossing the environment's boundary (a 1500x1500 square surrounding a 1200x1200 island) results in a negative reward of $-100$. Steering the aeroplane towards the surrounding water girdle (300 units wide) earns a positive reward of 100. There is also a small time-step penalty of $-1/MaxStep$. If the fire across the entire island is extinguished, with or without agent intervention, a positive reward of 10 is given. If

the fire reaches within 150 units of the village centre, the agent receives a penalty of $-50$.

A detailed task list and reward breakdown for the Aerial Wildfire Suppression environment is provided in the Appendix (A.9.5), along with further information on reward calculations in the Appendix (A.8.5).

# 3 RELATED WORK

While the HIVEX environments can be situated close to some existing MARL benchmarks in the domain of UAVs Lv et al. (2023); Cui et al. (2020); Qie et al. (2019); Pham et al. (2018), energy supply Riedmiller et al. (2001) and resource handling Han & Arndt (2021); Perolat et al. (2017); Ben Noureddine et al. (2017), we believe there is a gap for critical ecological challenges such as wildfires MacCarthy et al. (2022); Tyukavina et al. (2022), pollution WEF (2016) and deforestation Dow Goldman et al. (2020).

Many environment suits available are grid-based and have very simple 2D visual representations such as Level-Based Foraging Christianos et al. (2021), PressurePlate, Multi-Robot Warehouse (RWARE) Papoudakis et al. (2021), Pommerman Resnick et al. (2022), or Overcooked Carroll et al. (2020) and many more. By enriching the visual representation of these environments and reducing the level of abstraction, we believe we can attract a broader range of disciplines to engage with the HIVEX environments suite.

Procedurally generating environment features, such as level design, tasks Vinyals et al. (2019); Berner et al. (2019), and agent populations have been adopted in various environment suits, such as Meltingpot Leibo et al. (2021), Neural MMO Suarez et al. (2019) and Capture the Flag Jaderberg et al. (2019). We procedurally generate terrains in various terrain elevation levels for Wildfire Resource Management, Drone-Based Reforestation and Aerial Wildfire Suppression environments 23. The environments Wind Farm Control and Ocean Plastic Collection utilize noise maps and random sampling 21, 22, 23, 24, 25.

DeepMind's work Melting Pot is a suite of test scenarios for multi-agent reinforcement learning emphasising social situations Leibo et al. (2021). While we do not directly target social aspects in our environments, our previous work has shown significant performance improvements when introducing communication mechanisms in earlier versions of HIVEX environments ANONYMIZED. However, Melting Pot, with its 50 substrates (environments) and 256 unique scenarios (tasks), has influenced the structural design of our environment suite.

Work such as Neural MMO or LUX Chen et al. (2023) focuses on efficient large agent number environments. However, we believe that this is not as important for our work, as the scenarios we have presented do not require large amounts of agents. Nevertheless, we have shown that our environments scale well across increasing numbers of agents.

There is a trade-off between simulated environments and experience samples from the real world. While The latter might be expensive, mixtures of both can lead to success Shashua et al. (2021). HIVEX focuses on simulated environments. However, we would like to shorten the sim-to-real gap in future work.

# 4 EXPERIMENTS AND RESULTS

We have trained and tested all environments across all tasks and terrain elevation levels or patterns three times and report the average and the error margin 6. The test runs represent the baseline for the HIVEX environment suite. Extensive results can be found in the Appendix in the section Additional Results A.10. Furthermore, all checkpoints and logs can be found in the hivex-results repository. We have used Proximal Policy Optimization (PPO) Schulman et al. (2017) for all train and test runs (Appendix: Learning Algorithm A.4.1). We provide hyperparameters for training in the Hyperparameters section A.5.

We tested the scalability of selected HIVEX environments with larger agent numbers, including Wind Farm Control, Drone-Based Reforestation, and Aerial Wildfire Suppression. Wildfire Resource Management and Ocean Plastic Collection were excluded from scalability tests: the former has a fixed layout and agent count, while the latter's fixed amount of floating plastic would re-

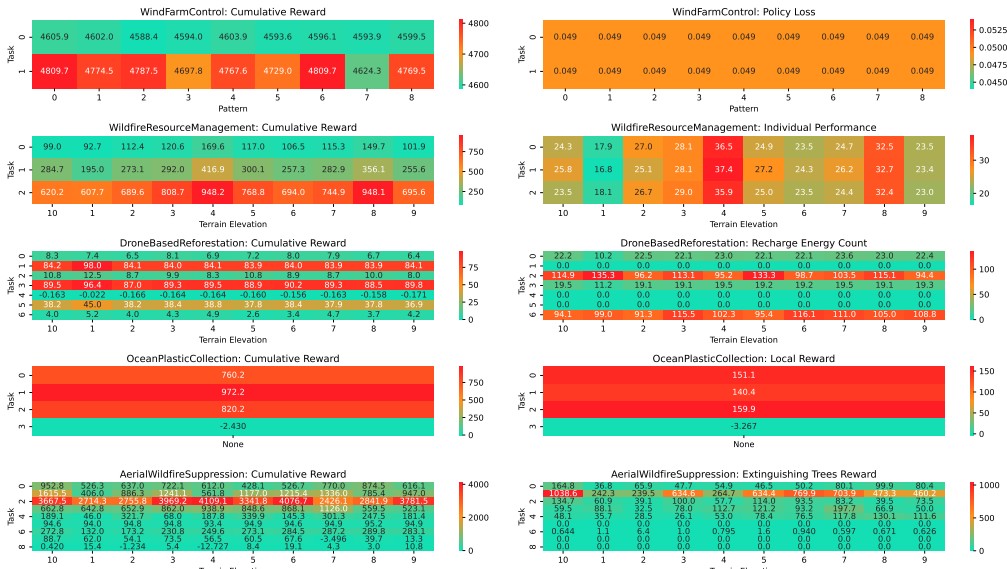

Figure 6: Average test results for all environments for Cumulative Reward and environment-specific metrics such as 1. Wind Farm Control: Policy Loss, 2. Wildfire Resource Management: Individual Performance is the isolated individual performance, 3. Drone-Based Reforestation: Recharge Energy Count, which indicates how often a drone returned to the drone station to recharge energy and pick up a new seed; 4. Ocean Plastic Collection: Local Reward, which is the reward for collecting plastic pebbles, 5. Aerial Wildfire Suppression: Extinguishing Trees Reward.

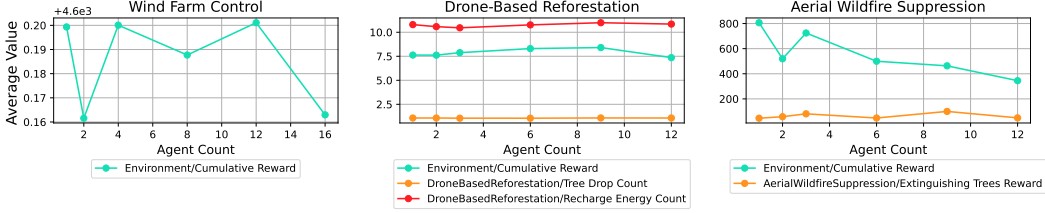

Figure 7: Agent Number Scalability Test of Wind Farm Control, Drone-Based Reforestation, and Aerial Wildfire Suppression environments.

duce per-agent performance with an increased agent count. Wind Farm Control has been tested on $[1, 2, 4, 8, 12, 16]$, Drone-Based Reforestation and Aerial Wildfire Suppression on $[1, 2, 3, 6, 9, 12]$ agent counts 7.

## 5 DISCUSSION

The cumulative reward performance in Wind Farm Control exhibits a stable trajectory across various layout patterns, indicating a well-optimized policy that effectively manages changing wind conditions. Despite minor fluctuations, the overall trend remains consistent across different tasks.

In Wildfire Resource Management, cumulative rewards show greater variability as task difficulty increases. Although rewards initially rise with terrain elevation levels, they plateau and fluctuate at higher levels, such as 4 and 8, marking the highest recorded reward. A higher terrain elevation level has steeper mountains and a more structured but sparse distribution of forest volume along mountain ranges. This suggests the model struggles in open fields where fire behaviour is less predictable. Nevertheless, the model performs reasonably in most scenarios, demonstrating its adaptability in

real-world wildfire resource allocation. This trend is further evident in the individual performance data.

The Drone-Based Reforestation task demonstrates relatively stable but declining cumulative rewards, indicating the model's efficiency in reforestation efforts despite struggling in more challenging scenarios involving steep terrain and sparse forest areas. The "Recharge Energy Count" metric remains steady, even as terrain elevation increases, suggesting that while the agent struggles to find optimal drop locations, it maintains consistent drop and recharge activity. This metric's stability across tasks suggests potential for improvement, such as testing more energy-demanding tasks or introducing tighter energy consumption constraints.

In Aerial Wildfire Suppression, task performance appears highly sensitive to terrain elevation, with rewards dropping as complexity increases. While the model performs well in scenarios with sparse forest volume and limited fire spread, it struggles in scenarios with denser forests where fires can spread in all directions. As in other tasks, higher terrain elevation reflects steeper terrain and sparser forest distribution, requiring more frequent water drops as fires spread more unpredictably. The "Extinguishing Trees Reward" metric also reflects this variability, emphasizing the need for refined strategies, such as pre-wetting trees to direct the fire in lower-terrain elevation scenarios.

Overall, the baseline model demonstrates varying success across difficulties and environments. The baseline results indicate that the model efficiently learns routine conditions, but its performance declines as the complexity of the tasks increases. This indicates that the environments effectively introduce new challenges across scenarios, patterns, or terrain elevation levels. Future work should focus on adding even more difficult scenarios and edge cases.

The scalability analysis reveals that multi-agent systems in all three environments - Wind Farm Control, Drone-Based Reforestation, and Aerial Wildfire Suppression - exhibit stable and positive performance trends as agent counts increase. In Wind Farm Control, the cumulative reward remains stable across all tested agent counts, indicating that the system scales effectively without significant performance degradation.

In Drone-Based Reforestation, the cumulative reward scales well, with only a minor decrease beyond 9 agents. Tree drop counts remain stable, reflecting consistent performance, while energy consumption shows a slight upward trend, demonstrating good scalability with manageable resource trade-offs.

For Aerial Wildfire Suppression, the cumulative reward is generally stable as agent numbers increase, with a slight dip before recovering toward 12 agents. The extinguishing reward follows a similar pattern, showing an upward trend as agents increase, indicating that the system scales well despite minor fluctuations. Overall, these environments demonstrate good scalability across agent counts with only minor trade-offs in specific metrics 7.

## 6 LIMITATIONS AND POTENTIAL IMPACTS

While our simulations provide a valuable foundation for MARL research in addressing critical ecological challenges, several limitations may affect their generalizability and real-world applicability. One major limitation is how accurately these simulations represent real-world scenarios. Despite efforts to closely model actual environments, simulations inevitably simplify complex conditions, often failing to capture unexpected environmental variables and interactions with dynamic objects. For instance, turbines in the Wind Farm Control environment can be turned much faster than in reality, and wind directions shift too quickly and randomly. In contrast, real-world wind tends to have a predominant direction in specific regions. In the Ocean Plastic Collection environment, vessel turning and acceleration speeds are significantly exaggerated. Similarly, in the Reforestation environment, agents can pick up seeds simply by being near the drone station, which does not reflect real-world conditions. Fire spreads much faster in the Wildfire Resource Management and Aerial Wildfire Suppression environments. Specifically, resources are distributed too quickly in the Wildfire Resource Management environment, while the claim is that the scenarios are in remote areas. Additionally, water-carrying planes turn much faster than would be possible in reality, even when fully loaded. Furthermore, the camera feed resolution in the Drone-Based Reforestation and Aerial Wildfire Suppression environments is lower than what would be needed in practice. Although the simulations perform well with low resolution, we anticipate more challenges with diverse objects in real-world scenarios.

These discrepancies could impact the real-world applicability of our findings, but there are still promising areas for implementation. For instance, algorithms developed in the Wind Farm Control environment, despite their simplified wind patterns, could contribute to optimizing wind farm layouts and improving maintenance strategies, as seen in efforts by companies like Siemens Gamesa, which integrates AI for predictive maintenance in real wind farms Su et al. (2023). Similarly, wildfire management strategies derived from simulations, though faster than real-world conditions, could assist in resource distribution planning and suppression tactics, akin to systems used by CAL FIRE in the United States Hernandez & Hoskins (2024). Lastly, despite its simplified nature, our reforestation environment could enhance large-scale efforts such as the Great Green Wall initiative in Africa, which seeks to restore degraded lands using new technologies Gravesen & Funder (2022). These applications demonstrate the potential utility of our simulations when combined with real-world data and in-field validation.

A key limitation of the current environment design is its potential for bias, as the terrains and landscapes are generated within a single climate zone. This restricts the diversity of environmental conditions, excluding deserts, rocky regions, and other ecosystems with distinct flora and fauna. To address this, future work could incorporate real geographic data from diverse global regions, including terrain, forest structure, and environmental variables like wind speed, precipitation, temperature, and cloud cover. Collaboration with companies and research labs will also be necessary to adjust agent-controlled objects to align with real-world capabilities. However, for specific applications such as wildfire or reforestation simulations, only certain areas of the world are particularly relevant, which naturally limits the range of applicable environments. For instance, wildfire simulations are most pertinent in regions such as Russia, Canada, and the United States, which experience the highest tree cover loss due to fires Tyukavina et al. (2022). Conversely, reforestation efforts are more urgent in areas like the Sahara, the Zinder and Maradi regions Pausata et al. (2020), and the Amazon Rainforest Dow Goldman et al. (2020). Thus, while the HIVEX environment suite offers a promising starting point, fine-tuning based on real-world data is essential to achieve meaningful real-world applications.

The HIVEX environment suite is designed for training and testing on accessible end-user hardware. Our simulations have been successfully executed on systems with an NVIDIA GeForce RTX 3090, an AMD Ryzen 9 7950X 16-Core Processor, and 64 GB of RAM specifications within the range of many gaming laptops and desktop computers. As such, researchers and practitioners do not need specialized, large-scale computational clusters, making our approach accessible to those with mid-range to high-end consumer hardware. Future optimizations could further reduce these requirements for even broader accessibility.

## 7 CONCLUSION

The HIVEX suite is a novel open-source benchmark that simulates real-world critical ecological challenges. Through procedurally generated environments and adjustable layout patterns or terrain elevation levels, it supports multi-agent and open-ended research across diverse tasks and scenarios. The wide range of environments, tasks and scenarios provide a broad spectrum of challenges and makes HIVEX a valuable tool for testing algorithm generalizability. Future work aims to narrow the sim-to-real gap by incorporating real-world data, such as terrain and weather conditions.

ACKNOWLEDGMENTS

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

918
919

# A APPENDIX

920
921

## A.1 RESOURCES

922
923
924
925
926

- NVIDIA GeForce RTX 3090
- Driver version 536.23
- AMD Ryzen 9 7950X 16-Core Processor
- 64 GB RAM

927
928

## A.2 MOTIVATION: CRITICAL ECOLOGICAL CHALLENGES

929
930
931
932
933
934
935
936
937
938
939
940
941
942
943
944
945
946
947

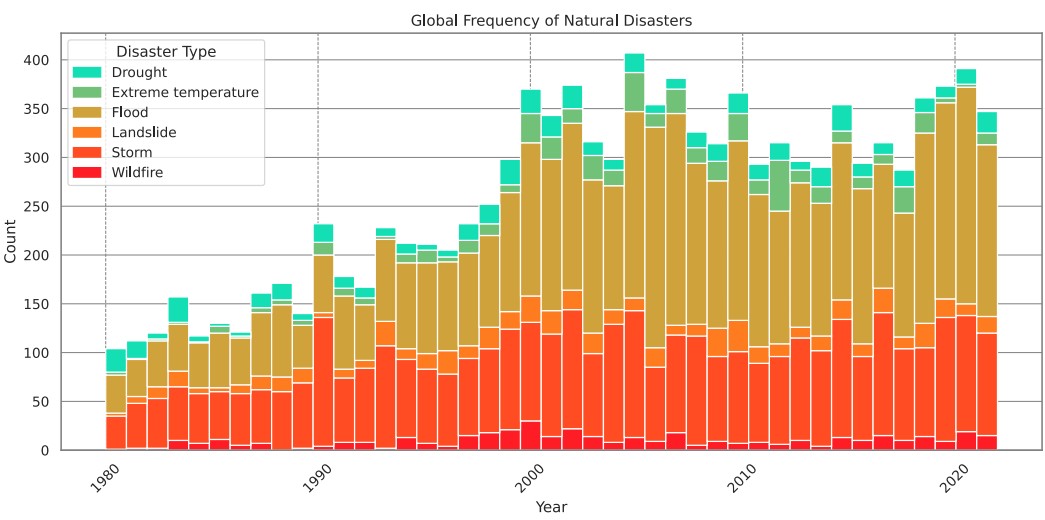

948
949
950
951

Figure 8: Climate-related global disasters frequency. The links between climate change and natural disasters are well documented in a wide variety of climate change literature. This graph depicts the trend in global climate-related disasters over time. Interactive plot and dataset can be explored here: `https://climatedata.imf.org/pages/climatechange-data`.

952
953
954
955
956
957
958
959
960
961

Climate change is manifesting more visibly and urgently than ever Archer & Rahmstorf (2010); Romm (2022). We are witnessing an increase in frequent and intense weather phenomena, such as storms, droughts, fires, and floods UCLouvain (2023). Figure 8 shows the aforementioned disaster types triple in frequency between 1980 and 2020. These events are reshaping ecosystems and critically impacting agriculture and natural resources, which are vital to human survival Change (2012). A concerning report by the Intergovernmental Panel on Climate Change (IPCC) in 2022 highlights the dire consequences of continued greenhouse gas emissions, warning that significant curbing measures are needed within the next three decades to avert catastrophic impacts. If the 1.5 °C degree increase in global warming cannot be negated, some impacts may be long-lasting or irreversible, such as the loss of ecosystems potentially fundamental to our existence Ipcc (2022).

962
963
964

### MITIGATION, ADAPTATION AND DISASTER RESPONSE

965
966

The battle against climate change encompasses three critical approaches: mitigation, adaptation and disaster response Commission (2022).

967
968
969
970
971

- Mitigation focuses on reducing emissions through transformative measures in electricity generation, transportation, building design, industry practices, and land use.
- Adaptation, on the other hand, is about enhancing resilience and improving disaster management strategies to prepare for the inevitable impacts of changing climate patterns.
- Disaster Response involves prompt and effective measures to manage emergencies caused by climate-related events. This includes providing immediate relief, medical aid, and re-

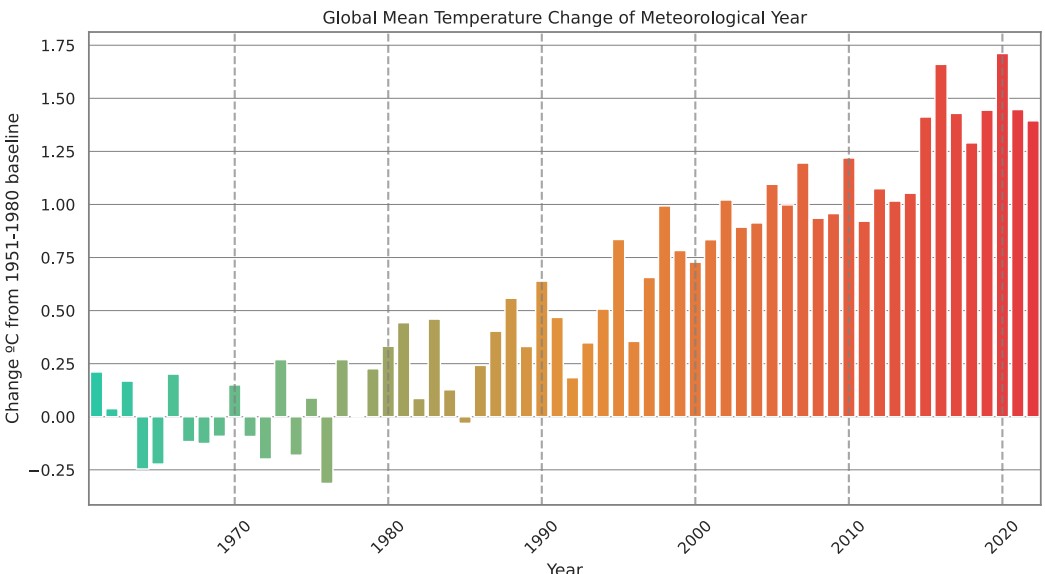

Figure 9: Annual Global surface temperature change. This indicator presents the global mean surface temperature change during the period 1961-2021, using temperatures between 1951 and 1980 as a baseline. This data is provided by the Food and Agriculture Organization Corporate Statistical Database (FAOSTAT) and is based on publicly available GISTEMP data from the National Aeronautics and Space Administration Goddard Institute for Space Studies (NASA GISS). Interactive plot and dataset can be explored here: `https://climatedata.imf.org/pages/climatechange-data`.

> construction assistance and implementing policies for rapid response and recovery to minimize the impact on affected communities.

This tripartite approach is essential, as highlighted by the IPCC report and echoed in the research by Collins et al. (2018), underscoring the importance of addressing both immediate and long-term aspects of climate change.

IRREVERSIBILITY

Recent research underscores the alarming irreversibility of certain impacts of climate change. A study at Arizona State University, published in the Proceedings of the National Academy of Sciences, explores the concept of 'rate-induced tipping' in ecological systems Panahi et al. (2023). This research is crucial in understanding when certain environmental systems, such as coral reefs, may reach a point of irreversible damage Hughes et al. (2018).
As ocean temperatures rise due to increased carbon emissions Venegas et al. (2023), corals and their symbiotic zooxanthellae (tiny cells that live within most types of coral polyps - they help the coral survive by providing it with food resulting from photosynthesis) are pushed towards a threshold beyond which severe bleaching occurs Sully et al. (2019), leading to a cascade of effects on the entire reef ecosystem. This bleaching, once initiated, cannot be reversed even if ocean temperatures were to subsequently stabilize, illustrating the permanent nature of some climate change impacts. The study emphasizes that even gradual changes in environmental parameters can suddenly trigger catastrophic system collapses, highlighting the urgency of addressing climate change proactively to prevent irreversible ecological damage Panahi et al. (2023).

TIMELINE AND URGENCY

The timeline for addressing climate change is critical and urgent. According to the latest insights, there's a pressing need to accelerate climate action significantly to limit global temperature rise

to 1.5 degrees Celsius. This target requires deep, rapid, and sustained greenhouse gas emissions reductions across all sectors within this decade. Emissions need to decrease immediately to stay within these limits and be cut by nearly half by 2030 Calvin et al. (2023). Figure 9 shows the global surface temperature change in Celsius degrees per year from the baseline temperature between 1951 and 1980 of the United Nations (1997).

The 2023 Yearbook of Global Climate Action, presented at the UN Climate Change Conference (COP28) Hughes et al. (2018), emphasizes the urgency of scaling up climate actions. It highlights the increase in stakeholders taking climate action but also points out that the pace and scale of these actions are insufficient to meet the 1.5-degree Celsius target. The Yearbook calls for accelerated, effective implementation of climate actions, emphasizing the critical role of governments in reducing barriers to lowering greenhouse gas emissions and the need for transformational changes in sectors like food, electricity, transport, industry, buildings, and land use.

A major UN report, "Climate Change 2023: Synthesis Report" by the Intergovernmental Panel on Climate Change (IPCC) Calvin et al. (2023), underlines the significant impacts already being felt globally and the increased frequency of extreme weather events due to climate change. The report stresses the necessity of integrating adaptation to climate change with actions to reduce or avoid greenhouse gas emissions. It also points out the importance of financial and technical support for developing countries from wealthier nations to achieve these goals De-Arteaga et al. (2018).

ROLE OF MACHINE LEARNING

The vast array of challenges presented by climate change also opens diverse opportunities for impactful action Kaack (2019); Ford et al. (2016). While the situation is grave, there is immense potential for innovative solutions in areas such as renewable energy, sustainable agriculture, and resource-efficient industrial practices. The commitment to tackling these challenges is about averting disaster and harnessing the opportunity for significant environmental, economic, and social progress Berendt (2019); Hager et al. (2019).

The last two years have brought climate change to the doorstep of many. Extreme heatwaves, wildfires, and floods make life increasingly difficult for animals and humans De-Arteaga et al. (2018). ML has emerged as a key tool for technological advancement in recent years. As ML and artificial intelligence (AI) use in societal and global initiatives grows, there's a pressing need to explore how these technologies can best address climate change challenges. Many in the ML field are eager to contribute but unsure of the best approach, while various sectors are increasingly seeking ML expertise.

ML has many applications in combating climate change for various time horizons and degrees of impact Rolnick et al. (2022); Ladi et al. (2022). Straight forward applications However, we think it's crucial to acknowledge its fundamental role in enhancing our understanding of climate complexities Yu et al. (2013); Faghmous & Kumar (2014). ML, with its advanced data analysis capabilities, is instrumental in deciphering the multifaceted nature of climate data. It aids scientists and researchers in identifying patterns and trends that are not immediately apparent, providing insights into phenomena like temperature changes, precipitation patterns, and extreme weather events Climate TRACE - (2022). This deepened understanding is the bedrock upon which targeted solutions for climate change mitigation and adaptation are developed.

In the critical battle against climate change, ML emerges as a pivotal ally, offering a diverse array of contributions across various domains. By enabling automatic monitoring through remote sensing, ML helps in identifying key environmental changes, such as deforestation, and in assessing post-disaster damages. This technology is particularly significant in the realm of ecosystem informatics and sustainability, where it aids in understanding complex ecological dynamics and biodiversity, supporting conservation efforts and sustainable resource management Dietterich (2009); Gomes et al. (2019); Lässig et al. (2016). ML's ability to process vast amounts of ecological data enhances our capacity to track species populations, monitor habitat changes, and predict ecological responses to various environmental stressors.

Further, ML accelerates scientific discovery, suggesting innovative materials for batteries, construction, and carbon capture technologies. Ecosystem informatics enables the identification of patterns and relationships within ecological systems, facilitating the development of strategies to protect and sustain these vital systems. Additionally, ML optimizes systems for enhanced efficiency, evident

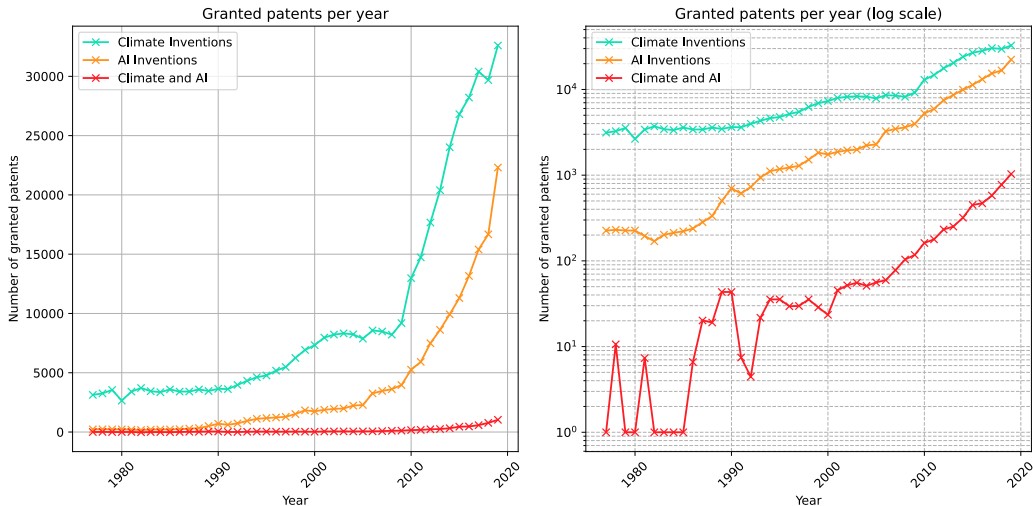

Figure 10: Left: Granted patents per year, with a steeper rise starting around 2010. Right: The rise on the left can be seen as exponential growth in climate AI patents (linear on a log scale), and this holds for climate patents and AI patents separately. Within climate patents, however, AI patents are not growing exponentially. Verendel (2023); Angelucci et al. (2018)

in applications like freight consolidation, carbon market design, and reduction of food waste Joppa (2017). Its ability to accelerate computationally intense physical simulations, like climate and energy scheduling models, is invaluable. The integration of ML in these areas not only addresses immediate environmental concerns but also fosters long-term sustainability and resilience of ecosystems, thus playing a crucial role in mitigating the impacts of climate change. Figure 10 shows an increase of patents granted for climate inventions, AI inventions and climate and AI between 1970 and 2020. This means we can directly link advancements in AI to innovation in climate-related topics.

The integration of ML in climate change mitigation not only benefits society but also propels advancements in ML itself, particularly in areas such as interpretability, causality, and uncertainty quantification. However, the challenge lies in the nature of climate-relevant data, which is often proprietary, sensitive, or not globally representative. Solutions like transfer learning and domain adaptation become crucial in addressing these data challenges. We aim to emphasize the significant potential that advancing state-of-the-art ML, utilizing real-world data and simulation environments, can go hand in hand with developing effective solutions for current pressing challenges.

### A.3 NATURAL SOCIETIES AND MULTI-AGENT RESEARCH

In MARL environments, groups of agents with baseline intelligence and ability can have a higher collective intelligence by acting together Cohen et al. (1997). A shared pool of information through a collective observation space can help individual agents to learn quicker. Additionally, as a group, they can achieve objectives that would be challenging to attain individually Guestrin et al. (2002); Decker (1987); Panait & Luke (2005); MATARIC (1998). However, acting as a collective requires collaboration. From the perspective of an individual agent, other agents in the collective and the consequences of their actions, i.e. change of the environment, can be seen as part of a dynamic environment Ravula et al. (2019). Perceiving others' actions and making sense of their intention is called intention reading, stated in the theory-of-mind (ToM) Hernandez-Leal et al. (2019). While this is an integral part of human collaborative activities, we will assume shared intentionality Tomasello et al. (2005).

In our quest to advance multi-agent systems and cooperative strategies, the study of animal societies like ants and meerkats offers invaluable lessons. These natural societies, characterized by intricate cooperation and complex social structures, provide a blueprint for understanding and designing efficient, self-organizing systems in human contexts.

**Ants and Cooperative Robots**: Researchers at Harvard University explored how ants cooperate to solve complex problems like transporting and building things using simple rules. They studied black carpenter ants and created a simulation to model their cooperative behaviour. This model was then used to develop robot ants (RAnts) that demonstrated similar cooperative behaviours to real ants, highlighting the potential for applying natural cooperation strategies in robotics Prasath et al. (2022). Recent work of ours explores distributed robotics for building architectural structures ANONYMIZED, in which robotics help each other to climb, add and remove bespoke building blocks for a dynamically changing spatial configuration.

**Ant Colonies and Social Evolution**: Certain ant species, which do not have a leader, can exhibit complex behaviours like the division of labour through self-organization. This challenges the notion that strong groups require strong leaders and suggests that even in the simplest groups, significant collaboration can occur. This research has implications for understanding the evolution of social behaviour and the early stages of complex society formation Gordon (2010; 2002).

**Meerkats and Cooperation**: Meerkats have been studied to understand the role of testosterone in female competition and cooperative breeding. High testosterone levels in matriarch meerkats play a key role in their success and aggression, influencing the cooperative structure of the group. This study reveals that cooperation can also arise through aggressive means, shedding light on a new mechanism for the evolution of cooperative breeding Clutton-Brock et al. (2001); Muller & Wrangham (2004).

**Meerkat Society Study**: The Kalahari Meerkat Project, led by Professor Tim Clutton-Brock, provides extensive insights into meerkat societies. The project has tracked over 3,000 meerkats, examining their life histories and the effects of climate change on their survival and development. This long-term study offers valuable data on cooperative breeding, kinship, and the resilience of meerkat groups in challenging environments Komdeur et al. (2008); Newman et al. (2016).

In the context of nature, Charles Darwin argues for the survival of the fittest (Darwin, 1977) and, therefore, the occurrence of competition. While in AI, the majority of significant work on MA systems consider two opposing agents only, the problems of interest of this work are cooperative MA systems, where groups of agents act together to achieve higher individual and collective goals (Cohen et al., 1997; Guestrin et al., 2002; Decker, 1987; Panait & Luke, 2005; MATARIC, 1998). Just like in human society or the animal world, individuals have unique or mixtures of motives. However, we can define agents with mixed or identical motives in an MA environment simulation. Assuming shared intentionality leaves us with the question of how to collaborate. Communication can play a crucial role in collaborating successfully. Human society uses language as a communication medium (Barón Birchenall, 2016). Agents can send signals of various types as a form of language. Nevertheless, observing others' behaviour can be a form of communication. Body language, a tail-wagging dog, or the red colour of an octopus can communicate internal states and intentions. But we can also design agents that directly share policies - state action transitions - or memory data of past experiences.

### A.4    LEARNING ALGORITHM

Addressing the intricacies and challenges in multi-agent systems that operate in dynamic and complex environments requires a sophisticated blend of algorithms and methodologies. Our approach employs Proximal Policy Optimization (PPO) Schulman et al. (2017) with parameter sharing for MA training 2.

At the heart of our model is the policy $\theta$, represented by a neural network with parameters that process the observations from the environment, factoring in past states and producing actions as outputs. Within the context of the HIVEX suite, PPO offers a stable reinforcement learning algorithm, ensuring that agents iteratively refine their strategies without drastic deviations. This is crucial given the suite's dynamic environmental events, from wildfires to ocean cleanups. PPO is an advanced reinforcement learning algorithm that seeks to improve policy-based learning by ensuring that the updated policy does not deviate too drastically from the previous policy. This is achieved by adding a constraint or penalty to the objective function to restrict extreme policy updates 1.

**Proximal Policy Optimization**: Two main concepts define the PPO (Schulman et al., 2017), a state-of-the-art, on-policy RL algorithm: 1. PPO performs the largest possible but safe gradi-

ent ascent learning step by estimating a trust region and 2. Advantage estimates how good an action in a specific state is compared to the average action. A trust region can be calculated as the quotient of the current policy to be refined $\pi_\theta(a_t|s_t)$ and the previous policy as follows $r_t(\theta) = \frac{\pi_\theta(a_t|s_t)}{\pi_{\theta_k}(a_t|s_t)} = \frac{current\ policy}{old\ policy}$. The advantage is the difference between the Q and the Value Function: $A(s,a) = Q(s,a) - V(s)$, where $s$ is the state and $a$ the action (Zychlinski, 2019). The Q function measures the overall expected reward given state $s$, performing action $a$, and denoted as: $\mathcal{Q}(s,a) = \mathbb{E}\left[\sum_{n=0}^{N} \gamma^n r_n\right]$. The Value Function, similar to the Q Function, measures overall expected reward, with the difference that the State Value is calculated after the action has been taken and is denoted as: $\mathcal{V}(s) = \mathbb{E}\left[\sum_{n=0}^{N} \gamma^n r_n\right]$.

### A.4.1 PSEUDOCODE

PPO-CLIP pseudocode (OpenAI, 2021; Schulman et al., 2017):

---

**Algorithm 1**

---

Input: initial policy parameters $\theta_0$, initial value function parameters $\phi_0$
**for** $k = 0, 1, 2, \ldots$ **do**
    Collect set of trajectories $\mathcal{D}_k = \{\tau_i\}$ by running policy $\pi_k = \pi(\theta_k)$ in the environment.
    Compute rewards-to-go $\hat{R}_t$.
    Compute advantage estimates, $\hat{A}_t$ (using any method of advantage estimation) based on the current value function $V_{\phi_k}$
    Update the policy by maximizing the PPO-Clip objective:
    $\theta_{k+1} = \arg\max_\theta \frac{1}{|\mathcal{D}_k|T} \sum_{\tau \in \mathcal{D}_k} \sum_{t=0}^{T} \min\left(\frac{\pi_\theta(a_t|s_t)}{\pi_{\theta_k}(a_t|s_t)} A^{\pi_{\theta_k}}(s_t, a_t), g(\epsilon, A^{\pi_{\theta_k}}(s_t, a_t))\right),$
    typically via stochastic gradient ascent with Adam.
    Fit value function by regression on mean-squared error:
    $\phi_{k+1} = \arg\min_\phi \frac{1}{|\mathcal{D}_k|T} \sum_{\tau \in \mathcal{D}_k} \sum_{t=0}^{T} \left((V_\phi(s_t) - \hat{R}_t\right)$
    typically via some gradient descent algorithm.
**end for**

---

Simple Multi-Agent PPO pseudocode:

---

**Algorithm 2**

---

**for** $iteration = 1, 2, \ldots$ **do**
    **for** $actor = 1, 2, \ldots, N$ **do**
        Run policy $\pi_{\theta_{old}}$ in environment for $T$ time steps
        Compute advantage estimates $\hat{A}_1, \ldots, \hat{A}_T$
    **end for**
    Optimize surrogate $L$ wrt. $\theta$, with $K$ epochs and minibatch size $M \leq NT$
    $\theta_{old} \leftarrow \theta$
**end for**

---

## A.5 HYPERPARAMETERS

### A.5.1 HYPERPARAMETER DESCRIPTION

| Hyperparameter | Typical Range | Description |
|---|---|---|
| Gamma | $0.8 - 0.995$ | discount factor for future rewards |
| Lambda | $0.9 - 0.95$ | used when calculating the Generalized Advantage Estimate (GAE) |
| Buffer Size | $2048 - 409600$ | how many experiences should be collected before updating the model |
| Batch Size | $512 - 5120$ (continuous), $32 - 512$ (discrete) | number of experiences used for one iteration of a gradient descent update. |
| Number of Epochs | $3 - 10$ | number of passes through the experience buffer during gradient descent |
| Learning Rate | $1e - 5 - 1e - 3$ | strength of each gradient descent update step |
| Time Horizon | $32 - 2048$ | number of steps of experience to collect per-agent before adding it to the experience buffer |
| Max Steps | $5e5 - 1e7$ | number of steps of the simulation (multiplied by frameskip) during the training process |
| Beta | $1e - 4 - 1e - 2$ | strength of the entropy regularization, which makes the policy "more random" |
| Epsilon | $0.1 - 0.3$ | acceptable threshold of divergence between the old and new policies during gradient descent updating |
| Normalize | $true/false$ | weather normalization is applied to the vector observation inputs |
| Number of Layers | $1 - 3$ | number of hidden layers present after the observation input |
| Hidden Units | $32 - 512$ | number of units in each fully connected layer of the neural network |
| Intrinsic Curiosity Module | | |
| Curiosity Encoding Size | $64 - 256$ | size of hidden layer used to encode the observations within the intrinsic curiosity module |
| Curiosity Strength | $0.1 - 0.001$ | magnitude of the intrinsic reward generated by the intrinsic curiosity module |

Table 6: Hyperparameters Description: `https://github.com/Unity-Technologies/ml-agents/blob/main/docs/Training-Configuration-File.md`

### A.5.2 TRAIN AND TEST HYPERPARAMETERS: WIND FARM CONTROL

```
behaviors:
  Agent:
    trainer_type: ppo
    hyperparameters:
      batch_size: 256
      buffer_size: 2048
      learning_rate: 0.0003 # testing: 0.0
      beta: 0.005
      epsilon: 0.2
      lambd: 0.95
      num_epoch: 3
      learning_rate_schedule: linear # testing: constant
    network_settings:
      normalize: false
      hidden_units: 64
      num_layers: 2
    reward_signals:
      extrinsic:
        gamma: 0.9
        strength: 1.0
    keep_checkpoints: 5
    max_steps: 8000000 # testing: 8000000
    time_horizon: 2048
    summary_freq: 40000 # testing: 40000
    threaded: true

engine_settings:
  no_graphics: true

env_settings:
  env_path: /dev_environments/Hivex_WindFarmControl_win
  seed: 5000 # testing: 6000

environment_parameters:
  # Pattern: 0 Default, 1 Grid, 2 Chain, 3 Circle, 4 Square, 5 Cross,
  # 6 Two_Rows, 7 Field, 8 Random
  pattern: [0, 1, 2, 3, 4, 5, 6, 7, 8]
  task: [0, 1] # Generate Energy: 0, Avoid Damage: 1
```

### A.5.3 TRAIN AND TEST HYPERPARAMETERS: WILDFIRE RESOURCE MANAGEMENT

```
behaviors:
  Agent:
    trainer_type: ppo
    hyperparameters:
      batch_size: 128
      buffer_size: 2048
      learning_rate: 0.0003 # testing: 0.0
      beta: 0.01
      epsilon: 0.2
      lambd: 0.95
      num_epoch: 3
      learning_rate_schedule: linear # testing: constant
    network_settings:
      normalize: false
      hidden_units: 512
      num_layers: 2
      vis_encode_type: simple
    reward_signals:
      extrinsic:
        gamma: 0.99
        strength: 1.0
      curiosity:
        gamma: 0.99
        strength: 0.02
        encoding_size: 256
        learning_rate: 0.0003 # testing: 0.0
    keep_checkpoints: 5
    max_steps: 4500000 # testing: 450000
    time_horizon: 2048
    summary_freq: 4500 # testing: 4500
    threaded: true

engine_settings:
  no_graphics: true

env_settings:
  env_path: /dev_environments/Hivex_WildfireResourceManagement_win
  seed: 5000 # testing: 6000

environment_parameters:
  terrain_level: [1, 2, 3, 4, 5, 6, 7, 8, 9, 10]
  task: [0, 1, 2] # Main: 0, Distribute All: 1, Keep All: 2
```

### A.5.4 TRAINING HYPERPARAMETERS: DRONE-BASED REFORESTATION

```
behaviors:
  Agent:
    trainer_type: ppo
    hyperparameters:
      batch_size: 1024
      buffer_size: 10240
      learning_rate: 0.0003 # testing: 0.0
      beta: 0.005
      epsilon: 0.2
      lambd: 0.95
      num_epoch: 3
      learning_rate_schedule: linear # testing: constant
    network_settings:
      normalize: false
      hidden_units: 128
      num_layers: 2
      vis_encode_type: resnet
    reward_signals:
      extrinsic:
        gamma: 0.99
        strength: 0.9
        network_settings:
          vis_encode_type: resnet
      curiosity:
        gamma: 0.99
        strength: 0.1
        encoding_size: 256
        learning_rate: 0.0003 # testing: 0.0
        network_settings:
          vis_encode_type: resnet
    keep_checkpoints: 5
    max_steps: 2000000 # testing: 2000000
    time_horizon: 10240
    summary_freq: 10000 # testing: 10000
    threaded: true

engine_settings:
  no_graphics: true

env_settings:
  env_path: /dev_environments/Hivex_DroneBasedReforestation_win
  seed: 5000 # testing: 6000

environment_parameters:
  terrain_level: [1, 2, 3, 4, 5, 6, 7, 8, 9, 10]
  task: [0, 1, 2, 3, 4, 5, 6, 7]
  # Main: 0, Find Closest Tree: 1, Group Up: 2, Pick Up Seed: 3,
  # Drop Seed: 4, Find High Potential Area: 5,
  # Find High Terrain: 6, Explore Furthest: 7
```

### A.5.5 TRAINING HYPERPARAMETERS: OCEAN PLASTIC COLLECTION

```
behaviors:
  Agent:
    trainer_type: ppo
    hyperparameters:
      batch_size: 1024
      buffer_size: 10240
      learning_rate: 0.0003 # testing: 0.0
      beta: 0.005
      epsilon: 0.2
      lambd: 0.95
      num_epoch: 3
      learning_rate_schedule: linear # testing: constant
    network_settings:
      normalize: false
      hidden_units: 128
      num_layers: 2
      vis_encode_type: resnet
    reward_signals:
      extrinsic:
        gamma: 0.99
        strength: 0.9
        network_settings:
          vis_encode_type: resnet
      curiosity:
        gamma: 0.99
        strength: 0.1
        encoding_size: 256
        learning_rate: 0.0003 # testing: 0.0
        network_settings:
          vis_encode_type: resnet
    keep_checkpoints: 5
    max_steps: 3000000 # testing: 150000
    time_horizon: 10240
    summary_freq: 15000 # testing: 15000
    threaded: true

engine_settings:
  no_graphics: true

env_settings:
  env_path: /dev_environments/Hivex_OceanPlasticCollection_win
  seed: 5000 # testing: 6000

environment_parameters:
  task: [0, 1, 2, 3]
  # Main: 0, Find High Pollution Area: 1,
  # Group up: 2, Avoid Plastic: 3
```

### A.5.6 TRAINING HYPERPARAMETERS: AERIAL WILDFIRE SUPPRESSION

```
behaviors:
  Agent:
    trainer_type: ppo
    hyperparameters:
      batch_size: 256
      buffer_size: 4096
      learning_rate: 0.0003
      beta: 0.005
      epsilon: 0.2
      lambd: 0.95
      num_epoch: 3
      learning_rate_schedule: linear
    network_settings:
      normalize: false
      hidden_units: 256
      num_layers: 2
      vis_encode_type: simple
    reward_signals:
      extrinsic:
        gamma: 0.995
        strength: 1.0
    keep_checkpoints: 5
    max_steps: 1800000 # testing: 180000
    time_horizon: 4096
    summary_freq: 9000 # testing: 9000
    threaded: true

engine_settings:
  no_graphics: true

env_settings:
  env_path: /dev_environments/Hivex_AerialWildfireSuppression_win
  num_envs: 12
  seed: 5000 # testing: 6000

environment_parameters:
  terrain_level: [1, 2, 3, 4, 5, 6, 7, 8, 9, 10]
  task: [0, 1, 2, 3, 4, 5, 6, 7, 8]
  # Main Task: 0, Maximize Extinguishing Trees: 1,
  # Maximize Preparing Trees: 2, Minimze Time of Fire Burning: 3,
  # Protect Village: 4, Pick Up Water: 5, Drop Water: 6,
  # Find Fire: 7, Find Village: 8
```

## A.6    ADDITIONAL ENVIRONMENT FEATURES AND PROCESS DIAGRAMS

### A.6.1    WIND FARM CONTROL

Wind Farm Control Environment

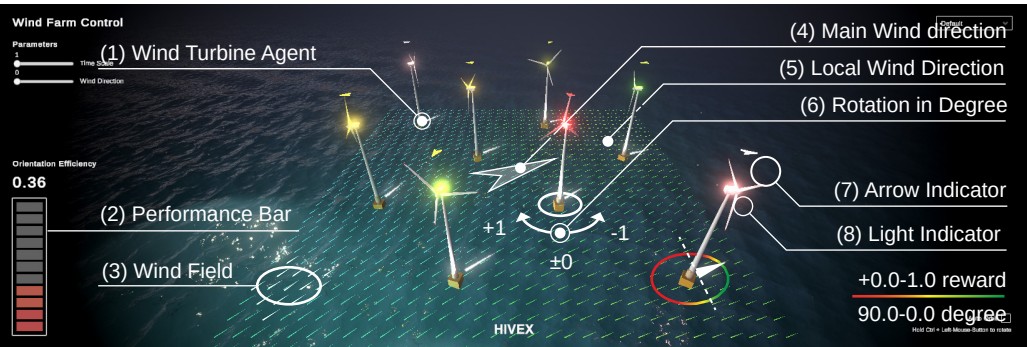

Environment Features

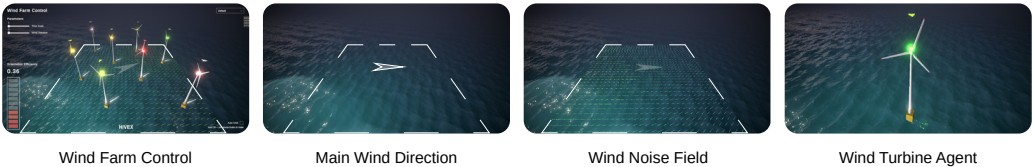

Wind Farm Control
Environment
      Main Wind Direction
          Wind Noise Field
             Wind Turbine Agent

Figure 11: Wind Farm Control - Main environment features: Main wind direction, wind noise field sample, agent controlled wind turbine.

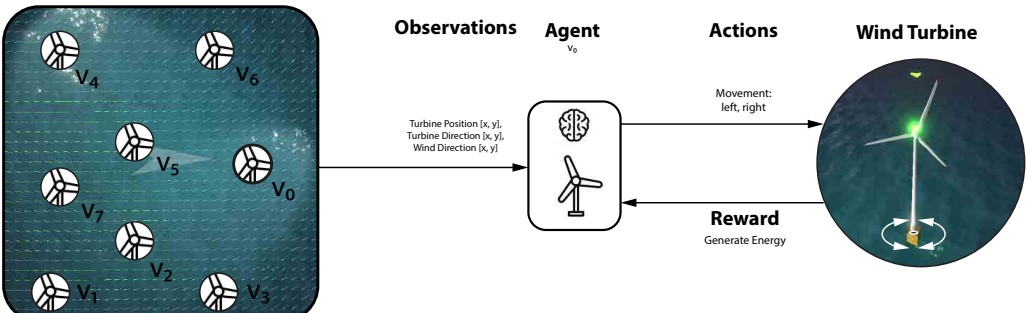

Figure 12: Wind Farm Control Process Diagram: The default layout of the WFC environment consists of eight wind turbines. Each turbine receives six vector inputs: its position (x, y), its orientation (x, y), and the local wind direction (x, y). The agent controlling each turbine has three discrete actions: do nothing, turn left, or turn right. The primary reward is based on the amount of wind energy generated when the turbine is optimally aligned with the wind direction.

### A.6.2 WILDFIRE RESOURCE MANAGEMENT

Wildfire Resource Management Environment

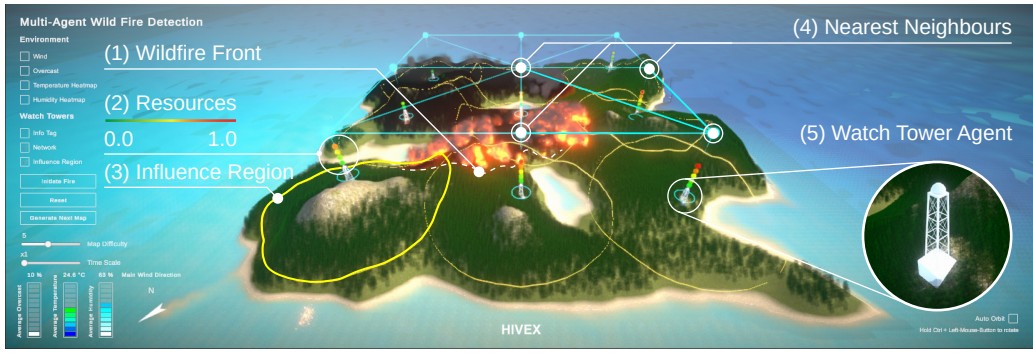

Environment Features

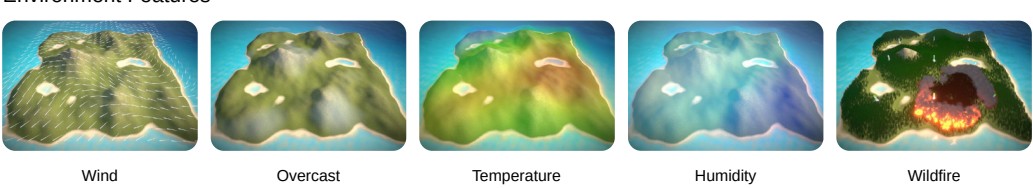

| Wind | Overcast | Temperature | Humidity | Wildfire |

Figure 13: Wildfire Resource Management - Main environment features: Wind field sample, overcast field sample, temperature field sample, humidity field sample, growing wildfire.

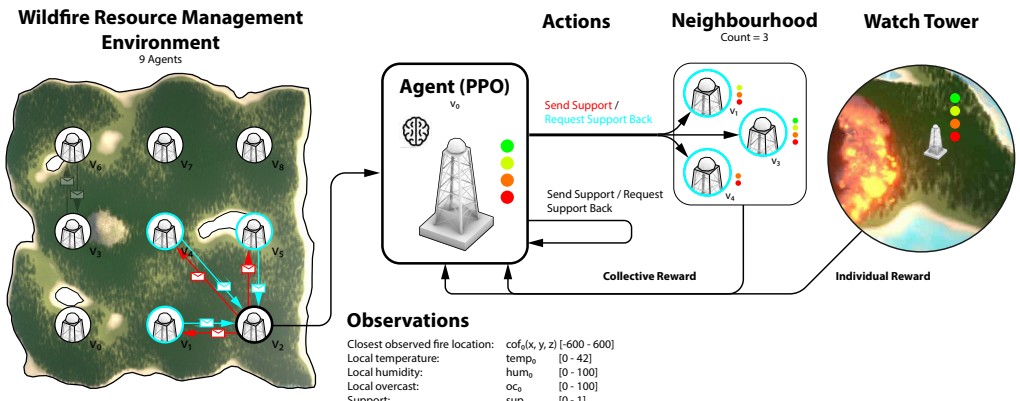

Figure 14: Wildfire Resource Management Process Diagram: The WRM environment consists of nine agents, each managing one of nine watchtowers. Each agent observes three environmental factors: temperature, humidity, and cloud cover, as well as whether a fire has been detected within 600 meters and the current resource level of its watchtower. Each watchtower starts with 1.0 resources, which can be allocated in 0.1 increments to either the agent's own tower or neighboring towers. Agents receive maximum rewards when their watchtower is well-resourced and a fire is approaching. For each step where the fire approaches and the watchtower is adequately prepared, the agent receives a high reward.

### A.6.3 DRONE-BASED REFORESTATION

Drone-Based Reforestation Environment

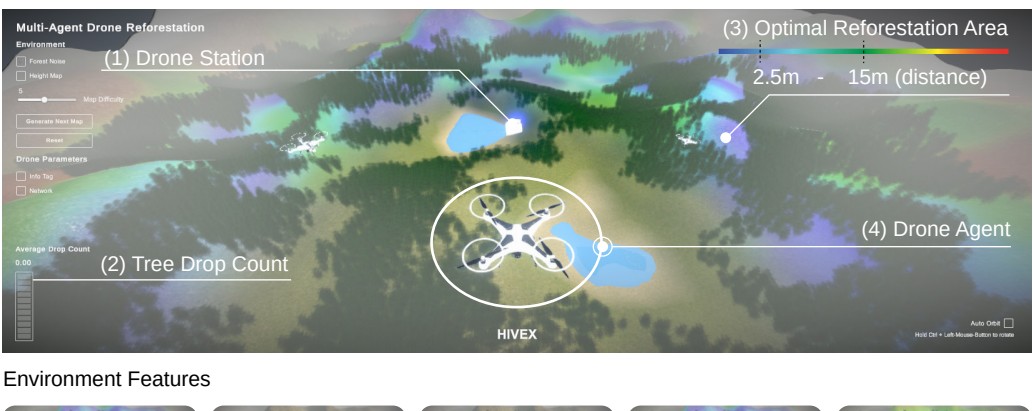

Environment Features

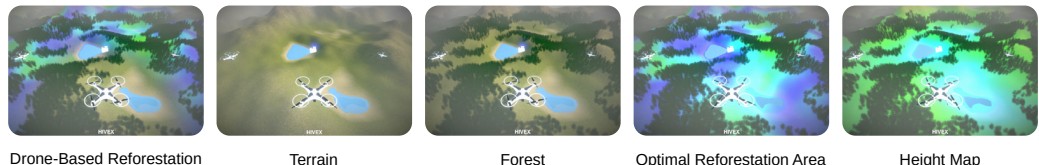

| Drone-Based Reforestation Environment | Terrain | Forest | Optimal Reforestation Area | Height Map |

Figure 15: Drone-Based Reforestation - Main environment features: Terrain sample, forest sample, non-visible to agent optimal reforestation area, non-visible to agent height map.

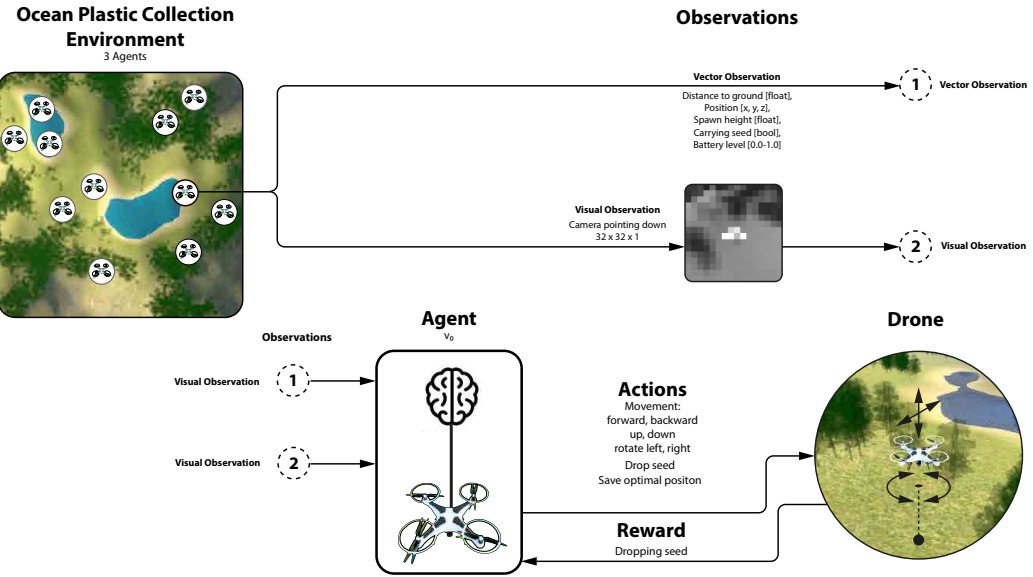

Figure 16: Drone-Based Reforestation Process Diagram: The default DBR environment features three agents, each controlling a drone. Each agent's observations include a vector with data such as the drone's distance to the ground, position (x, y, z), spawn height, whether it's carrying a seed, battery levels, and terrain, forest, and height maps. Additionally, agents receive a 32x32 grayscale visual observation. Agents can perform actions such as moving forward, backward, up, down, rotating left or right, saving optimal positions, and dropping a seed if carrying one. Rewards are given for successful seed drops, with bonuses for drops in highly fertile areas.

### A.6.4 OCEAN PLASTIC COLLECTION

Ocean Plastic Collection Environment

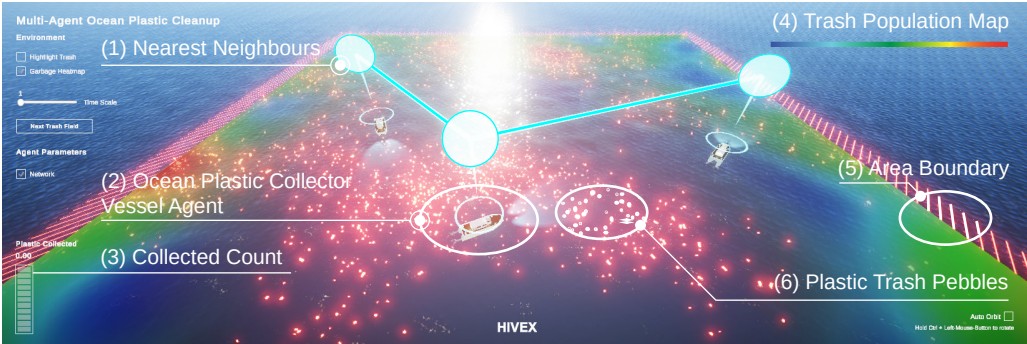

Environment Features

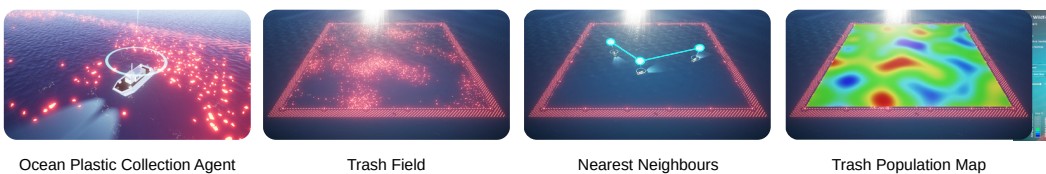

Figure 17: Ocean Plastic Collection - The main environment features an Agent-controlled ocean plastic collection vessel, trash field sample, nearest neighbours, and trash population map.

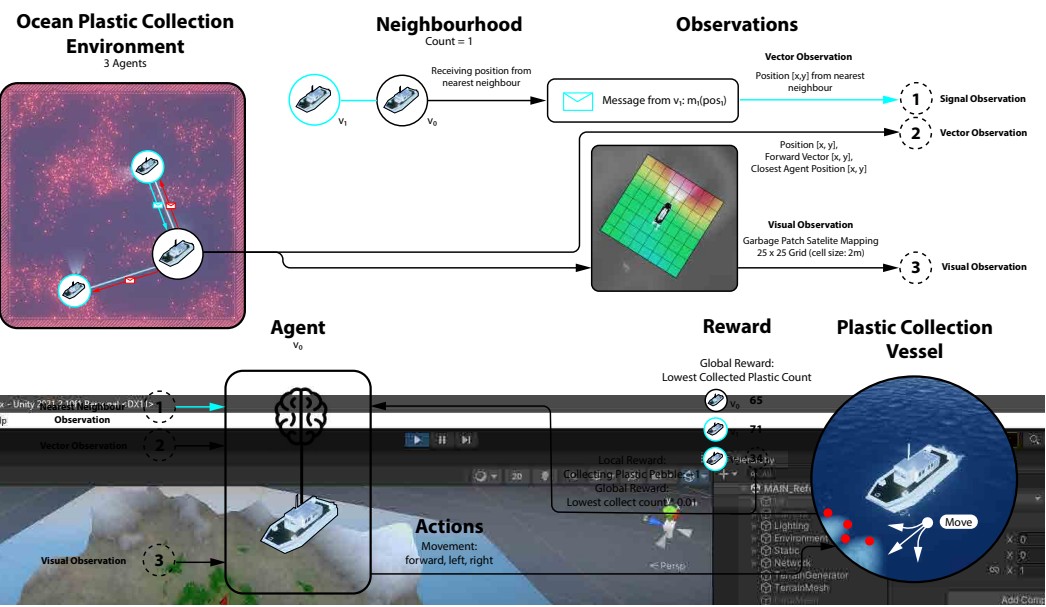

Figure 18: Ocean Plastic Collection Process Diagram: The default OPC environment includes three agents, each controlling a plastic collection vessel. Agents receive a 25x25 visual grid, where each cell represents 2 meters, along with vector observations such as their position (x, y), forward direction (x, y), and the position of the nearest agent (x, y). Agents can move forward, turn left, or turn right. Rewards are granted for each plastic pebble successfully collected from the ocean.

### A.6.5 AERIAL WILDFIRE SUPPRESSION

Aerial Wildfire Suppression Environment

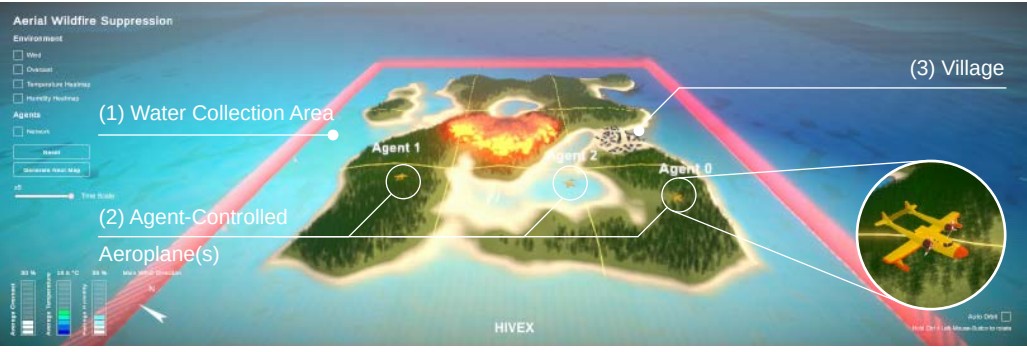

Environment Features

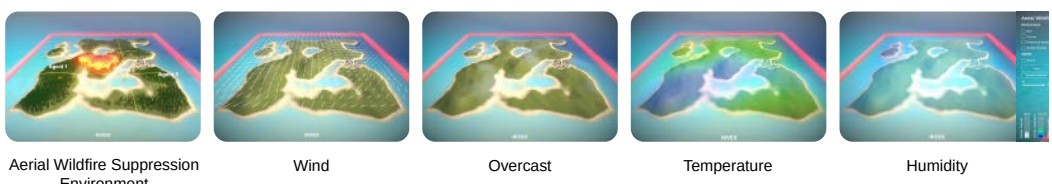

Figure 19: Aerial Wildfire Suppression Environment: (1) Water Collection Area, (2) Agent-controlled Wildfire Suppression Aeroplanes, (3) Village. Environment Features: Wind field sample, overcast field sample, temperature field sample, humidity field sample..

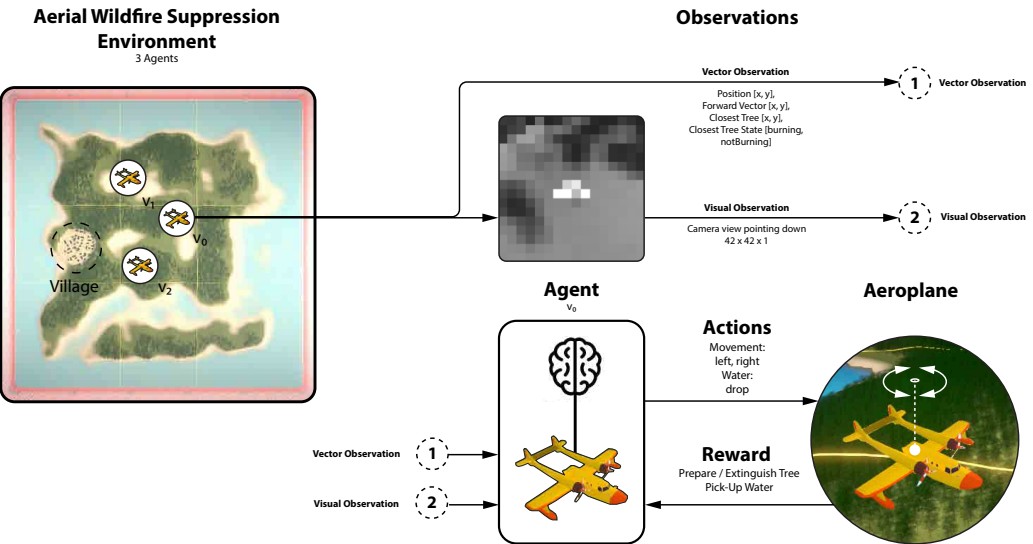

Figure 20: Aerial Wildfire Suppression Process Diagram: The default AWS environment consists of three agents, each controlling an airplane. Each agent receives both vector and visual observations. The vector observations include position (x, y), forward direction (x, y), the position of the nearest tree (x, y), and the tree's state: either [burning] or [not burning]. The visual observation is a 42x42 grayscale grid. Agents can steer left, steer right, or release water. Rewards are given for extinguishing burning trees, with smaller rewards for preparing non-burning but alive trees. A small reward is also granted for picking up water.

## A.7 ENVIRONMENT SCENARIO SAMPLES

### A.7.1 WIND FARM CONTROL

Figure 21: Wind Farm turbine layout patterns 0-7 [Default, Grid, Chain, Circle, Square, Cross, Two Rows, Field] and various seeds for the layout pattern 8 [Random].

## A.7.2 WILDFIRE RESOURCE MANAGEMENT

Figure 22: Wildfire Resource Management environment samples showing terrain elevation levels 1-10, top to bottom, and random seeds 0-7, left to right.

### A.7.3 Drone-Based Reforestation

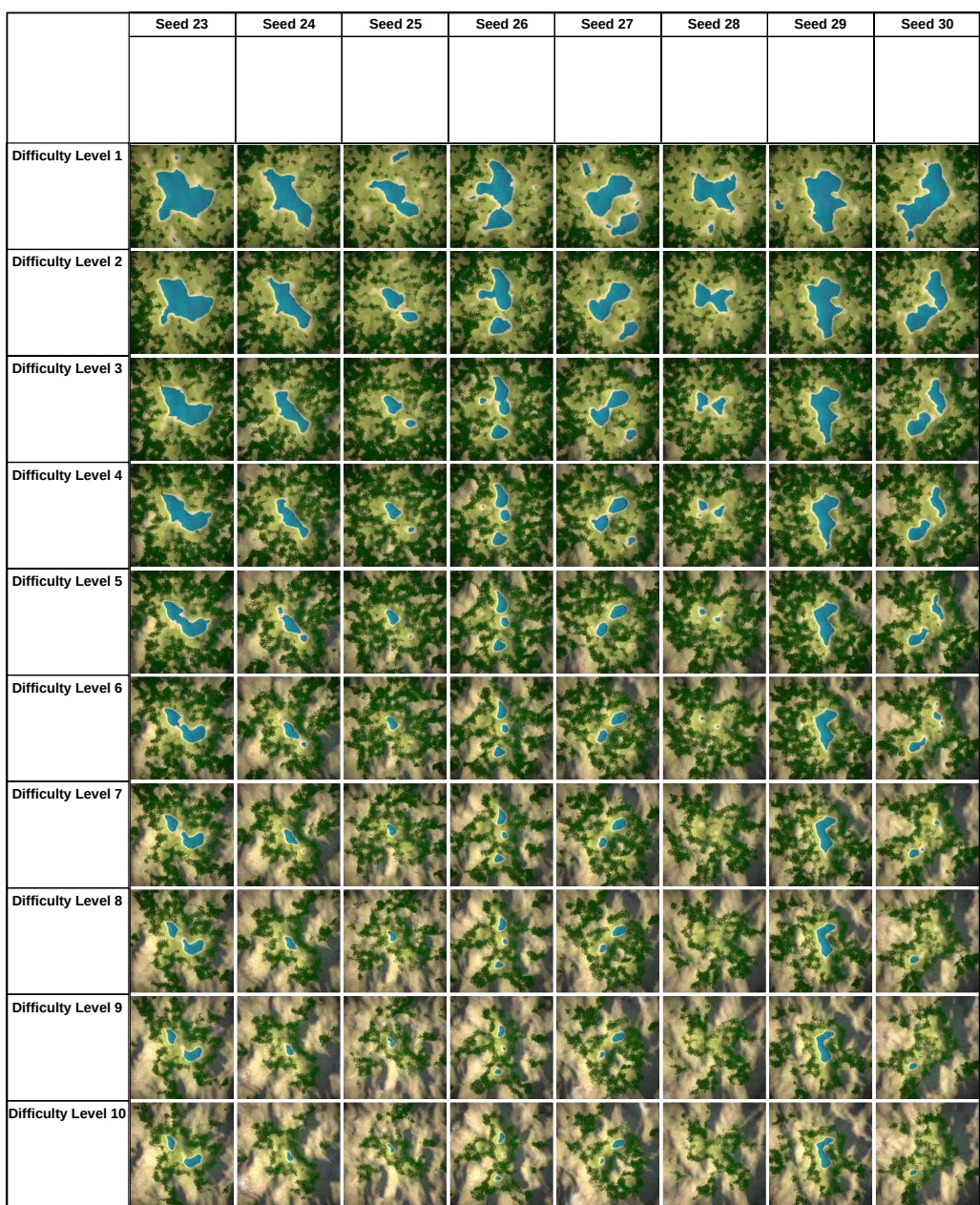

Figure 23: Drone-Based Reforestation environment samples showing terrain elevation levels 1-10, top to bottom, and random seeds 23-30, left to right.

A.7.4    OCEAN PLASTIC COLLECTION

Figure 24: Ocean Plastic Collection environment samples seeds 0-19 with pollution heatmap and spawn positions for agent-controlled vessels.

## A.7.5 AERIAL WILDFIRE SUPPRESSION

Figure 25: Aerial Wildfire Suppression environment samples showing terrain elevation levels 1-10 , top to bottom, and random seeds 0-7, left to right.

## A.8 REWARD DESCRIPTION AND CALCULATION

### A.8.1 WIND FARM CONTROL

**Reward Description**

1. **Generate Energy** - This is a positive reward given at each time-step, in the range $[0, 1]$. This reward corresponds to the performance of each wind turbine and is being calculated as described in equation 4. Orienting the wind turbine against the wind yields a high reward.
2. **Avoid Damage** - This is a positive reward given at each time-step, in the range $[0, 1]$. We remap the angle between the wind direction and the turbine's orientation linearly from $[0, 90]$ degrees to $[0, 1]$ reward and from $[90, 180]$ degrees to $[1, 0]$ reward. Orienting the wind turbine so that the rotor blades are parallel to the wind direction yields high reward.

**Reward Calculation**

**1. Generate Energy** - First, we need to describe how performance is calculated for each wind turbine:

Let us define:

- $a_{\text{turbine}} = 0.1$ — Acceleration of the turbine motor.
- $\theta$ — The angle between the wind turbine orientation and the wind direction at the turbine.
- $P(\theta) = 0.0$ — Performance, initialized as $0.0$, dependent on angle $\theta$.
- $W(\theta)$ — Wind force at wind turbine, dependent on angle $\theta$.
- $d$ — Wind turbine drag.

Calculation steps:

1. Calculation of wind force $W$ based on angle $\theta$:

$$W(\theta) = \begin{cases} 0 & \text{if } \theta < 0.5 \\ \text{Map}(\theta, 0.5, 1, 0, 1) & \text{if } 0.5 \le \theta \le 1 \end{cases} \tag{1}$$

The "Map" function linearly interpolates the value of force from 0 to 1 as angle increases from 0.5 to 1.

2. Calculation of drag:

$$d = -0.1 \times P(\theta) \tag{2}$$

3. Updating performance $P$ with drag and wind force:

$$P(\theta) = P(\theta) + d + W(\theta) \times a_{\text{turbine}} \tag{3}$$

4. Clamping performance $P(\theta)$ between 0 and 1 is the reward $R(\theta)$:

$$R(\theta) = \max(0, \min(1, P(\theta))) \tag{4}$$

Here, $\max(0, \min(1, P(\theta)))$ limits $P(\theta)$ within the interval $[0, 1]$, ensuring it neither falls below 0 nor exceeds 1.

**2. Avoid Damage** The avoid damage reward $R(\theta)$ can be calculated as follows:

Let us define:

- $\theta$ — The angle between the wind turbine orientation and the wind direction at the turbine.

Calculation steps:

1. Calculation of avoid damage reward based on angle $\theta$:

$$R(\theta) = \begin{cases} \frac{\theta}{90} & \text{if } 0 \le \theta \le 90 \\ 2 - \frac{\theta}{90} & \text{if } 90 < \theta \le 180 \end{cases} \tag{5}$$

### A.8.2 WILDFIRE RESOURCE MANAGEMENT

**Reward Description**

1. **Watch Tower Performance** - This is a positive reward given at each time step, corresponding to the performance of the agent-controlled watch tower only. This reward is weighted by the resources distributed by self to self. Equation 9 describes how the individual performance and reward are calculated.
2. **Neighbour Performance** - This is a positive reward given at each time step, corresponding to the sum of the performance of the neighbouring agent-controlled watch towers. This reward is weighted by the resources distributed by self to neighbouring watch towers. Equation 10 describes how the neighbour performance and reward are calculated. Agents receive additional rewards if they distribute useful resources to neighbouring watch towers.
3. **Collective Performance** - This is a positive reward given at each time step, corresponding to the sum of the performance of all agent-controlled watch towers. Equation 12 describes how the collective performance and reward are calculated.

**Reward Calculation**

**1. Watch Tower Performance** - First, we need to calculate the performance of each watch tower agent. Let us define:

- $d_{\text{thresh}} = 200$ — Threshold distance to a fire, used for normalization.
- $\vec{x}_0$ and $\vec{x}_1$ — 3D vector positions of the closest observed fire at timesteps 0 and 1, respectively.
- $C(\vec{x})$ — Function calculating the distance from the current watch tower to the closest observed fire at position $\vec{x}$.
- $d_0 = C(\vec{x}_0)$ and $d_1 = C(\vec{x}_1)$ — Distances to the closest fire at timesteps 0 and 1.
- $d_{1normalized}$ — Normalized distance at timestep 1: $d_{1normalized} = \frac{d_1}{d_{\text{thresh}}}$.
- $m$ — Indicates whether the fire is moving towards the tower: $m = (C(\vec{x}_1) < C(\vec{x}_0))$.
- $s = 270$ and $a = 5$ — Constants for the broken power law.

Calculation steps:

1. The remapped distance factor based on the direction of movement is given by:

$$d'_{1normalized} = \begin{cases} 0.5 - 0.5 \times d_{1normalized} & \text{if } m \\ 0.5 + 0.5 \times d_{1normalized} & \text{if not } m \end{cases} \tag{6}$$

2. The adjusted distance factor using the broken power law is:

$$d''_{1normalized} = \left(1 + \left(d'_{1normalized} \times \frac{1000}{s}\right)^a\right)^{-\frac{1}{2}} \tag{7}$$

3. This is the watch tower performance metric $p$:

$$P = d''_{1normalized} \tag{8}$$

Now, we can calculate the reward $R(p, r_{\text{distributed}}, r_{\text{supporting}})$ by defining:

- $r_{\text{distributed}}$ — Total supporting resources distributed from self and others.
- $r_{\text{supporting}}$ — The amount of supporting resources from self only.
- $P$ — Performance metric calculated as outlined above.

Calculation steps:

$$R(p, r_{\text{distributed}}, r_{\text{supporting}}) = P \times r_{\text{supporting}} \times r_{\text{supporting}} \tag{9}$$

**2. Neighbour Performance Reward**: We now describe how the neighbour reward is calculated.

Let us define the following:

- $p_i$ — Represents the performance metric for the $i$-th watch tower.
- $n$ — The number of neighbouring watch towers is 3.
- $R_{\text{neighbourhood}}$ — The neighbour reward across neighbouring watch towers $n \in N$.

The neighbour performance reward $R(n, p_i)$ calculation involves the following steps:

1. Sum over neighbouring watch towers individual performance:

$$R_{\text{neighbourhood}}(n, p_i) = \sum_{i=1}^{n} p_i \tag{10}$$

**3. Collective Performance Reward**: We now describe how the collective reward is calculated.

Let us define the following:

- $p_i$ — Represents the performance metric for the $i$-th watch tower.
- $n$ — The total number of watch towers.
- $R_{\text{collective}}$ — The collective reward across all watch towers $n \in N$.

The collective performance reward $R_{\text{collective}}$ calculation involves the following steps:

1. Compute the Mean Squared Error $\text{MSE}(n, p_i)$ of watch tower performances:

$$\text{MSE}(n, p_i) = \frac{1}{n} \sum_{i=1}^{n} p_i^2 \tag{11}$$

2. Calculate the collective reward:

$$R_{\text{collective}}(n, p_i) = 1 - \left| 1 - \sqrt{\text{MSE}(n, p_i)} \right| \tag{12}$$

A.8.3    OCEAN PLASTIC COLLECTION

**Reward Description**

1. **Collect Trash** - This is a positive reward of 1 given for each floating plastic pebble collected.
2. **Lowest Collected Trash Count** - This is a positive reward given at each time step for the lowest collected trash count amongst all agents. The lowest trash count is scaled by 0.01. The steps to calculate the lowest collected trash count reward can be found in Equation 15.
3. **Crossed Border** - This is a negative reward of $-100$ given when the border is crossed.
4. **Collided with Other Vessel** - This is a negative reward of $-100$ given when colliding with other vessel.
5. **Close to Other Vessel** - This is a positive reward of 1 given at each time step when the distance to the other vessel is smaller than or equal to 10. The steps to calculate the close to other vessel reward can be found in Equation 18.
6. **Nearby Trash Count Delta** - This is a positive reward given when the nearby trash field population is higher than it has been until this time step. The reward given is the delta between the previous nearby trash field population count and the current. A nearby trash field population count is calculated by finding all floating plastic pebbles around a vessel with a radius smaller than or equal to 25. The steps to calculate the nearby trash count delta reward can be found in Equation 21.
7. **Collide with Trash** - This is a negative reward of $-1$ given when the agent-controlled vessel is colliding with a floating plastic pebble.

**Reward Calculation**

**1. Collect Trash** - To calculate the Collect Trash reward, let us define the following:

- $r_t$ — Reward for each trash pebble collected.

Calculation steps:

1. Get the number of collected trash pebbles:

$$r_t = \sum_{i=1}^{N} \mathbb{I}(p_i \text{ is collected}) \tag{13}$$

**2. Lowest Collected Trash Count** - To calculate the lowest collected trash count reward, let us define the following:

- $a = 0.01$ — Lowest trash count factor.
- $T$ — Set of all agents lowest collected trash count.

Calculation steps:

1. Get the lowest trash count from all agents:

$$M(T) = \min(t_1, t_2, \ldots, t_n), \text{ where } t \in T \tag{14}$$

2. Calculate the lowest collected trash count reward $R(T)$:

$$R(T) = M(T) \times a \tag{15}$$

**3. Crossed Border** - To calculate the Crossed Border reward, let us define the following:

- $eh = 200$ — The environment half extend.
- $\vec{p}$ — The vessel position.
- $r_{cb}$ — Crossed boundary reward.

Calculation steps:

1. We can now calculate the Crossed Border reward:

$$r_{cb} = \begin{cases} -100 & \text{if } (p_x > eh \text{ or } p_x < -eh \text{ or } p_y > eh \text{ or } p_y < -eh) \\ 0 & \text{otherwise} \end{cases} \tag{16}$$

**4. Collided with Other Vessel** - To calculate the Collided with Other Vessel reward, let us define the following:

- $\vec{p}$ — The vessel position.
- $N_p$ — Neighbouring vessel positions.
- $r_c$ — Collision reward.

Calculation steps:

1. We can now calculate the Collided with Other Vessel reward:

$$r_c = \begin{cases} -100 & \text{if } \exists \vec{n} \in N_p \text{ such that } \vec{p} \text{ collides with } \vec{n} \\ 0 & \text{otherwise} \end{cases} \tag{17}$$

**5. Close to Other Vessel** - To calculate the lowest collected trash count reward, let us define the following:

- $d$ — Distance to closest neighbouring vessel.
- $d_{\text{thresh}}$ — Distance threshold to closest neighbouring vessel.

Calculation steps:

1. Calculate close to other vessel reward $r$.

$$r = \begin{cases} 10 & \text{if } d < d_{\text{thresh}} \\ 0 & \text{otherwise} \end{cases} \tag{18}$$

**6. Nearby Trash Count Delta** - To calculate the nearby trash count delta reward, let us define the following:

- $d_{\text{threshold}} = 25$ — Trash count nearby distance threshold.
- $P$ — All floating plastic pebble positions, $\{\vec{p_1}, \vec{p_2}, \ldots, \vec{p_n}\} \in P$.
- $ntc_{\text{old}} = 0$ — Old nearby trash count.
- $ntc_{\text{current}}$ — Current nearby trash count.
- $ntc_{\text{difference}}$ — Difference between the old and current nearby trash count.

Calculation steps:

1. The nearby trash count is calculated by considering only floating plastic pebbles with a distance below $d_{\text{threshold}}$:

$$ntc_{\text{current}} = \sum_{i=1}^{n} [\text{dist}(p_i) < d_{\text{threshold}}] \tag{19}$$

2. If the current nearby trash count $ntc_{\text{current}}$ is larger than the old nearby trash count $ntc_{\text{old}}$, the difference between the two is the reward $r(ntc_{\text{difference}})$:

$$ntc_{\text{difference}} = ntc_{\text{current}} - ntc_{\text{old}} \tag{20}$$

$$r(ntc_{\text{difference}}) = \max(0, ntc_{\text{difference}}) \tag{21}$$

3. Finally the old nearby trash count $ntc_{\text{old}}$ is updated with the current nearby trash count $ntc_{\text{current}}$:

$$ntc_{\text{old}} = ntc_{\text{current}} \tag{22}$$

**7. Collide with Trash** - To calculate the Collide with Trash reward, let us define the following:

- $\vec{p}$ — The vessel position.
- $P_t$ — All trash pebble positions.
- $r_p$ — Collision reward.

Calculation steps:

1. We can now calculate the Collide with Trash reward:

$$r_c = \begin{cases} -100 & \text{if } \exists \vec{n} \in P_t \text{ such that } \vec{p} \text{ collides with } \vec{p_t} \\ 0 & \text{otherwise} \end{cases} \tag{23}$$

### A.8.4 DRONE-BASED REFORESTATION

**Reward Description**

1. **Drop Seed** - This is a positive reward given at each seed drop. The drop seed reward consists of the quality of the drop location, a seed reward, in the range of $[0, 20]$ and distance to other seeds and existing trees, a distance reward, in the range of $[0, 10]$. Therefore, the resulting total drop seed reward is in the range of $[0, 30]$. The steps to calculate the total drop seed reward can be found in Equation 32.
2. **Deplete Energy Holding Seed** - This is a negative reward of $-1/(\text{episode length}/2)$ given at each time step if the drone is carrying a seed. The deplete energy reward at each time step is higher when carrying a seed than if not carrying a seed. The episode length is 2000.
3. **Deplete Energy No Seed** - This is a negative reward of $-1/(\text{episode length})$ given at each time step if the drone is not carrying a seed. The episode length is 2000.
4. **Pick-up Seed** - This is an optional positive reward given when a drone is returned to the drone station. There are two tasks in which this reward is given. In "Subtask: Pick-up Seed at Base" a reward of 100 is given and in "Subtask: Explore Furthest Distance and Return to Base" the reward is the furthest distance that has been explored and can be in the range of $[0, 200]$.
5. **Incremental Running Back** - After a seed has been dropped, this reward is given incrementally when flying back to the drone station. If the distance to the drone station at time-step $t_{-1}$ is larger than the current distance, this reward is given at incremental steps of $2.5$. The range of the incremental running back reward is $[0, 20]$, which can be modified by the running back multiplier, depending on the seed drop quality. If the Seed has been dropped 50 meters away from the drone station, an incremental running back reward can be received 20 times. The steps to calculate the incremental running back reward can be found in Equation 40.
6. **Group-up** - This is a positive reward of 10, given at each time-step, if the distance to any neighbouring drone is smaller than 5. The steps to calculate the group-up reward can be found in Equation 42.
7. **High Fertility Location Delta** - This is a reward given every time a higher fertility potential seed drop location has been found. The reward is the delta between the old and the new potential, if the new potential is higher than the old. The range of the reward is $[0, 1]$. The steps to calculate the high fertility location delta reward can be found in Equation 44.
8. **High Landscape Point Delta** - This is a reward given every time a higher point on the terrain landscape has been found. The reward is the delta between the old and the new height, if the new height is higher than the old. The reward range is $[0, 40]$, as 40 is the environment's height boundary. The steps to calculate the high landscape point delta reward can be found in Equation 46.

9. **Far Distance Explored Delta** - This is a reward given every time a further distance has been explored. The reward is the delta between the old distance and the new, if the new distance is further than the old. The reward range is $[0, 200]$, as 200 is the environment's half extend. The steps to calculate the far distance explored delta reward can be found in Equation 48.

10. **Find Close Tree** - This is a reward given when a tree has been found within a 20 meter radius. The reward given is 100. The steps to calculate the find close tree reward can be found in Equation 24.

**Reward Calculation**

**1. Drop Seed** - To calculate the drop seed reward, we need to calculate the actual seed drop reward and a distance reward. To calculate the seed drop reward, let us define the following:

- $dot_{max} = 75$ — Maximum distance to other trees.
- $dot_{min} = 2.5$ – Minimum distance to other trees.
- $dnt$ — Closest distance to new trees.
- $det$ — Closest distance to existing trees.
- $sdrm = 20$ — Seed drop reward multiplier.
- $r_s(det, dot_{min}, dot_{max})$ — Seed drop reward.

Calculation steps:

1. The following condition needs to hold true for this reward to be larger than 0. This ensures that the newly dropped seed is far enough from existing and seeds dropped in the past, but also that the seed is not too far away from the existing forest.

$$(dot_{min} \leq det \leq dot_{max}) \text{ and } (dnt \geq dot_{min}) \tag{24}$$

2. First, we remap the distance to existing and new trees to $[1, 0]$ so that a high reward is given when the seed is dropped close to existing or new trees.

$$r_s(det, dot_{min}, dot_{max}) = \text{Remap}(det, dot_{min}, dot_{max}, 1, 0) \tag{25}$$

3. Applying Multiplier:

$$r_s(det, dot_{min}, dot_{max}) = r_s(det, dot_{min}, dot_{max}) \times \text{sdrm} \tag{26}$$

We now describe how the distance reward is calculated. Let us define:

- $sdd$ — Seed drop distance to drone station.
- $ew = 200$ — Environment half extend.
- $drm = 10$ — Distance reward multiplier.
- $r_d(sdd_{normalized}, drm)$ — Distance reward.

Calculation steps:

1. The seed drop reward needs to be larger than 0 for the distance reward to be applied.

$$0 < r_s(det, dot_{min}, dot_{max}) \tag{27}$$

2. Calculate the distance reward using the normalized seed drop distance to the drone station.

$$sdd_{normalized} = sdd/ew \tag{28}$$

$$r_d(sdd_{normalized}, drm) = sdd_{normalized} \times drm \tag{29}$$

The total reward for dropping a seed consists of the drop seed reward 24 and the distance reward 27.

- $r_s$ — Seed drop reward, calculated as described above.
- $r_d$ — Distance reward, calculated as described above.
- $r_{sd}(r_s, r_d)$ — The total seed drop reward.

$$r_{sd}(r_s, r_d) = r_s + r_d \tag{30}$$

**2. Deplete Energy Holding Seed** - To calculate the deplete energy holding seed reward, let us define the following:

- episode length$_{\text{max}}$ = 2000 — Max episode length.
- $der_{\text{holding seed}}$(episode length$_{\text{max}}$) — Deplete energy reward while holding a seed.

$$der_{\text{holding seed}}(\text{episode length}_{\text{max}}) = -1/(\text{episode length}_{\text{max}}/2) \tag{31}$$

**3. Deplete Energy No Seed** - To calculate the deplete energy no seed reward, let us define the following:

- episode length$_{\text{max}}$ = 2000 — Max episode length.
- $der_{\text{no seed}}$(episode length$_{\text{max}}$) — Deplete energy reward without holding a seed.

$$der_{\text{no seed}}(\text{episode length}_{\text{max}}) = -1/(\text{episode length}_{\text{max}}) \tag{32}$$

**4. Pick-up Seed** - To calculate the Pick-up Seed reward, let us define the following:

- $p$ — Drone position.
- $d$ — Drone station position.
- $r_{ps}$ — Pick-up seed reward.

Calculation steps:

1. We can now calculate the Pick-up Seed reward:

$$r_{ps} = \begin{cases} 1 & \text{if distance}(p, d) = 0 \\ 0 & \text{otherwise} \end{cases} \tag{33}$$

**5. Incremental Running Back** - To calculate the incremental running back reward we need to calculate the seed drop reward 24 and distance reward 27. Let us define the following:

- $d_0$ — Current distance to drone station at time-step 0 in incremental steps.
- $\vec{p_0}$ — Current position at time-step 0.
- $\vec{dp}$ — Drone station position.
- $s = 2.5$ — Incremental step size towards drone station.
- $r_s$ — Seed drop reward, calculated as described above.
- $r_d$ — Distance reward, calculated as described above.
- $r_p = 20$ — Possible intermediate reward for running back to the drone station.
- $sdrm = 20$ — Seed drop reward multiplier.
- $drm = 10$ — Distance reward multiplier.
- $rbm$ — Running back multiplier.
- $r_{sd}(r_s, r_d)$ — The total seed drop reward.
- $r_{rb}$ — Reward for running back, given incrementally at step $s$ sized increments.
- $d_{\text{init}}$ — Initial distance to drone station, this is assigned when a seed has been dropped.
- $d_{\text{charge}} = 7.5$ — Distance to drone station to charge and pick-up seed.

Calculation steps:

1. The condition for the reward to be given is that the current distance from the drone to the drone station is smaller than in time-step $t_{-1}$. The current distance $d_0$ is calculated as follows:

$$d_0 = \left\lfloor \sqrt{\sum_{i=1}^{n}(p_{0i} - dp_i)^2}/s \right\rfloor \tag{34}$$

If $d_0 < d_{-1}$ continue with next step. $\tag{35}$

2. Let us first calculate the running back multiplier $rbm$ by normalizing the sum of seed drop and distance rewards.

$$rbm = (r_s + r_d)/(sdrm + drm) \tag{36}$$

3. We can now calculate the reward for running back to the drone station:

$$r_{rb} = (r_p \times rbm)/(d_{\text{init}} - d_{\text{charge}}/s) \tag{37}$$

4. Finally, we need to ensure that the reward $r_{rb}$ is equal to or above $0$ and equal to or below $r_p$:

$$r_{rb} = \begin{cases} 0 & \text{if } r_{rb} \leq 0 \\ r_p & \text{if } r_{rb} > r_p \\ r_{rb} & \text{otherwise} \end{cases} \tag{38}$$

**6. Group-up** - To calculate the group-up reward we need to define the following:

- $n_c$ — Closest neighbour.
- $d_{\text{thresh}} = 5$ — Distance threshold to closest drone.
- $\vec{p}$ — Current local drone position.
- $d_{cn}$ — Distance to closest neighbour
- $r_{gu}$ — Reward for grouping up.

Calculation steps:

1. Let us calculate the distance to the closest neighbour:

$$d_{cn} = \sqrt{\sum_{i=1}^{n} (p_i - n_{ci})^2} \tag{39}$$

2. We can now calculate the reward for grouping up:

$$r_{gu} = \begin{cases} 0 & \text{if } d_{\text{thresh}} \leq d_{cn} \\ 10 & \text{otherwise} \end{cases} \tag{40}$$

**7. High Fertility Location Delta** - To calculate the high fertility location delta reward, let us define the following:

- $dot_{\max} = 75$ — Maximum distance to other trees.
- $dot_{\min} = 2.5$ – Minimum distance to other trees.
- $dnt$ — Closest distance to new trees.
- $det$ — Closest distance to existing trees.
- $\vec{p}$ — Current local drone position.
- $d_{cet}$ — Distance to closest existing tree.
- $d_{cds}$ — Distance to closest dropped seed.
- $pot_{\text{old}} = 0$ — Old potential seed drop fertility, initialized as 0.
- $pot_{\text{current}}$ — Current potential seed drop fertility.
- $r_{fl}$ — High fertility location delta reward.

Calculation steps:

1. If $det$ is smaller or equal to $dot_{\max}$, $det$ is larger or equal to $dot_{\min}$ and $dnt$ is larger or equal to $dot_{\min}$, then follow the next calculation step, otherwise the reward $r_{fl}$ is 0.
2. Calculate the current potential:

$$pot_{\text{current}} = \text{Map}(det, dot_{\min}, dot_{\max}, 1, 0) \tag{41}$$

3. We can now calculate the high fertility location reward:

$$r_{fl} = \begin{cases} pot_{\text{current}} - pot_{\text{old}} & \text{if } pot_{\text{old}} < pot_{\text{current}}, \text{delta of current and old potential} \\ 0 & \text{otherwise} \end{cases} \tag{42}$$

**8. High Landscape Point Delta** - To calculate the high landscape point delta reward, let us define the following:

- $\vec{p}$ — Current local drone position.
- $h_{\text{old}} = 0$ — Old height, initialized as 0.
- $h_{\text{current}}$ — Current height.
- $h_{(\vec{x})}$ — Get height at position $\vec{x}$.
- $r_h$ — Height delta reward.

Calculation steps:

1. Calculate the current height:

$$h_{\text{current}} = h_{(\vec{p})} \tag{43}$$

2. We can now calculate the hight landscape point delta reward:

$$r_{fl} = \begin{cases} h_{\text{current}} - h_{\text{old}} & \text{if } h_{\text{old}} < h_{\text{current}}, \text{delta of current and old height} \\ 0 & \text{otherwise} \end{cases} \tag{44}$$

**9. Far Distance Explored Delta** - To calculate the far distance explored delta reward, let us define the following:

- $\vec{p}$ — Current local drone position.
- $d_{\text{old}} = 0$ — Old furthest distance to drone station, initialized as 0.
- $d_{\text{current}}$ — Current furthest distance to drone station.
- $d_{(\vec{x})}$ — Get distance to drone station at position $\vec{x}$.
- $r_{fd}$ — Far distance delta reward.

Calculation steps:

1. Calculate the current furthest distance:

$$d_{\text{current}} = \begin{cases} d_{(\vec{p})} & \text{if } d_{(\vec{p})} > d_{\text{old}} \\ d_{\text{old}} & \text{otherwise} \end{cases} \tag{45}$$

2. We can now calculate the far distance delta reward:

$$r_{fd} = \begin{cases} d_{\text{current}} - d_{\text{old}} & \text{if } d_{\text{old}} < d_{\text{current}}, \text{delta of current and old furthest distance} \\ 0 & \text{otherwise} \end{cases} \tag{46}$$

**10. Find Close Tree** - To calculate the find close tree reward, let us define the following:

- $\vec{p}$ — Current local drone position.
- $ew = 200$ — Environment half extend.
- $d_{cet}$ — Distance to closest existing tree.
- $cet(\vec{x})$ — Get closest existing tree given a location.
- $r_{ct}$ — Find close tree reward.

Calculation steps:

1. Let us calculate the distance to the closest existing tree and normalize using the environment half extend:

$$d_{cet} = cet(\vec{p})/ew \tag{47}$$

2. If $d_{cet} < 20$ a reward of 100 is given:

$$r_{ct} = \begin{cases} 100 & \text{if } d_{cet} \leq 20 \\ 0 & \text{otherwise} \end{cases} \tag{48}$$

### A.8.5 AERIAL WILDFIRE SUPPRESSION

**Reward Description**

1. **Crossed Border** - This is a negative reward of $-100$ given when the border of the environment is crossed. The border is a square around the island in the size of $1500$ by $1500$. The island is $1200$ by $1200$.
2. **Pick-up Water** - This is a positive reward of 1 given when the agent steers the aeroplane towards the water. The island is $1200$ by $1200$ and there is a girdle of water around the island with a width of $300$.
3. **Fire Out** - This is a positive reward of 10 given when the fire on the whole island dies out, with or without the active assistance of the agent.
4. **Too Close to Village** - This is a negative reward of $-50$ given when the fire is closer than $150$ to the centre of the village.
5. **Time Step Burning** - This is a negative reward of $-0.01$ given at each time-step, while the fire is burning.
6. **Find Fire** - This is a positive reward of 100 given when a burning tree has been found.

7. **Find Village** - This is a positive reward of $100$ given when the village has been found, and the distance between the current local aeroplane position and the village is less than $150$.
8. **Extinguishing Tree** - This is a positive reward in the range of $[0, 5]$ given for each tree that has been in the state burning in time-step $t_{-1}$ and is now extinguished by dropping water at its location.
9. **Preparing Tree** - This is a positive reward in the range of $[0, 1]$ given for each tree that has been in the state not burning in time-step $t_{-1}$ and is now wet by dropping water at its location.

### Reward Calculation

**1. Crossed Border** - To calculate the Crossed Border reward, let us define the following:

- $eh = 750$ — The environment half extend.
- $\vec{p}$ — The drone position.
- $r_{cb}$ — Crossed boundary reward.

Calculation steps:

1. We can now calculate the Crossed Border reward:

$$r_{cb} = \begin{cases} -100 & \text{if } (p_x > eh \text{ or } p_x < -eh \text{ or } p_y > eh \text{ or } p_y < -eh) \\ 0 & \text{otherwise} \end{cases} \tag{49}$$

**2. Pick-up Water** - To calculate the Pick-up Water reward, let us define the following:

- $eh = 750$ — The environment half extend.
- $ih = 600$ — Island half extend.
- $\vec{p}$ — The drone position.
- $r_{pw}$ — Pick-up Water reward.

Calculation steps:

1. We can now calculate the Pick-up Water reward:

$$r_{pw} = \begin{cases} 1 & \text{if } (p_x < eh \text{ or } p_x > -eh \text{ or } p_y < eh \text{ or } p_y > -eh) \\ & \text{and } (p_x > ih \text{ or } p_x < -ih \text{ or } p_y > ih \text{ or } p_y < -ih) \\ 0 & \text{otherwise} \end{cases} \tag{50}$$

**3. Fire Out** - To calculate the Fire Out reward, let us define the following:

- $T$ — All tree states.
- $r_{nb}$ — No burning tree reward.

Calculation steps:

1. We can now calculate the Fire Out reward:

$$r_{nb} = \begin{cases} 10 & \text{if } \forall t \in T, \, t \neq \text{"burning"} \\ 0 & \text{otherwise} \end{cases} \tag{51}$$

**4. Too Close to Village** - To calculate the Too Close to Village reward, let us define the following:

- $T_c$ — All tree states, closer to or equal to $150$ to the village.
- $r_{cv}$ — Too Close to Village reward.

Calculation steps:

1. We can now calculate the Fire Out reward:

$$r_{cc} = \begin{cases} -50 & \text{if } \exists t \in T_c, \, t = \text{"burning"} \\ 0 & \text{otherwise} \end{cases} \tag{52}$$

**5. Time Step Burning** - To calculate the Time Step Burning reward, let us define the following:

- $T$ — All tree states.

- $r_{tsb}$ — Time Step Burning reward.

Calculation steps:

1. We can now calculate the Time Step Burning reward:

$$r_{tsb} = \begin{cases} -0.01 & \text{if } \forall t \in T,\ t = \text{"burning"} \\ 0 & \text{otherwise} \end{cases} \tag{53}$$

**6. Find Fire** - To calculate the Find Fire reward, let us define the following:

- $\vec{p}$ — The drone position.
- $d_t = 150$ — Distance threshold.
- $T$ — All tree states.
- $r_f$ — Find Fire reward.

Calculation steps:

1. We can now calculate the Find Fire reward:

$$r_f = \begin{cases} 100 & \text{if } \exists t \in T \text{ such that distance}(p) < d_t \text{ meters and } t = \text{"burning"} \\ 0 & \text{otherwise} \end{cases} \tag{54}$$

**7. Find Village** - To calculate the Find Village reward, let us define the following:

- $\vec{p}$ — The drone position.
- $d_t = 150$ — Distance threshold.
- $r_v$ — Find Village reward.

Calculation steps:

1. We can now calculate the Find Village reward:

$$r_v = \begin{cases} 100 & \text{if distance}(\vec{p}) \leq d_t \text{ meters} \\ 0 & \text{otherwise} \end{cases} \tag{55}$$

**8. Extinguishing Tree** - To calculate the Extinguish Tree reward, let us define the following:

- $T$ — All tree states.
- $r_e$ — Extinguish Tree reward.

Calculation steps:

1. We can now calculate the Extinguish Tree reward:

$$r_e = 5 \sum_{t \in T} \mathbb{I}(t_{\text{previous}} = \text{"burning" and } t_{\text{current}} = \text{"extinguished"}) \tag{56}$$

**9. Preparing Tree** - To calculate the Preparing Tree reward, let us define the following:

- $T$ — All tree states.
- $r_p$ — Preparing Tree reward.

Calculation steps:

1. We can now calculate the Preparing Tree reward:

$$r_e = \sum_{t \in T} \mathbb{I}(t_{\text{previous}} = \text{"not Burning" and } t_{\text{current}} = \text{"wet"}) \tag{57}$$

## A.9 TASK DESCRIPTION AND REWARD SCALE

### A.9.1 WIND FARM CONTROL

**Task Description**

1. **Main Task: Generate Energy** - This is the main task of the environment. The agent's goal is to rotate the wind turbine to be oriented against the wind direction and hence generate energy.

2. **Subtask: Avoid Damage** - This is a subtask to turn the wind turbine 90 degrees away so that the wind turbine rotor blades are parallel to the wind direction, avoiding damage to the wind turbine's rotor blades.

**Reward Scale**

Table 7: Main- and Sub-Task Reward Scale

| | Task | |
| --- | --- | --- |
| Reward | 1. | 2. |
| 1. Generate Energy | 1 | 0 |
| 2. Avoid Damage | 0 | 1 |

### A.9.2 WILDFIRE RESOURCE MANAGEMENT

**Task Descriptions**

1. **Main Task: Distribute Resources** - This is the main task of the environment. The goal of the agent is to distribute a total of 1.0 resources at each time step to self or neighbouring watch towers. If the agent is out of resources, it has to remove resources from self or neighbouring watch towers before re-distribution. The resources should be distributed to the watch towers where the fire is closest and incoming.
2. **Subtask: Keep All** - This is a subtask with the same goal as the main task, however distributing resources to self yields higher rewards than distributing them to neighbouring watch towers.
3. **Subtask: Distribute All** - This is a subtask with the same goal as the main task, however distributing resources to neighbouring watch towers yields higher rewards than distributing them to self.

**Reward Scale**

Table 8: Main- and Sub-Task Reward Scale

| | Task | | |
| --- | --- | --- | --- |
| Reward | 1. | 2. | 3. |
| 1. Watch Tower Performance | 1 | 10 | 1 |
| 2. Neighbourhood Performance | 1 | 1 | 10 |
| 2. Collective Performance | 1 | 1 | 1 |

### A.9.3 OCEAN PLASTIC COLLECTION

**Task Description**

1. **Main Task: Plastic Collection** - This is the main task of the environment. The goal for the agent is to accelerate and steer the plastic collection vessel to collect as many floating plastic pebbles as possible while avoiding crashing into other vessels and crossing the environments border.
2. **Subtask: Find Highest Polluted Area** - This is a subtask with the goal of finding the highest trash population area in a given scenario.
3. **Subtask: Group Up** - This is a subtask with the goal of finding other vessels and staying close to other vessels while collecting as many floating plastic pebbles as possible.
4. **Subtask: Avoid Plastic** - This is a subtask with the goal of avoiding floating plastic pebbles.

**Reward Scale**

Table 9: Main- and Sub-Task Reward Scale

| Reward | Task | | | |
| --- | --- | --- | --- | --- |
| | 1. | 2. | 3. | 4. |
| 1. Collect Trash | 1 | 1 | 1 | -1 |
| 2. Global Lowest Trash Collected | 1 | 1 | 1 | 0 |
| 3. Crossed Border | 1 | 1 | 1 | 1 |
| 4. Collided with Other Vessel | 1 | 1 | 1 | 1 |
| 5. Close to Other Vessel | 0 | 0 | 1 | 0 |
| 6. Nearby Trash Count Delta | 0 | 1 | 0 | 0 |
| 7. Collide with Trash | 0 | 0 | 0 | 1 |

### A.9.4 DRONE-BASED REFORESTATION

**Task Description**

1. **Main Task: Maximize Collective Planted Tree Count** - This is the main task of the environment. The goal for the agent is to pick up a seed and re-charge batteries at the drone station, explore to find fertile ground for the seed, that is, a location that is close to existing trees, and drop the seed while maintaining enough battery charge to return to the drone station.
2. **Subtask: Find Closest Forest Perimeter** - This is a subtask with the goal of finding the closest forest perimeter.
3. **Subtask: Pick-up Seed at Base** - This is a subtask with the goal of going back to the drone station, picking up a seed, and recharging the battery. In this subtask, the initial position of drones is random instead of at the drone station.
4. **Subtask: Drop Seed** - This is a subtask with the goal of finding the most fertile soil and dropping a seed.
5. **Subtask: Find Highest Potential Seed Drop Location** - This is a subtask with the goal of finding soil with the highest fertility.
6. **Subtask: Find Highest Point on Landscape** - This is a subtask with the goal of finding the highest point on the landscape.
7. **Subtask: Explore Furthest Distance and Return to Base** - This is a subtask with the goal of exploring the furthest from the drone station and returning.

**Reward Scale**

Table 10: Main- and Sub-Task Reward Scale

| Reward | Task | | | | | | | |
|---|---|---|---|---|---|---|---|---|
| | 1. | 2. | 3. | 4. | 5. | 6. | 7. | 8. |
| 1. Drop Seed | 1 | 0 | 0 | 0 | 1 | 0 | 0 | 0 |
| 2. Deplete Energy Holding Seed | 1 | 1 | 1 | 1 | 1 | 1 | 1 | 1 |
| 3. Deplete Energy No Seed | 1 | 1 | 1 | 1 | 1 | 1 | 1 | 1 |
| 4. Pick-up Seed | 1 | 0 | 100 | 1 | 1 | 0 | 0 | 0-200 |
| 5. Incremental Running Back | 1 | 0 | 0 | 1 | 1 | 0 | 0 | 1 |
| 6. High Fertility Location Delta | 0 | 0 | 0 | 0 | 0 | 1 | 0 | 0 |
| 7. High Landscape Point Delta | 0 | 0 | 0 | 0 | 0 | 0 | 1 | 0 |
| 8. Far Distance Explored Delta | 0 | 0 | 0 | 0 | 0 | 0 | 0 | 1 |
| 9. Find Close Tree | 0 | 1 | 0 | 0 | 0 | 0 | 0 | 0 |

### A.9.5 AERIAL WILDFIRE SUPPRESSION

**Task Description**

1. **Main Task: Minimize Time Fire Burning and Prevent Fire From Moving Towards Village** - This is the main task of the environment. The goal for the agent is to pick up water and extinguish as many burning trees as possible or prepare a forest that is not yet burning. A secondary goal is to protect the village from approaching fire by extinguishing burning trees before they get too close to the village or redirecting the fire by preparing trees.
2. **Subtask: Maximize Extinguished Burning Trees** - This is a subtask with the goal of extinguishing as many burning trees as possible.
3. **Subtask: Maximize Preparing Non-Burning Trees** - This is a subtask with the goal of preparing as many non-burning trees as possible.
4. **Subtask: Minimize Time Fire Burning** - This is a subtask with the goal of minimizing the time of trees burning.

5. **Subtask: Protect Village** - This is a subtask with the goal of protecting the village from approaching fire.
6. **Subtask: Pick Up water** - This is a subtask with the goal of picking up water.
7. **Subtask: Drop Water** - This is a subtask with the goal of dropping water anywhere.
8. **Subtask: Find Fire** - This is a subtask with the goal of finding a burning tree.
9. **Subtask: Find Village** - This is a subtask with the goal of finding the village.

**Reward Scale**

Table 11: Main- and Sub-Task Reward Scale

| Reward | Task | | | | | | | | |
|---|---|---|---|---|---|---|---|---|---|
| | 1. | 2. | 3. | 4. | 5. | 6. | 7. | 8. | 9. |
| 1. Crossed Border | 1 | 1 | 1 | 1 | 1 | 1 | 1 | 1 | 1 |
| 2. Pick-up Water | 1 | 1 | 1 | 1 | 1 | 100 | 1 | 0 | 0 |
| 3. Fire Out | 1 | 1 | 1 | 1 | 1 | 0 | 0 | 0 | 0 |
| 4. Too Close to Village | 1 | 1 | 1 | 1 | 10 | 0 | 0 | 0 | 0 |
| 5. Time Step Burning | 0 | 0 | 0 | 1 | 0 | 0 | 0 | 0 | 0 |
| 6. Find Fire | 0 | 0 | 0 | 0 | 0 | 0 | 0 | 1 | 0 |
| 7. Find Village Drop Water | 0 | 0 | 0 | 0 | 0 | 0 | 0 | 0 | 1 |
| 8. Extinguishing Tree | 1 | 10 | 1 | 1 | 1 | 1 | 1 | 0 | 0 |
| 9. Preparing Tree | 1 | 1 | 5 | 1 | 1 | 1 | 1 | 0 | 0 |

## A.10 ADDITIONAL RESULTS

### A.10.1 WIND FARM CONTROL: TRAIN & TEST METRICS

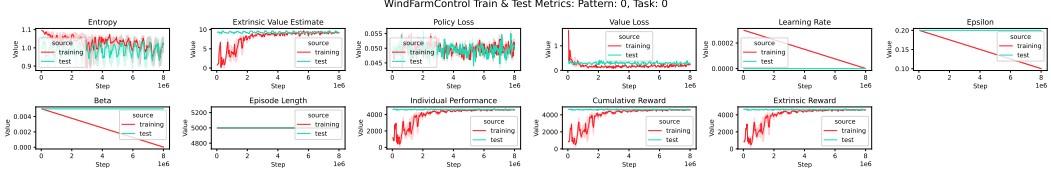

Figure 26: Wind Farm Control: Train & Test Metrics: Pattern 0, Task 0.

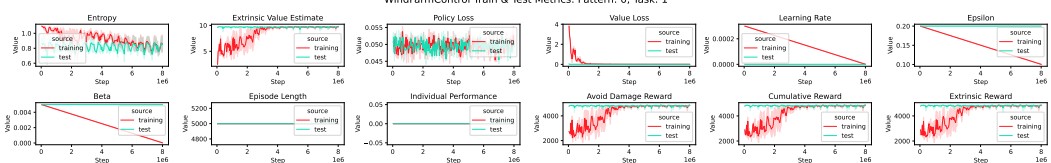

Figure 27: Wind Farm Control: Train & Test Metrics: Pattern 0, Task 1.

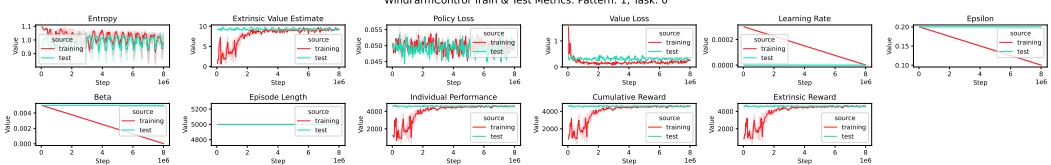

Figure 28: Wind Farm Control: Train & Test Metrics: Pattern 1, Task 0.

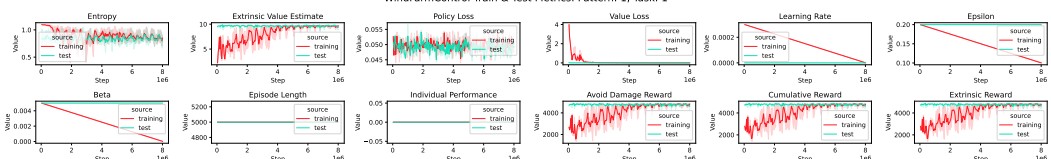

Figure 29: Wind Farm Control: Train & Test Metrics: Pattern 1, Task 1.

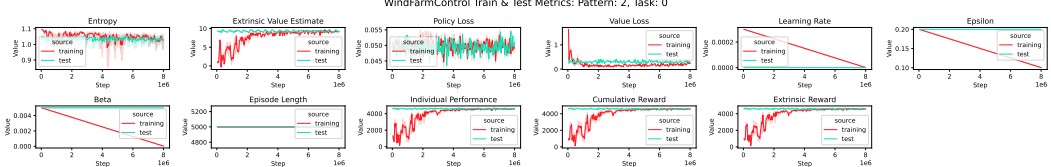

Figure 30: Wind Farm Control: Train & Test Metrics: Pattern 2, Task 0.

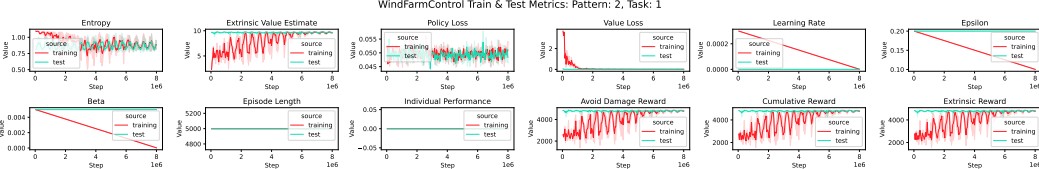

Figure 31: Wind Farm Control: Train & Test Metrics: Pattern 2, Task 1.

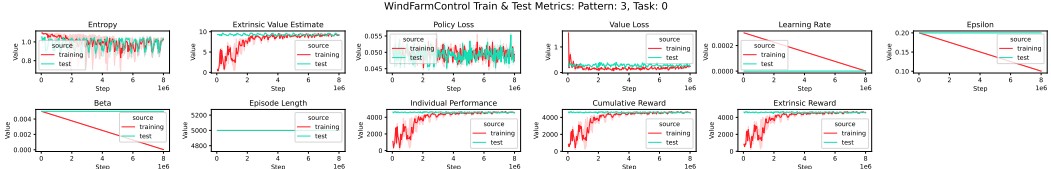

Figure 32: Wind Farm Control: Train & Test Metrics: Pattern 3, Task 0.

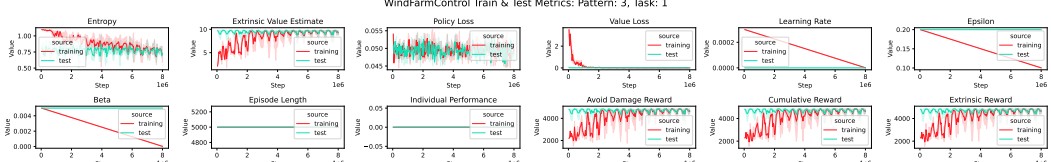

Figure 33: Wind Farm Control: Train & Test Metrics: Pattern 3, Task 1.

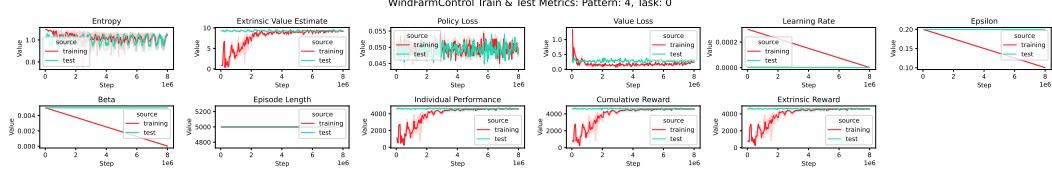

Figure 34: Wind Farm Control: Train & Test Metrics: Pattern 4, Task 0.

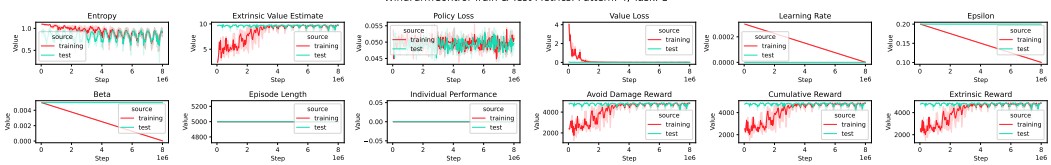

Figure 35: Wind Farm Control: Train & Test Metrics: Pattern 4, Task 1.

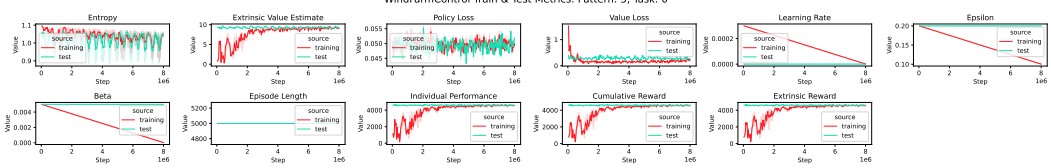

Figure 36: Wind Farm Control: Train & Test Metrics: Pattern 5, Task 0.

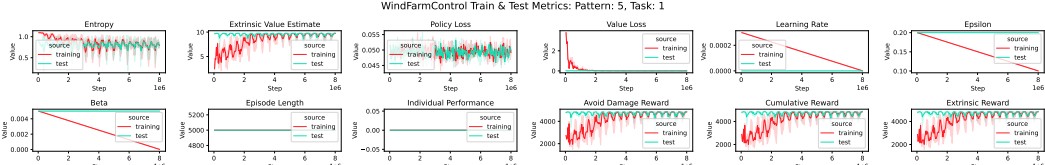

Figure 37: Wind Farm Control: Train & Test Metrics: Pattern 5, Task 1.

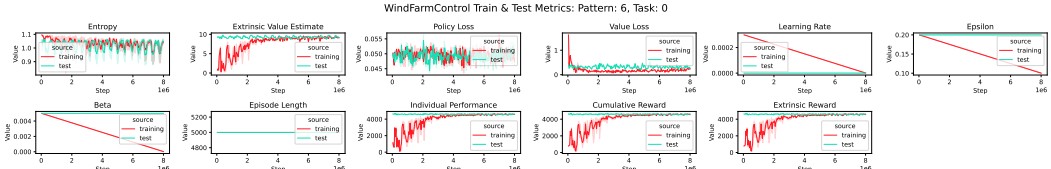

Figure 38: Wind Farm Control: Train & Test Metrics: Pattern 6, Task 0.

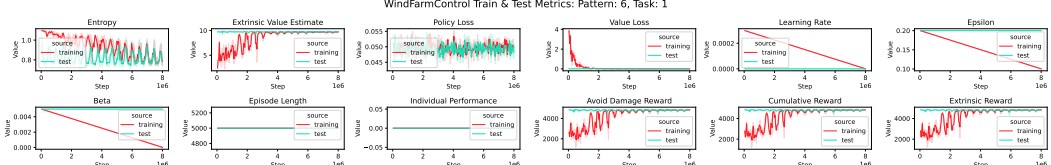

Figure 39: Wind Farm Control: Train & Test Metrics: Pattern 6, Task 1.

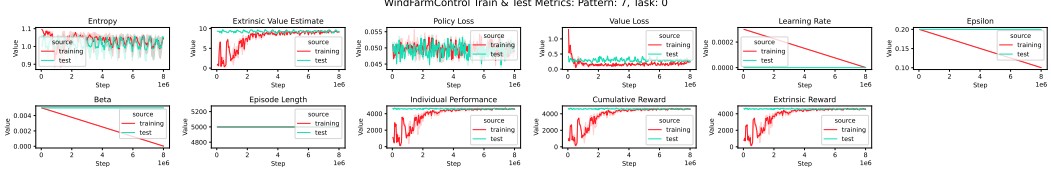

Figure 40: Wind Farm Control: Train & Test Metrics: Pattern 7, Task 0.

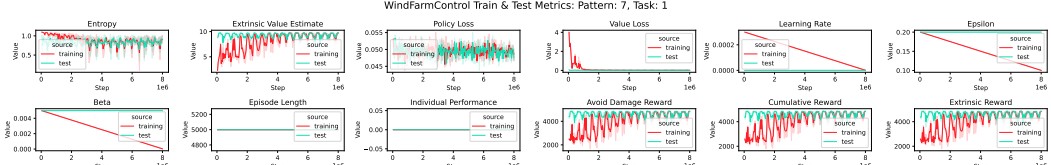

Figure 41: Wind Farm Control: Train & Test Metrics: Pattern 7, Task 1.

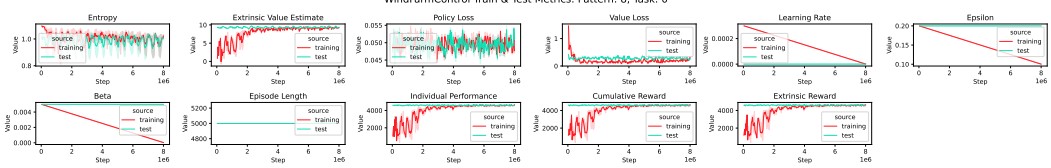

Figure 42: Wind Farm Control: Train & Test Metrics: Pattern 8, Task 0.

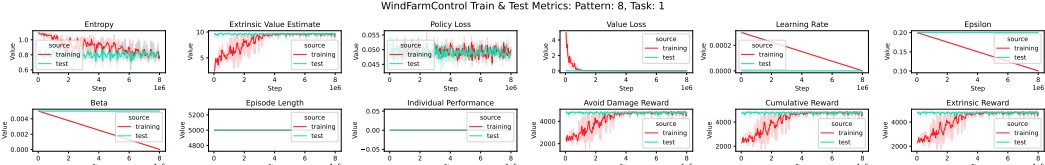

Figure 43: Wind Farm Control: Train & Test Metrics: Pattern 8, Task 1.

### A.10.2 WIND FARM CONTROL: AVERAGE TEST METRIC - TASK VS PATTERN

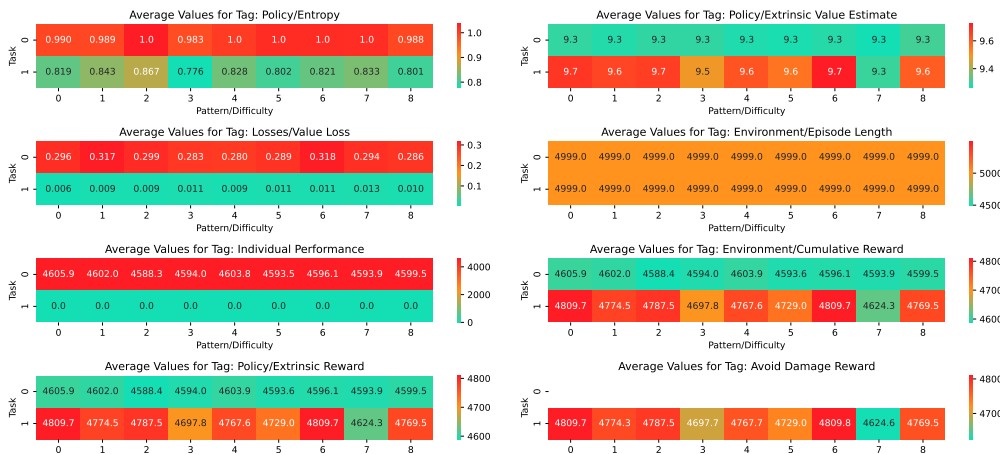

Figure 44: Wind Farm Control: Average Train & Test Metrics.

### A.10.3 WILDFIRE RESOURCE MANAGEMENT: TRAIN & TEST METRICS

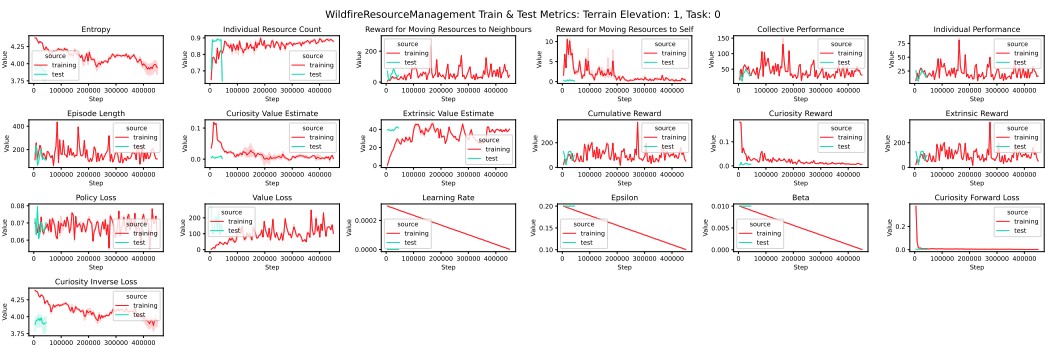

Figure 45: Wildfire Resource Management: Train & Test Metrics: Terrain Elevation 1, Task 0.

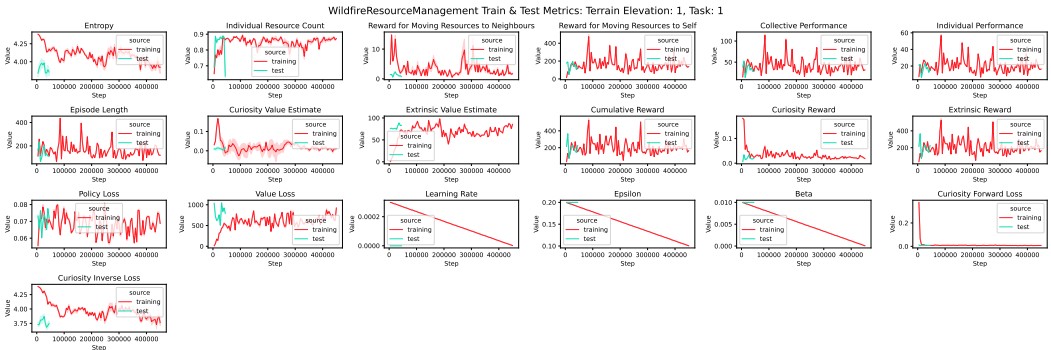

Figure 46: Wildfire Resource Management: Train & Test Metrics: Terrain Elevation 1, Task 1.

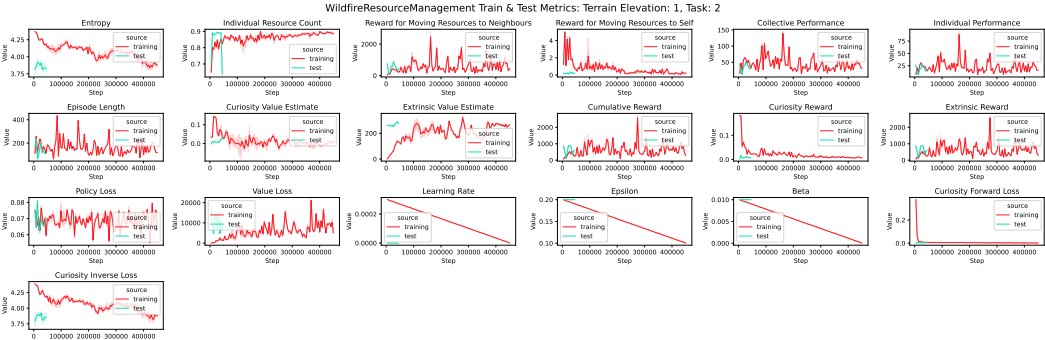

Figure 47: Wildfire Resource Management: Train & Test Metrics: Terrain Elevation 1, Task 2.

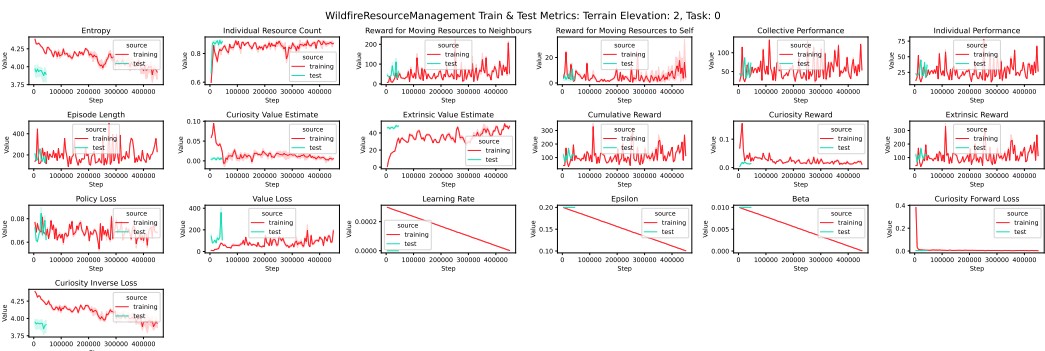

Figure 48: Wildfire Resource Management: Train & Test Metrics: Terrain Elevation 2, Task 0.

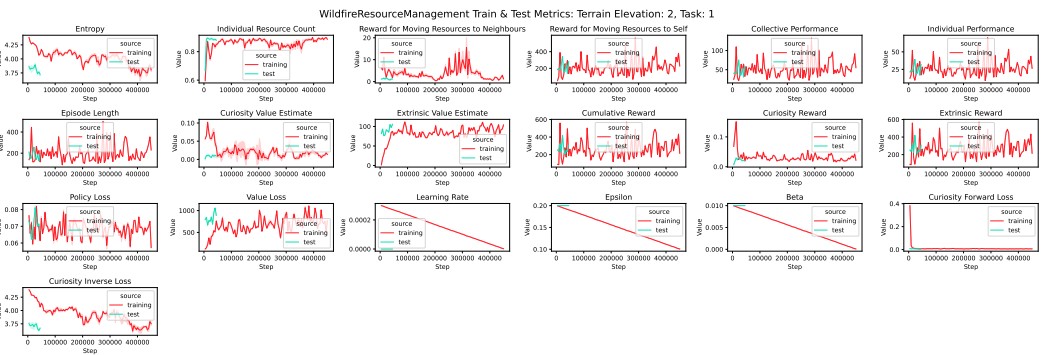

Figure 49: Wildfire Resource Management: Train & Test Metrics: Terrain Elevation 2, Task 1.

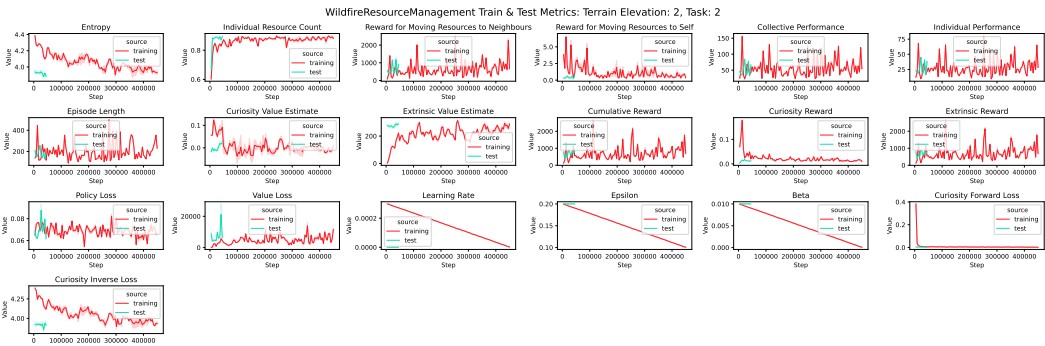

Figure 50: Wildfire Resource Management: Train & Test Metrics: Terrain Elevation 2, Task 2.

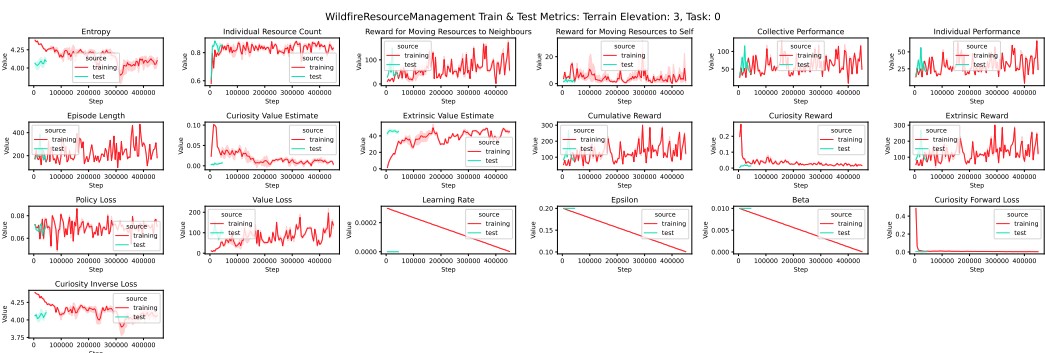

Figure 51: Wildfire Resource Management: Train & Test Metrics: Terrain Elevation 3, Task 0.

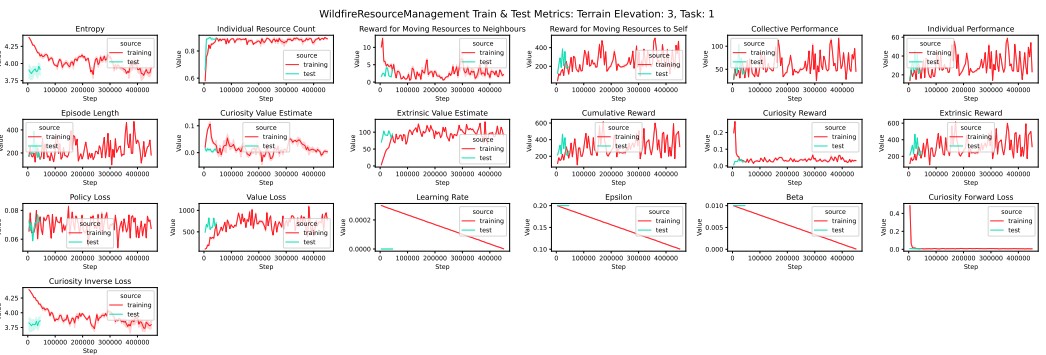

Figure 52: Wildfire Resource Management: Train & Test Metrics: Terrain Elevation 3, Task 1.

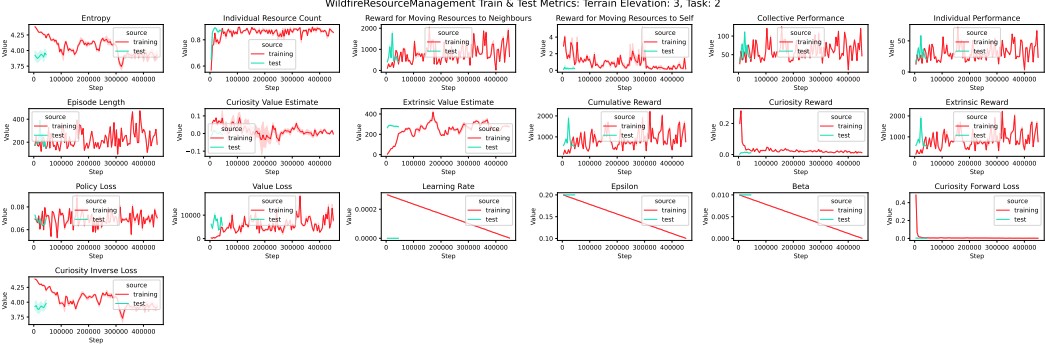

Figure 53: Wildfire Resource Management: Train & Test Metrics: Terrain Elevation 3, Task 2.

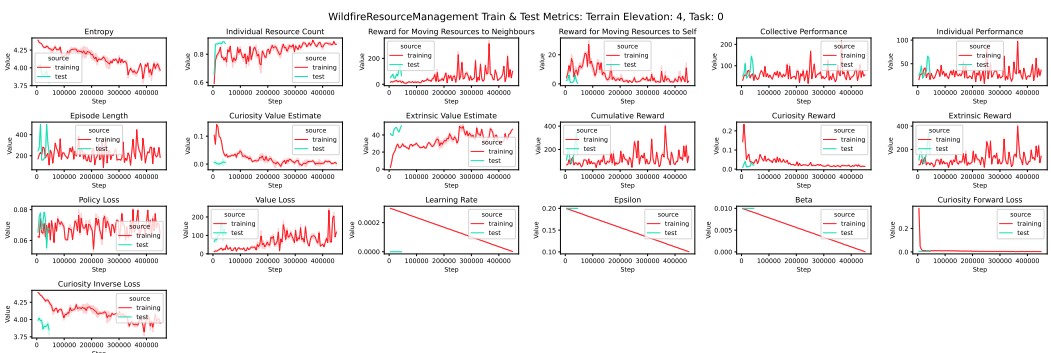

Figure 54: Wildfire Resource Management: Train & Test Metrics: Terrain Elevation 4, Task 0.

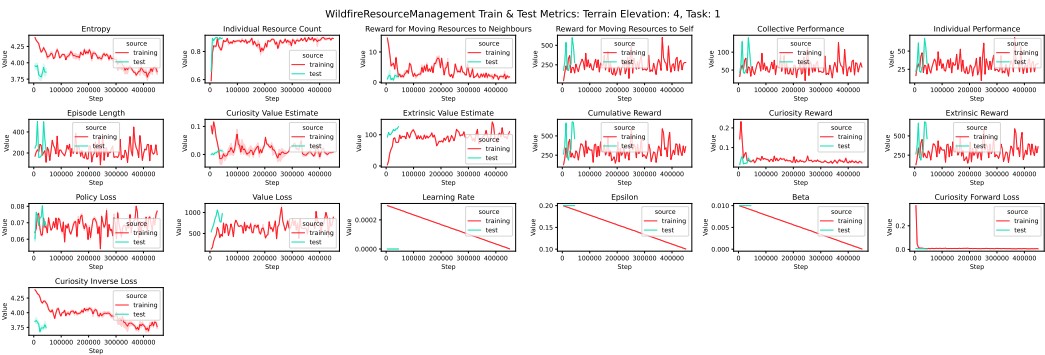

Figure 55: Wildfire Resource Management: Train & Test Metrics: Terrain Elevation 4, Task 1.

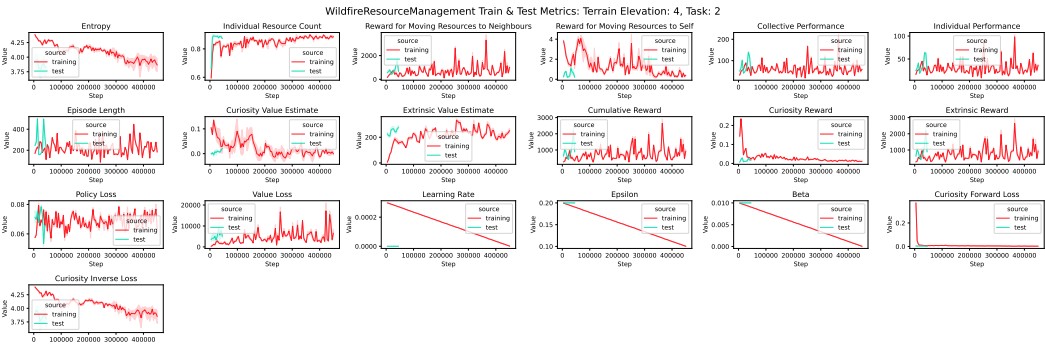

Figure 56: Wildfire Resource Management: Train & Test Metrics: Terrain Elevation 4, Task 2.

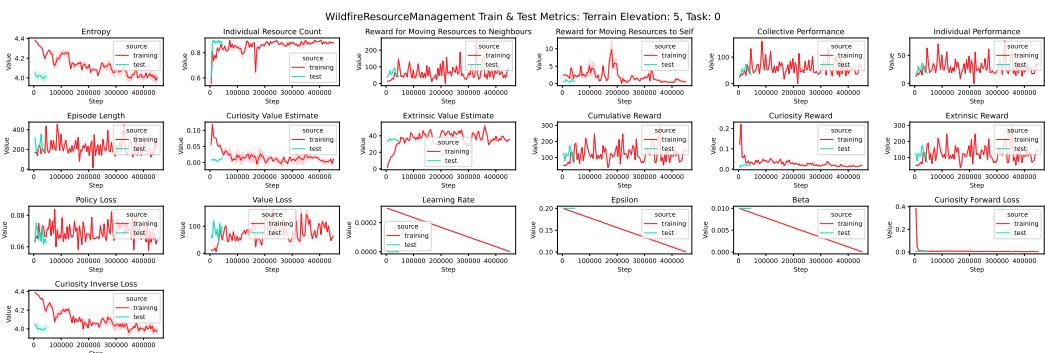

Figure 57: Wildfire Resource Management: Train & Test Metrics: Terrain Elevation 5, Task 0.

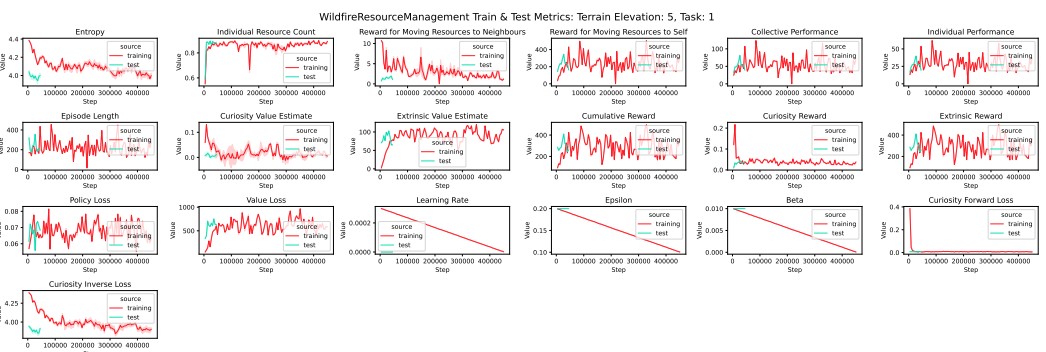

Figure 58: Wildfire Resource Management: Train & Test Metrics: Terrain Elevation 5, Task 1.

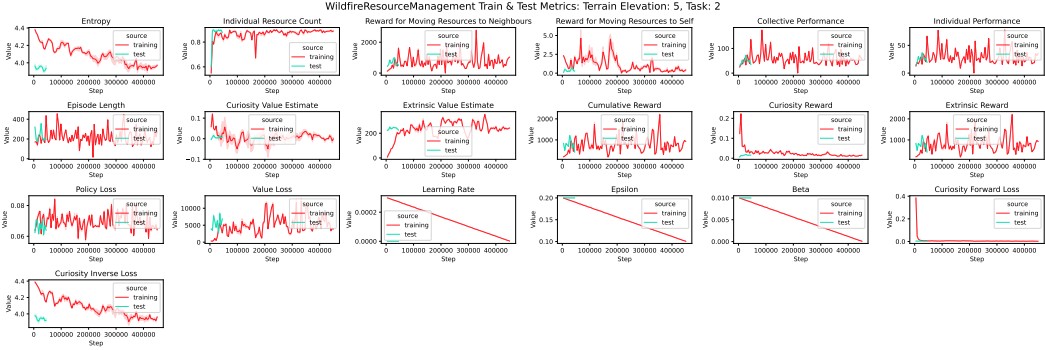

Figure 59: Wildfire Resource Management: Train & Test Metrics: Terrain Elevation 5, Task 2.

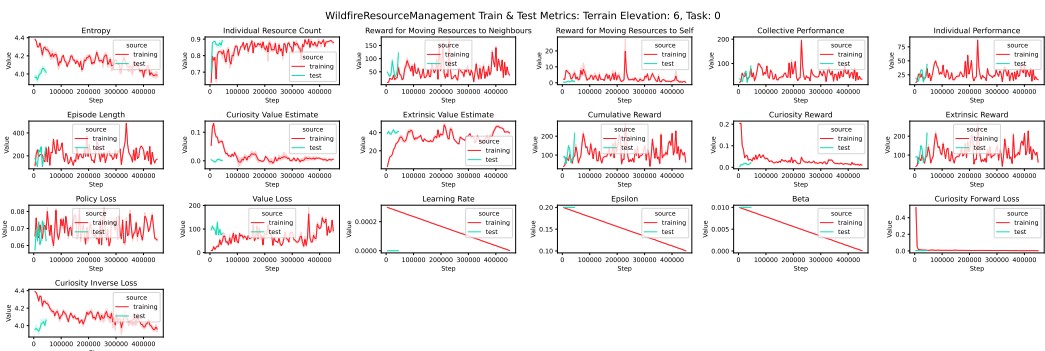

Figure 60: Wildfire Resource Management: Train & Test Metrics: Terrain Elevation 6, Task 0.

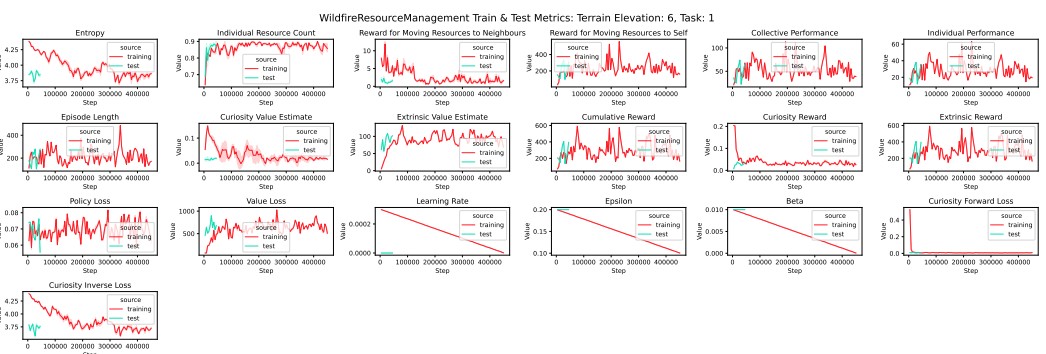

Figure 61: Wildfire Resource Management: Train & Test Metrics: Terrain Elevation 6, Task 1.

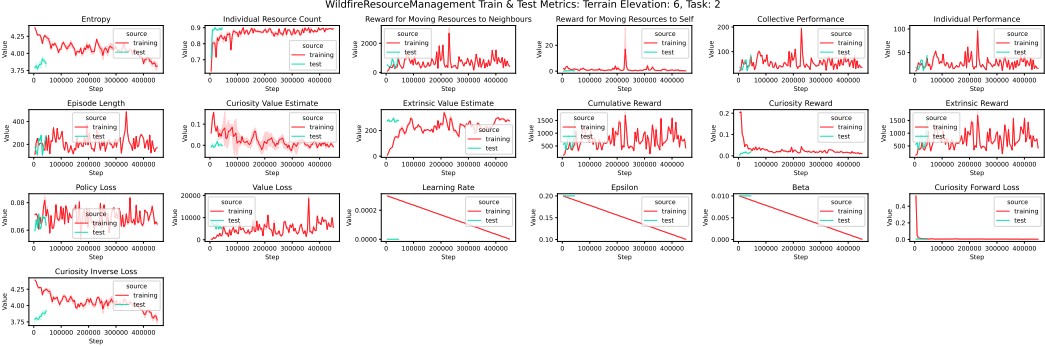

Figure 62: Wildfire Resource Management: Train & Test Metrics: Terrain Elevation 6, Task 2.

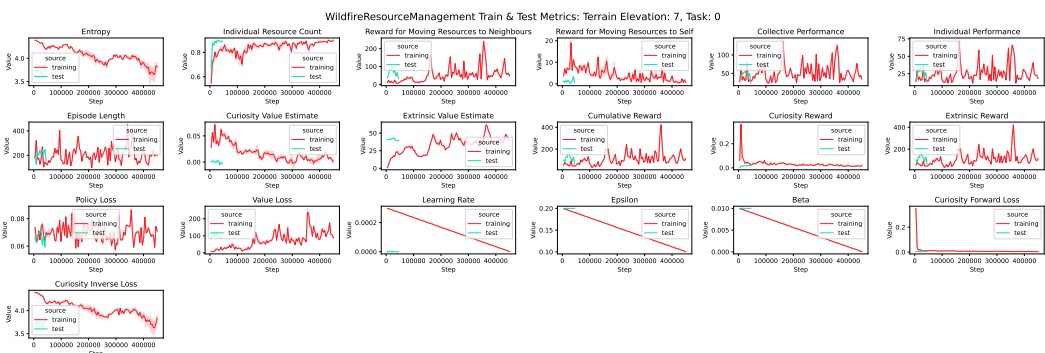

Figure 63: Wildfire Resource Management: Train & Test Metrics: Terrain Elevation 7, Task 0.

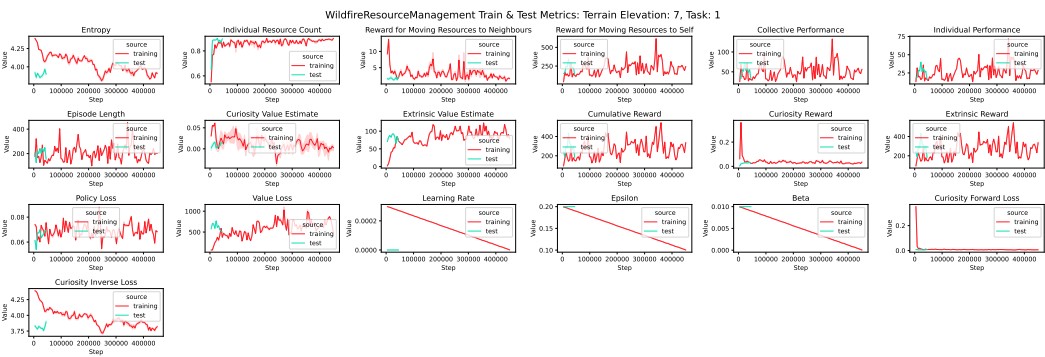

Figure 64: Wildfire Resource Management: Train & Test Metrics: Terrain Elevation 7, Task 1.

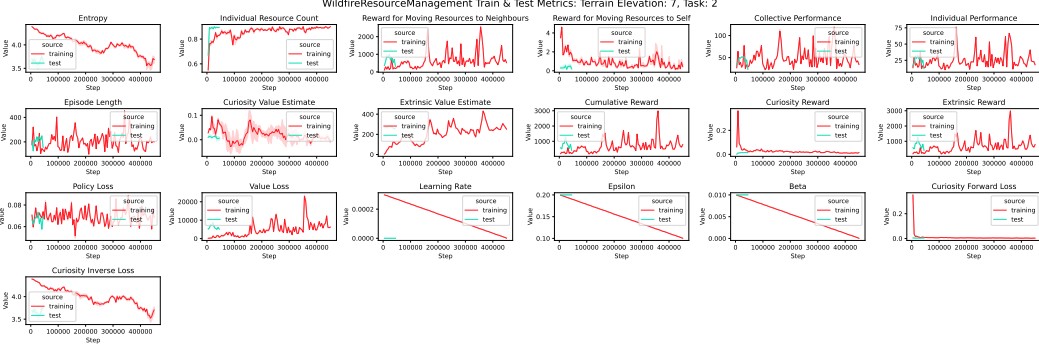

Figure 65: Wildfire Resource Management: Train & Test Metrics: Terrain Elevation 7, Task 2.

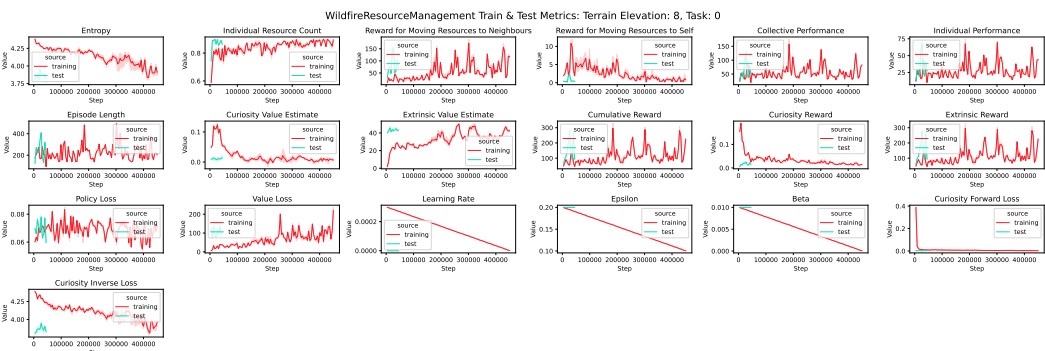

Figure 66: Wildfire Resource Management: Train & Test Metrics: Terrain Elevation 8, Task 0.

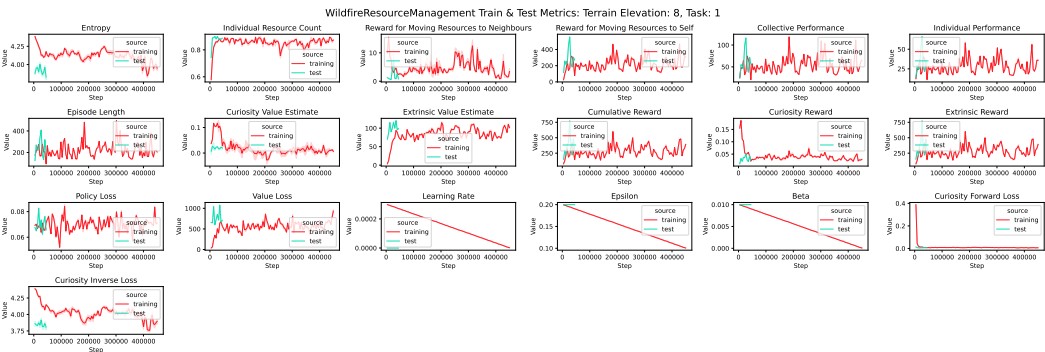

Figure 67: Wildfire Resource Management: Train & Test Metrics: Terrain Elevation 8, Task 1.

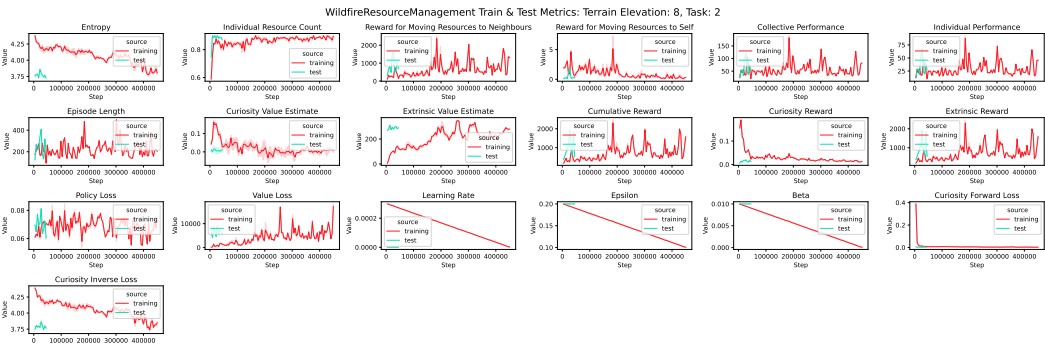

Figure 68: Wildfire Resource Management: Train & Test Metrics: Terrain Elevation 8, Task 2.

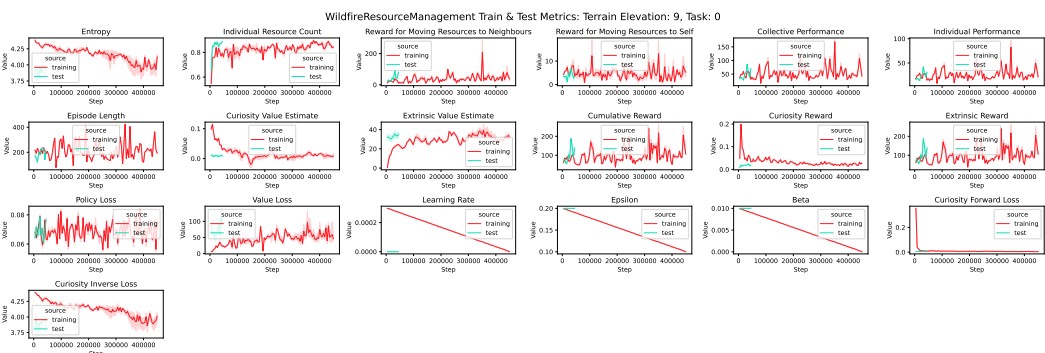

Figure 69: Wildfire Resource Management: Train & Test Metrics: Terrain Elevation 9, Task 0.

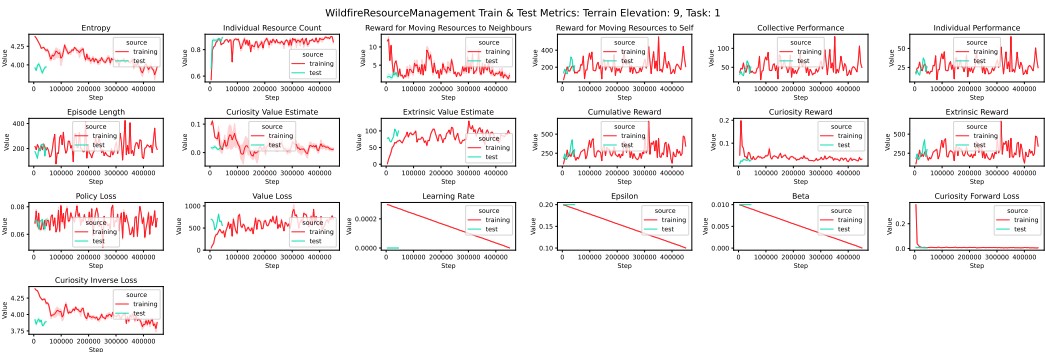

Figure 70: Wildfire Resource Management: Train & Test Metrics: Terrain Elevation 9, Task 1.

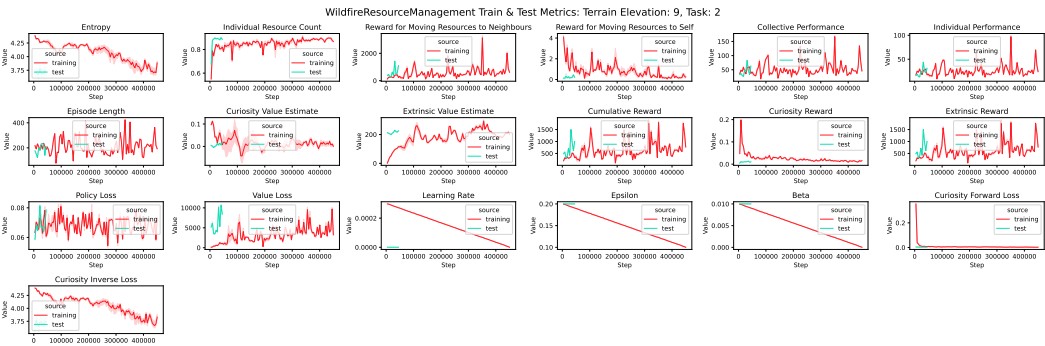

Figure 71: Wildfire Resource Management: Train & Test Metrics: Terrain Elevation 9, Task 2.

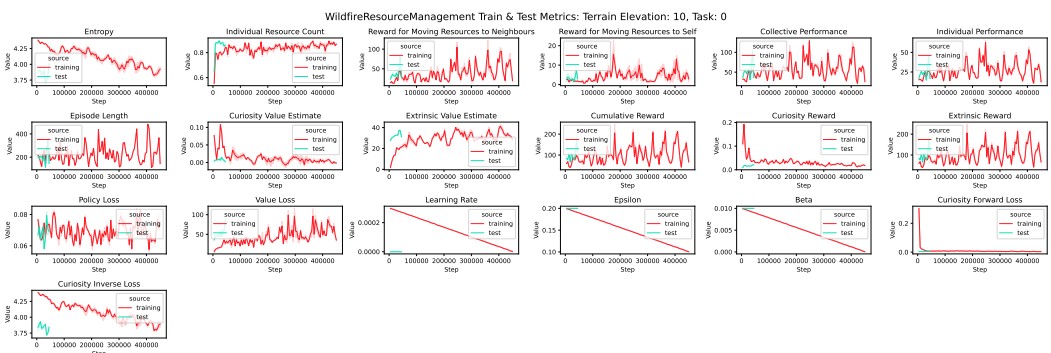

Figure 72: Wildfire Resource Management: Train & Test Metrics: Terrain Elevation 10, Task 0.

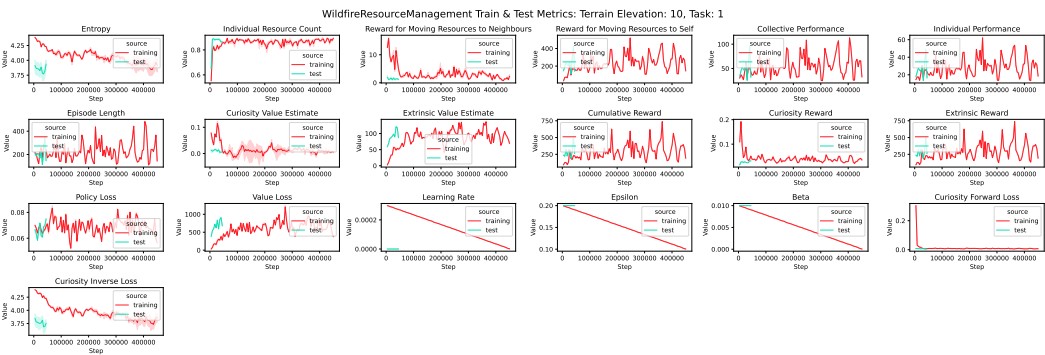

Figure 73: Wildfire Resource Management: Train & Test Metrics: Terrain Elevation 10, Task 1.

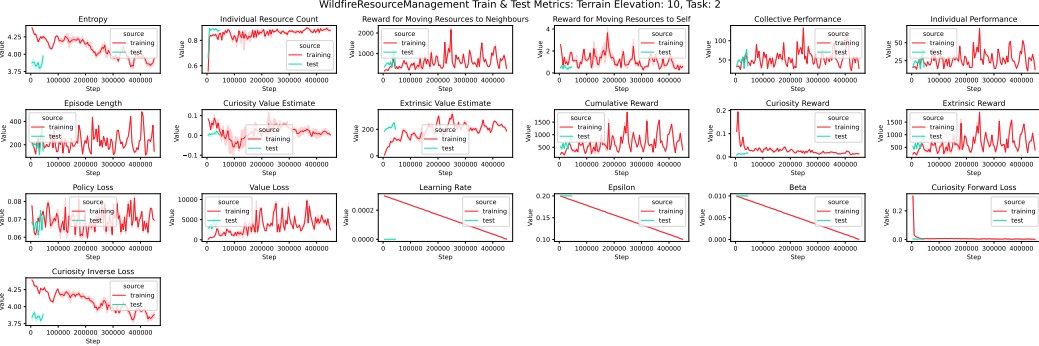

Figure 74: Wildfire Resource Management: Train & Test Metrics: Terrain Elevation 10, Task 2.

### A.10.4 Wildfire Resource Management: Average Test Metric - Task VS Pattern

Figure 75: Wildfire Resource Management: Average Train & Test Metrics.

### A.10.5 OCEAN PLASTIC COLLECTION: TRAIN & TEST METRICS

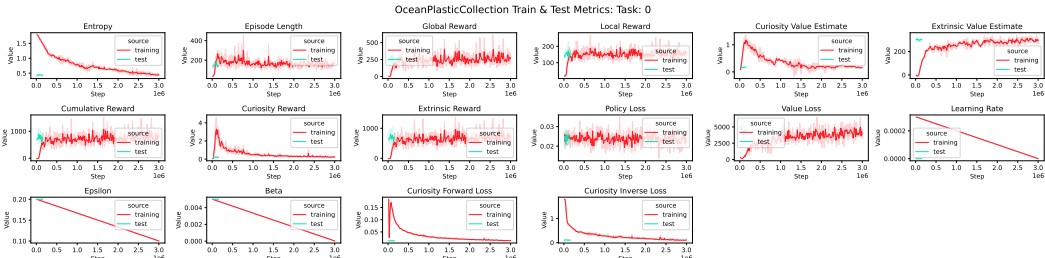

Figure 76: Ocean Plastic Collection: Train & Test Metrics: Task 0.

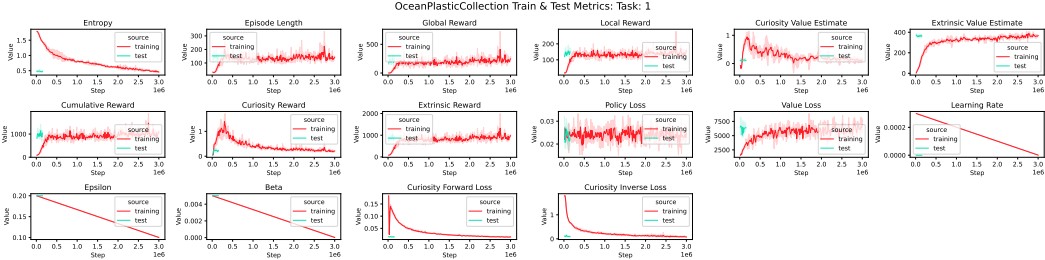

Figure 77: Ocean Plastic Collection: Train & Test Metrics: Task 1.

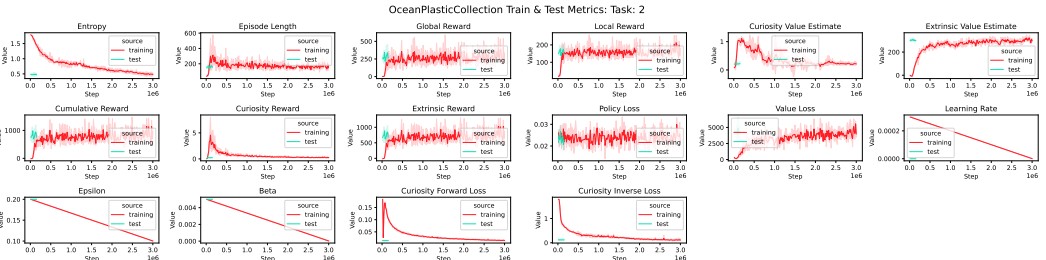

Figure 78: Ocean Plastic Collection: Train & Test Metrics: Task 2.

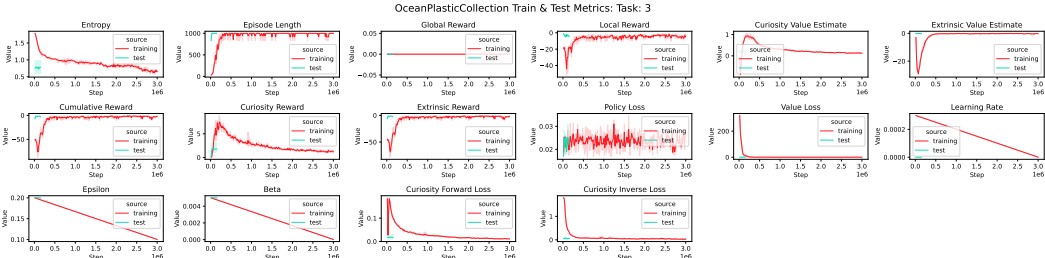

Figure 79: Ocean Plastic Collection: Train & Test Metrics: Task 3.

### A.10.6 OCEAN PLASTIC COLLECTION: AVERAGE TEST METRIC - TASK VS PATTERN

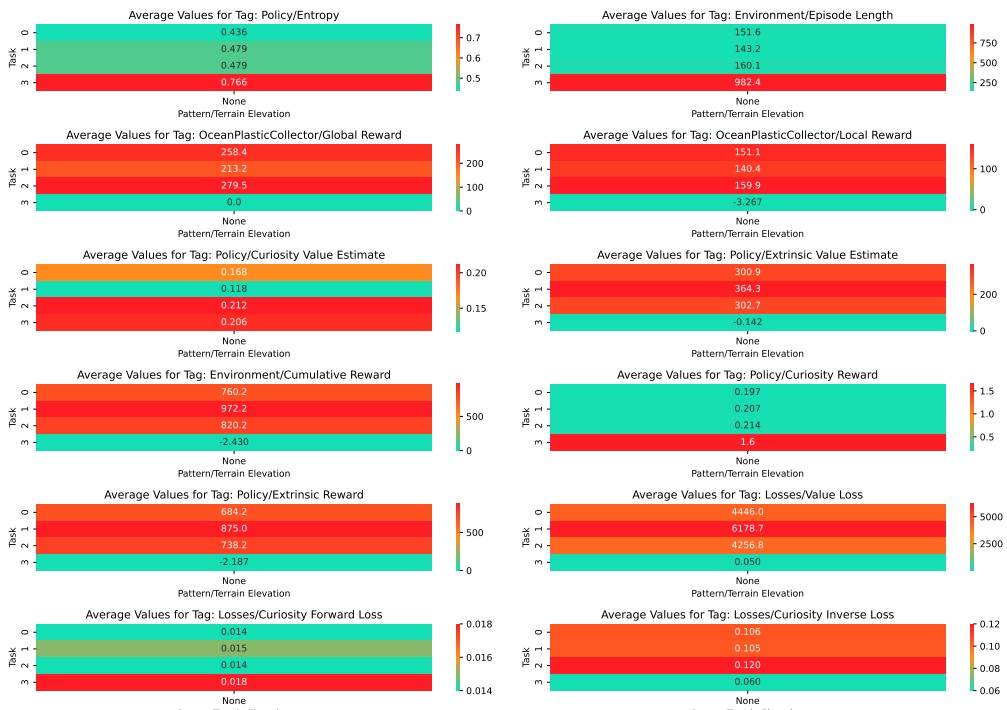

Figure 80: Ocean Plastic Collection: Average Train & Test Metrics.

### A.10.7 DRONE-BASED REFORESTATION: TRAIN & TEST METRICS

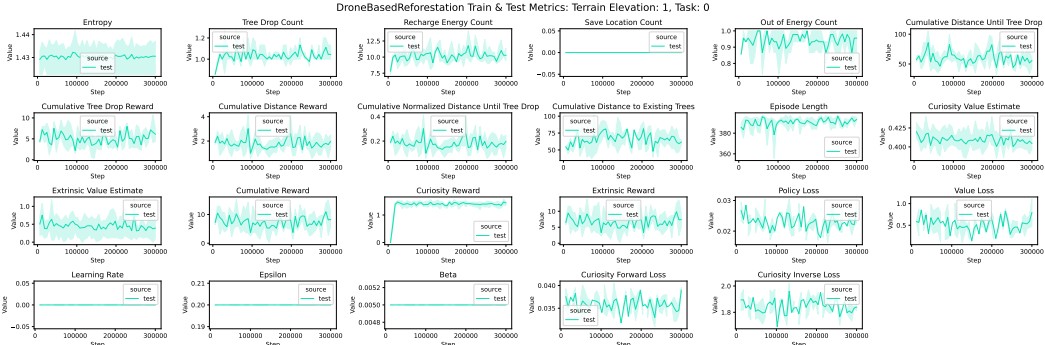

Figure 81: Drone-Based Reforestation: Train & Test Metrics: Terrain Elevation 1, Task 0.

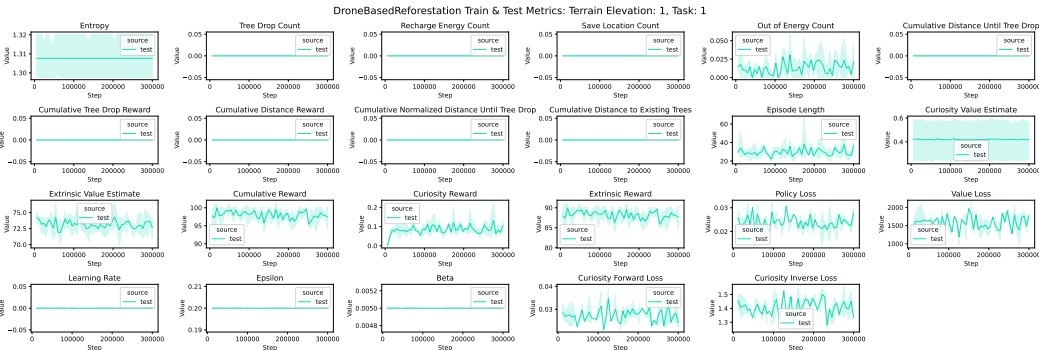

Figure 82: Drone-Based Reforestation: Train & Test Metrics: Terrain Elevation 1, Task 1.

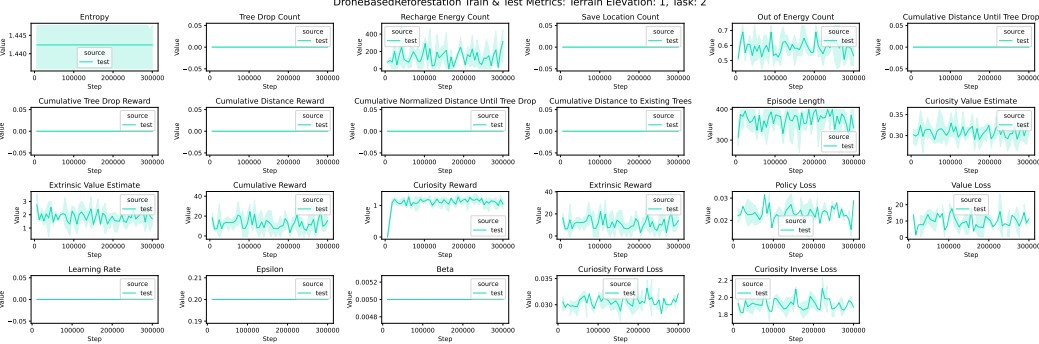

Figure 83: Drone-Based Reforestation: Train & Test Metrics: Terrain Elevation 1, Task 2.

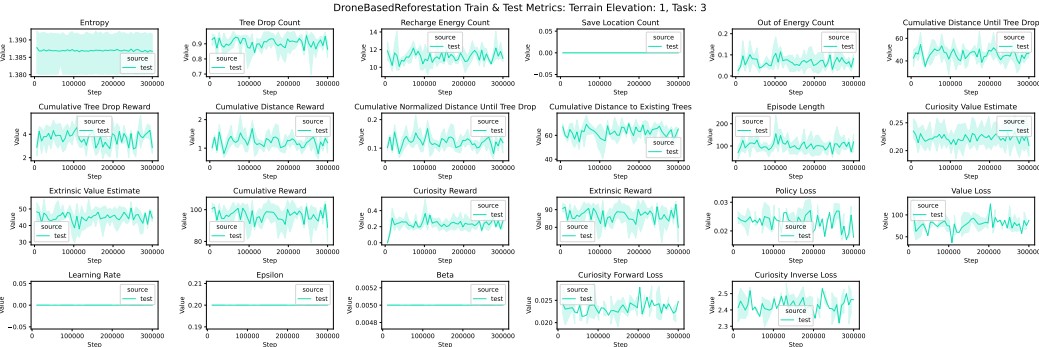

Figure 84: Drone-Based Reforestation: Train & Test Metrics: Terrain Elevation 1, Task 3.

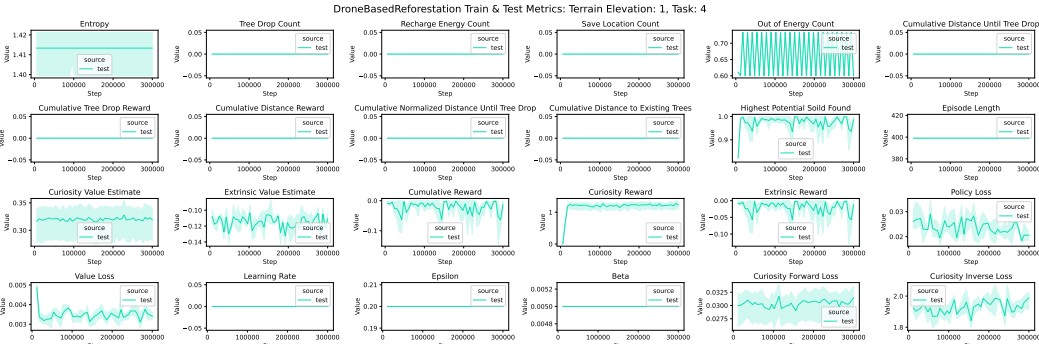

Figure 85: Drone-Based Reforestation: Train & Test Metrics: Terrain Elevation 1, Task 4.

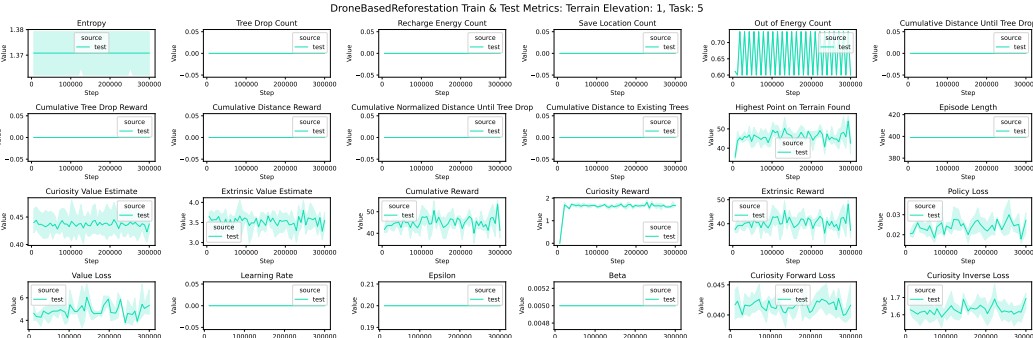

Figure 86: Drone-Based Reforestation: Train & Test Metrics: Terrain Elevation 1, Task 5.

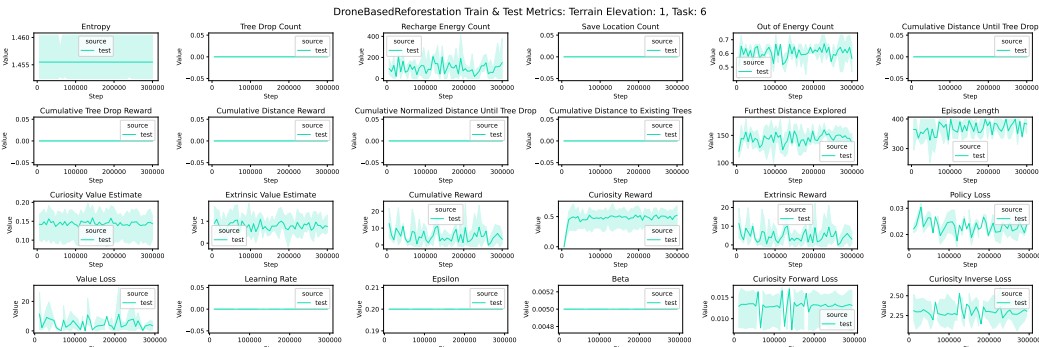

Figure 87: Drone-Based Reforestation: Train & Test Metrics: Terrain Elevation 1, Task 6.

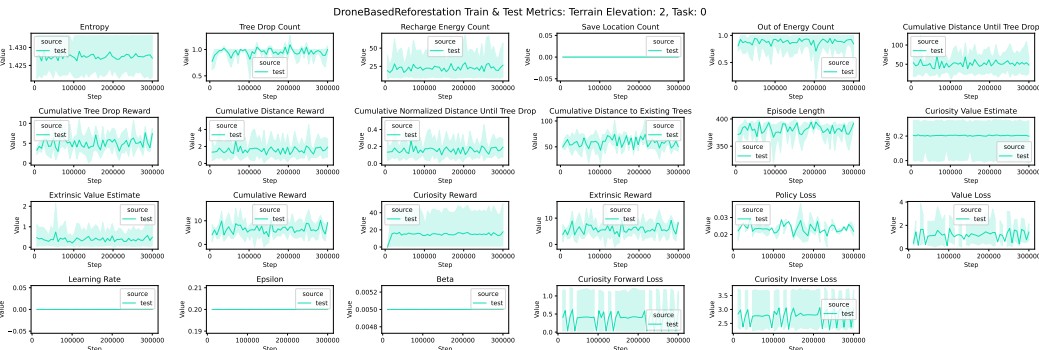

Figure 88: Drone-Based Reforestation: Train & Test Metrics: Terrain Elevation 2, Task 0.

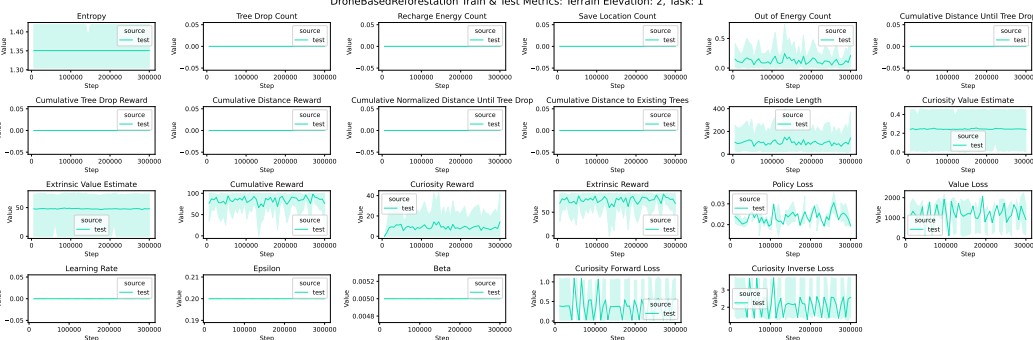

Figure 89: Drone-Based Reforestation: Train & Test Metrics: Terrain Elevation 2, Task 1.

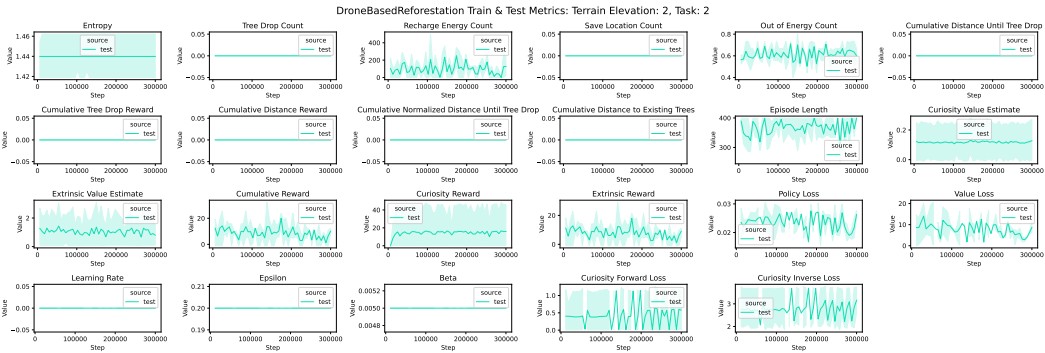

Figure 90: Drone-Based Reforestation: Train & Test Metrics: Terrain Elevation 2, Task 2.

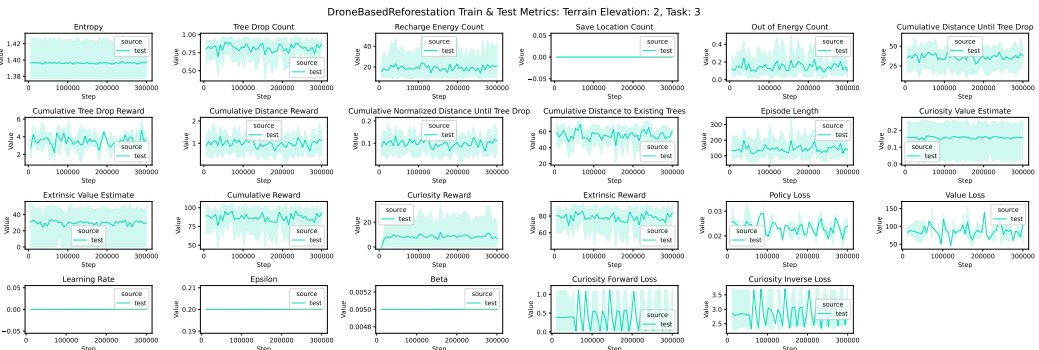

Figure 91: Drone-Based Reforestation: Train & Test Metrics: Terrain Elevation 2, Task 3.

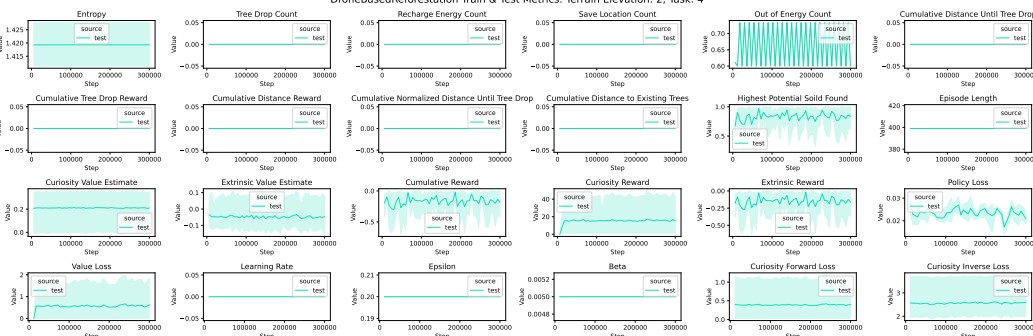

Figure 92: Drone-Based Reforestation: Train & Test Metrics: Terrain Elevation 2, Task 4.

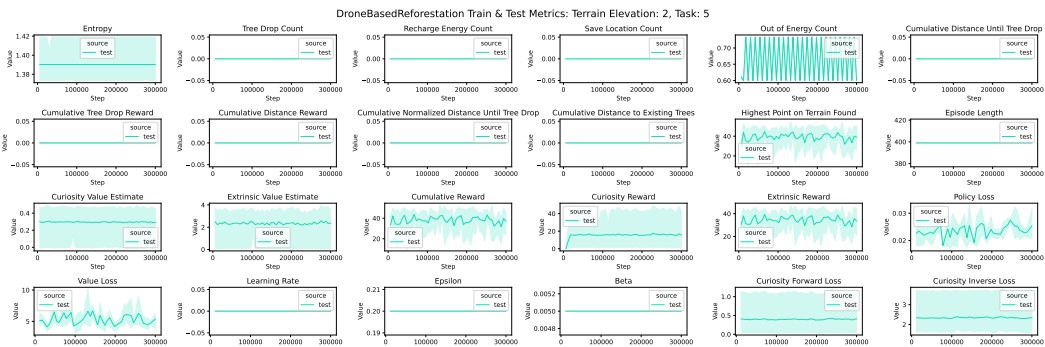

Figure 93: Drone-Based Reforestation: Train & Test Metrics: Terrain Elevation 2, Task 5.

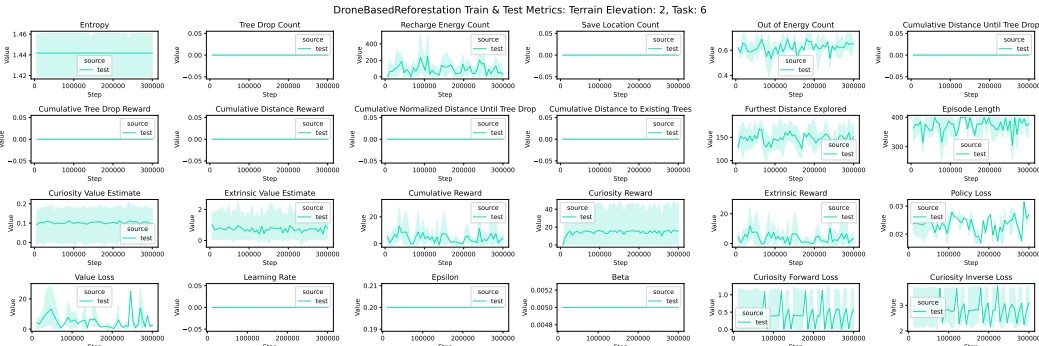

Figure 94: Drone-Based Reforestation: Train & Test Metrics: Terrain Elevation 2, Task 6.

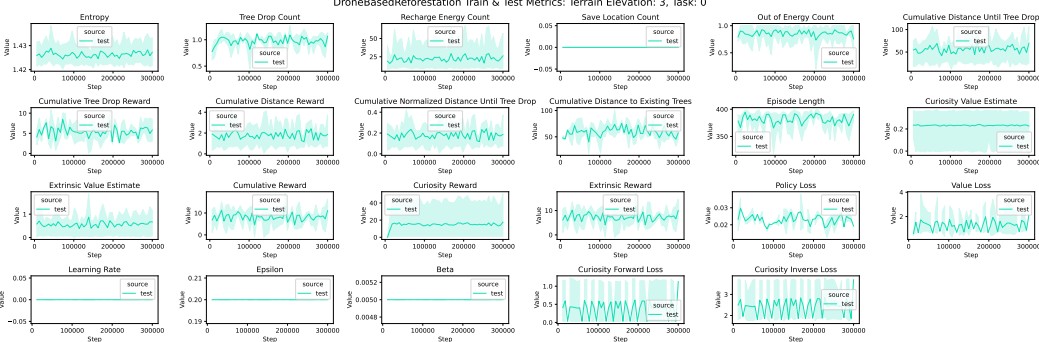

Figure 95: Drone-Based Reforestation: Train & Test Metrics: Terrain Elevation 3, Task 0.

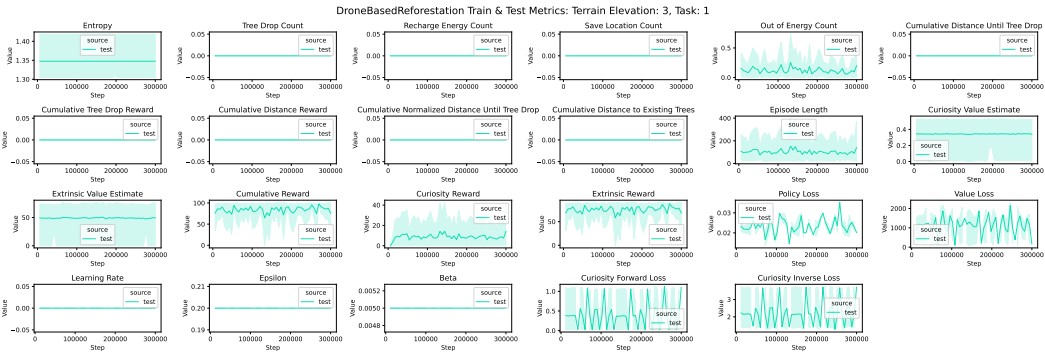

Figure 96: Drone-Based Reforestation: Train & Test Metrics: Terrain Elevation 3, Task 1.

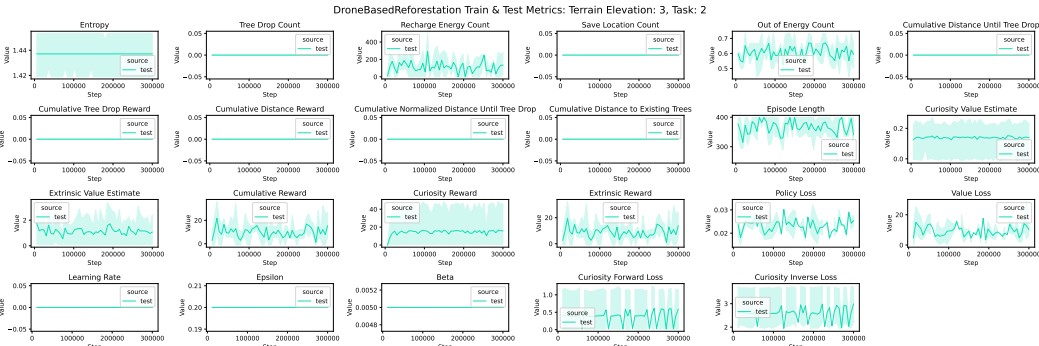

Figure 97: Drone-Based Reforestation: Train & Test Metrics: Terrain Elevation 3, Task 2.

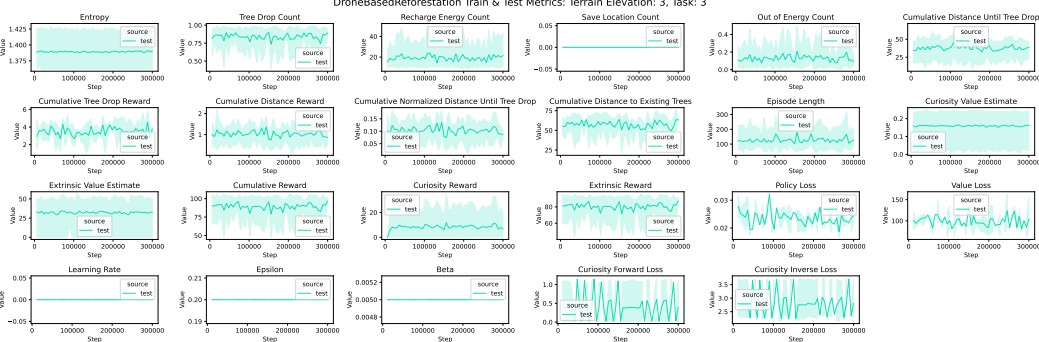

Figure 98: Drone-Based Reforestation: Train & Test Metrics: Terrain Elevation 3, Task 3.

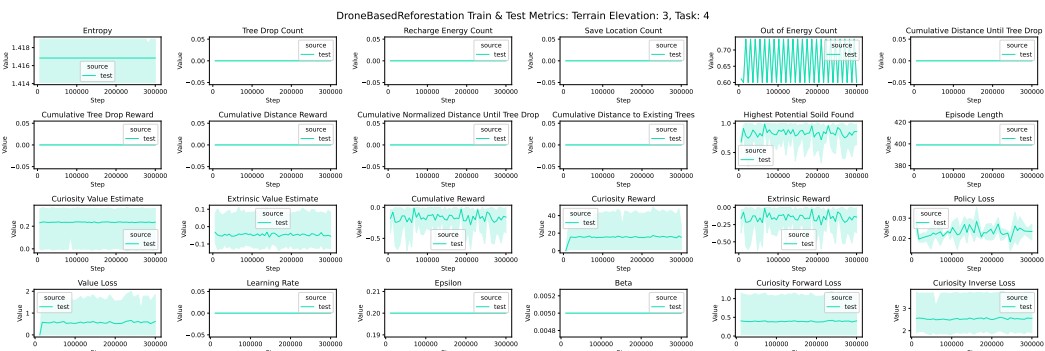

Figure 99: Drone-Based Reforestation: Train & Test Metrics: Terrain Elevation 3, Task 4.

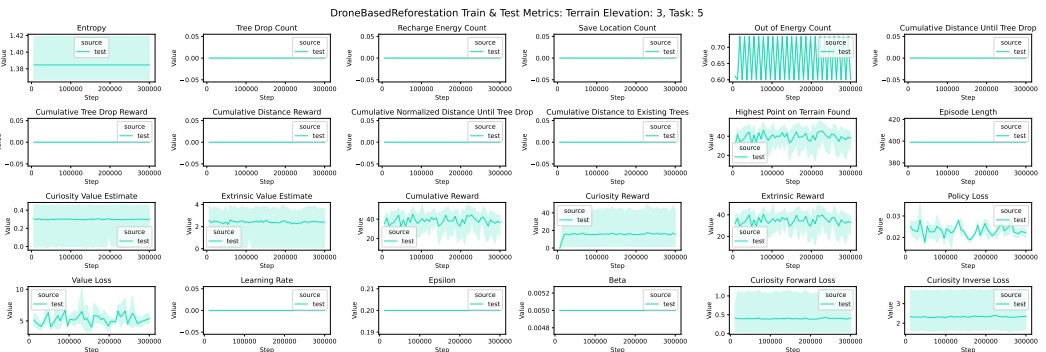

Figure 100: Drone-Based Reforestation: Train & Test Metrics: Terrain Elevation 3, Task 5.

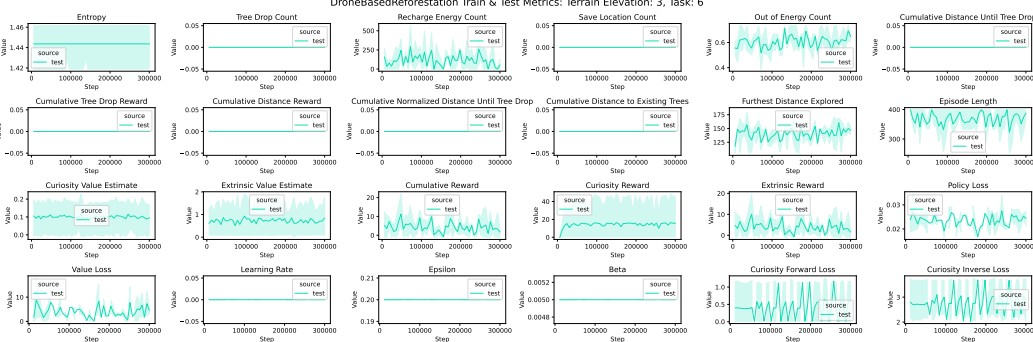

Figure 101: Drone-Based Reforestation: Train & Test Metrics: Terrain Elevation 3, Task 6.

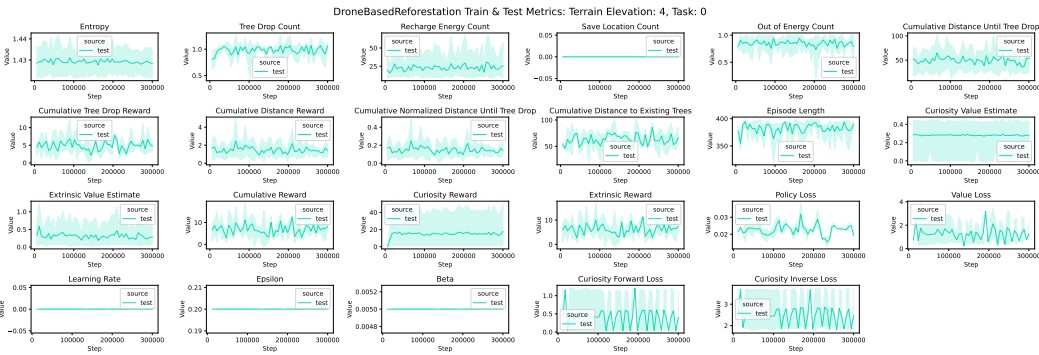

Figure 102: Drone-Based Reforestation: Train & Test Metrics: Terrain Elevation 4, Task 0.

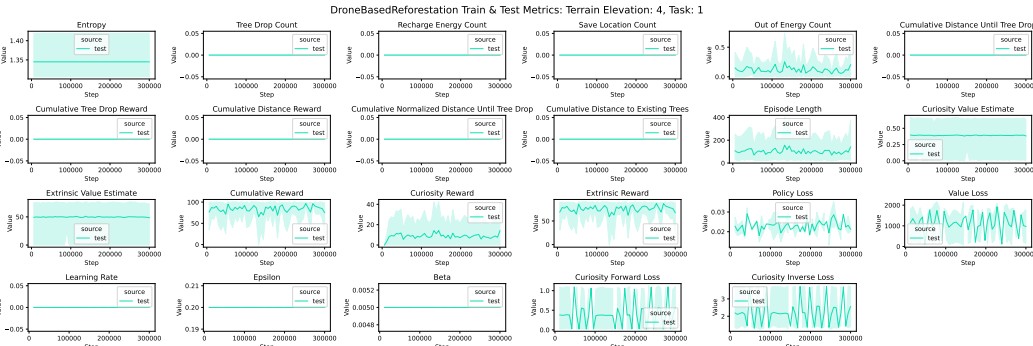

Figure 103: Drone-Based Reforestation: Train & Test Metrics: Terrain Elevation 4, Task 1.

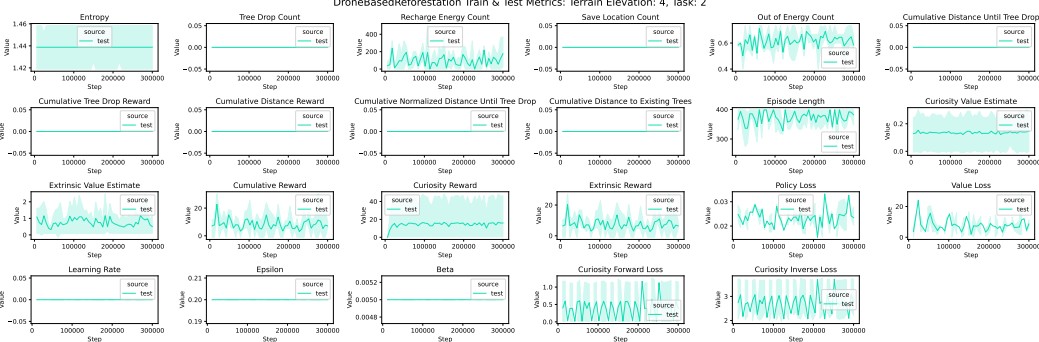

Figure 104: Drone-Based Reforestation: Train & Test Metrics: Terrain Elevation 4, Task 2.

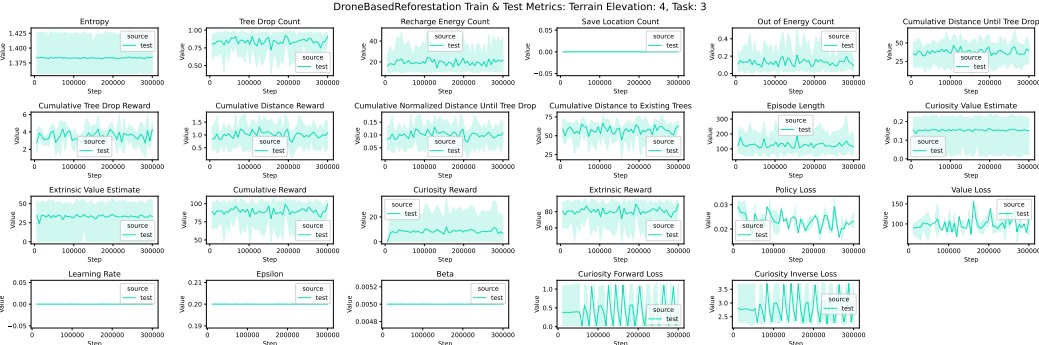

Figure 105: Drone-Based Reforestation: Train & Test Metrics: Terrain Elevation 4, Task 3.

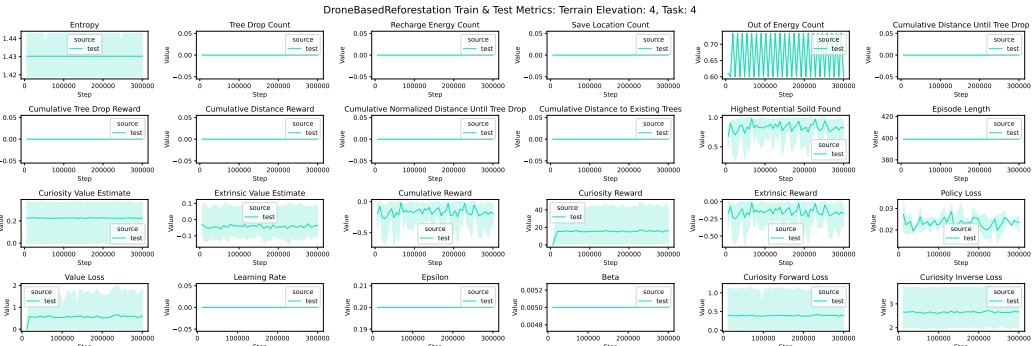

Figure 106: Drone-Based Reforestation: Train & Test Metrics: Terrain Elevation 4, Task 4.

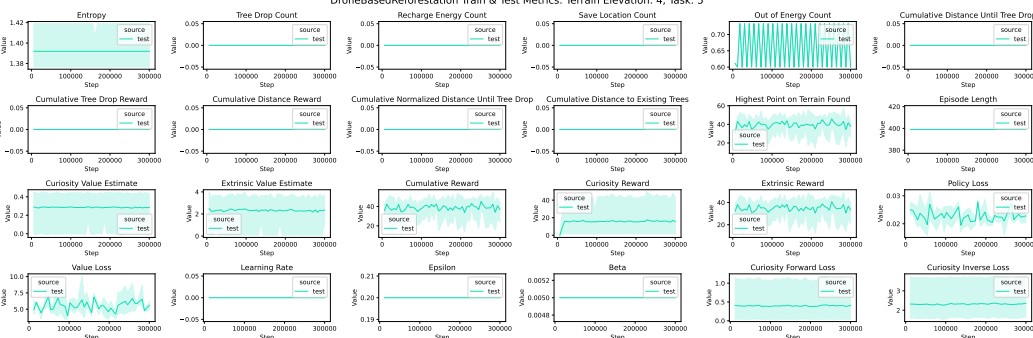

Figure 107: Drone-Based Reforestation: Train & Test Metrics: Terrain Elevation 4, Task 5.

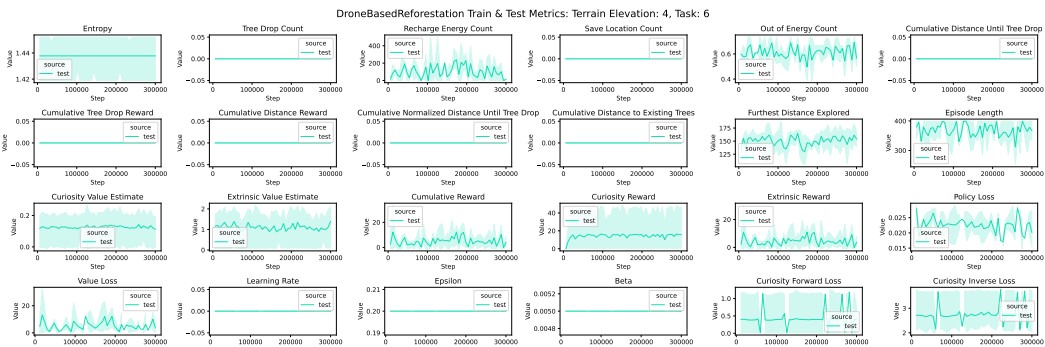

Figure 108: Drone-Based Reforestation: Train & Test Metrics: Terrain Elevation 4, Task 6.

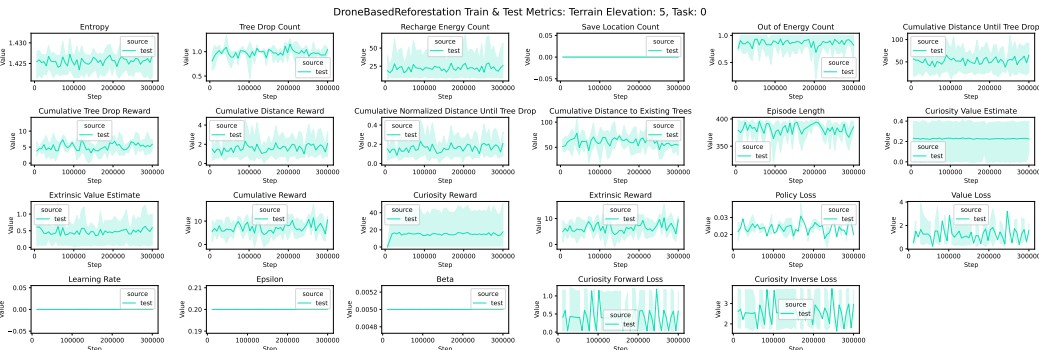

Figure 109: Drone-Based Reforestation: Train & Test Metrics: Terrain Elevation 5, Task 0.

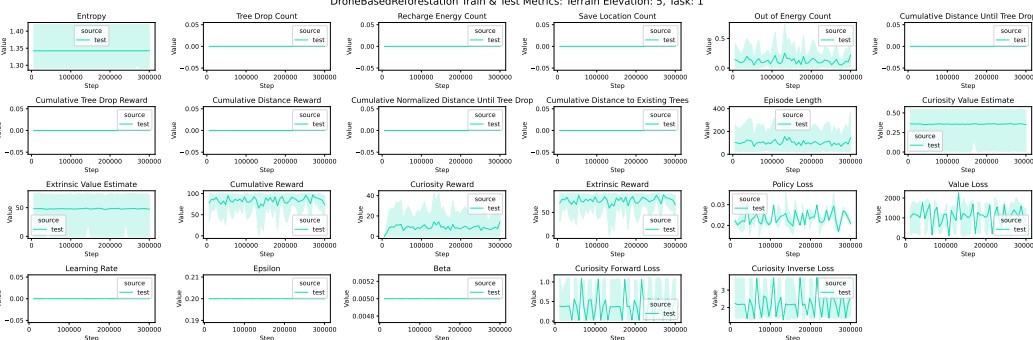

Figure 110: Drone-Based Reforestation: Train & Test Metrics: Terrain Elevation 5, Task 1.

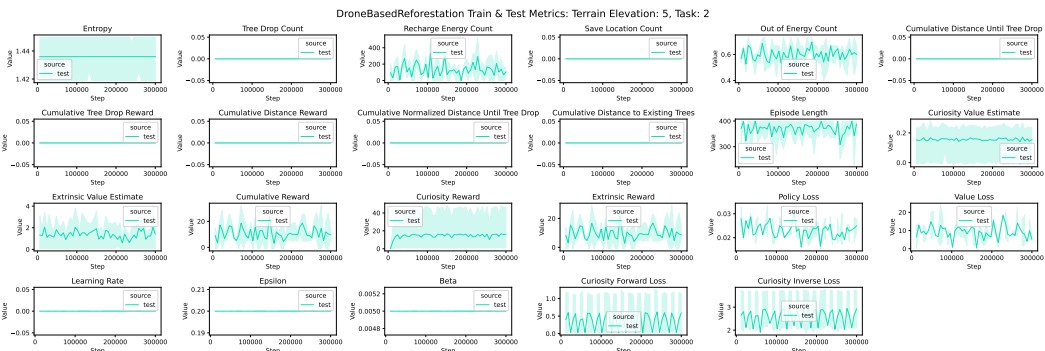

Figure 111: Drone-Based Reforestation: Train & Test Metrics: Terrain Elevation 5, Task 2.

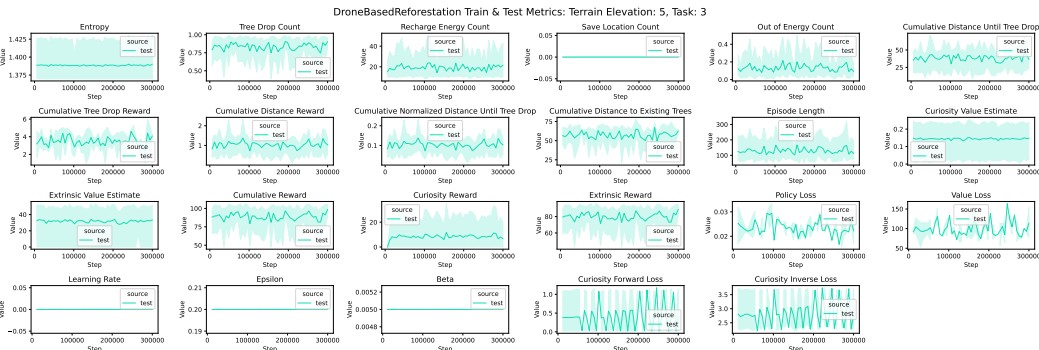

Figure 112: Drone-Based Reforestation: Train & Test Metrics: Terrain Elevation 5, Task 3.

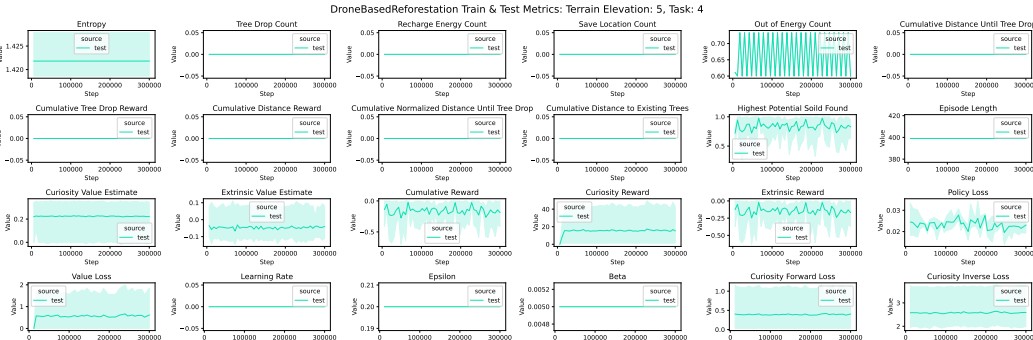

Figure 113: Drone-Based Reforestation: Train & Test Metrics: Terrain Elevation 5, Task 4.

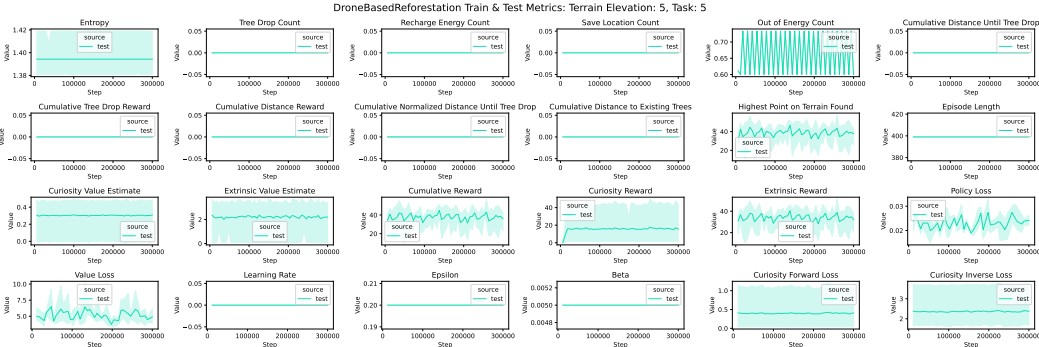

Figure 114: Drone-Based Reforestation: Train & Test Metrics: Terrain Elevation 5, Task 5.

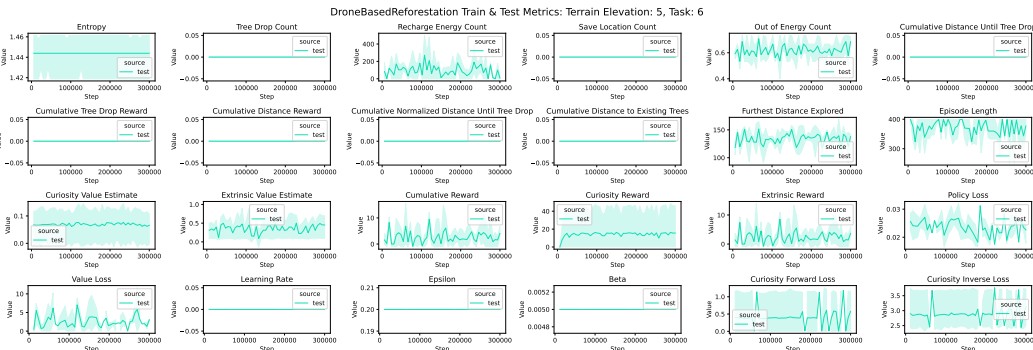

Figure 115: Drone-Based Reforestation: Train & Test Metrics: Terrain Elevation 5, Task 6.

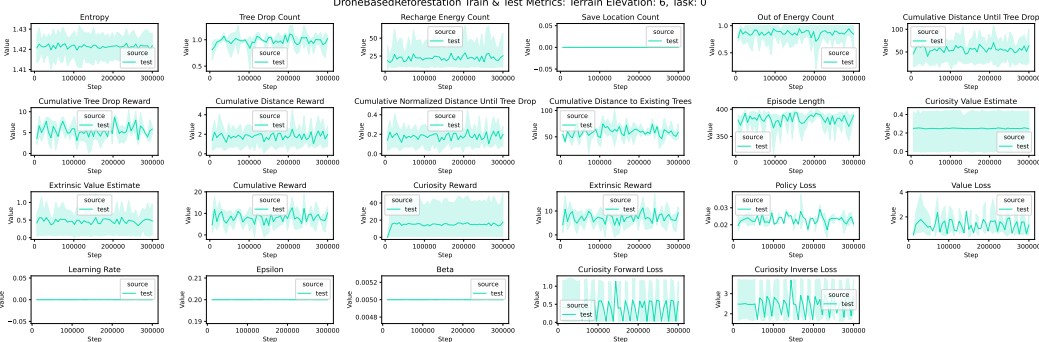

Figure 116: Drone-Based Reforestation: Train & Test Metrics: Terrain Elevation 6, Task 0.

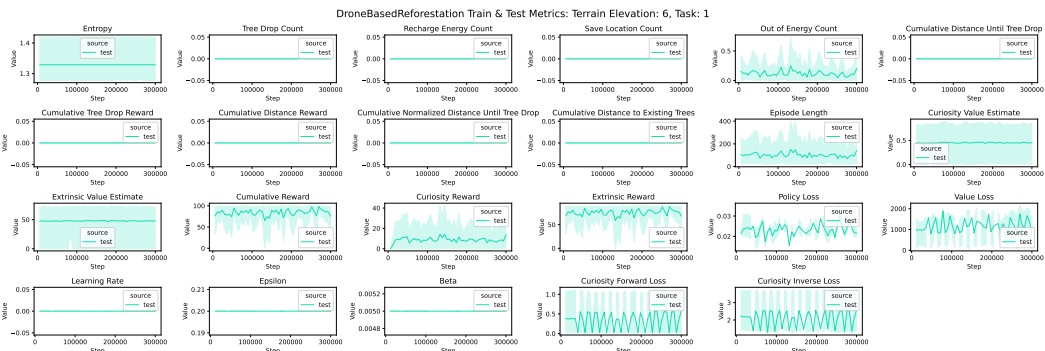

Figure 117: Drone-Based Reforestation: Train & Test Metrics: Terrain Elevation 6, Task 1.

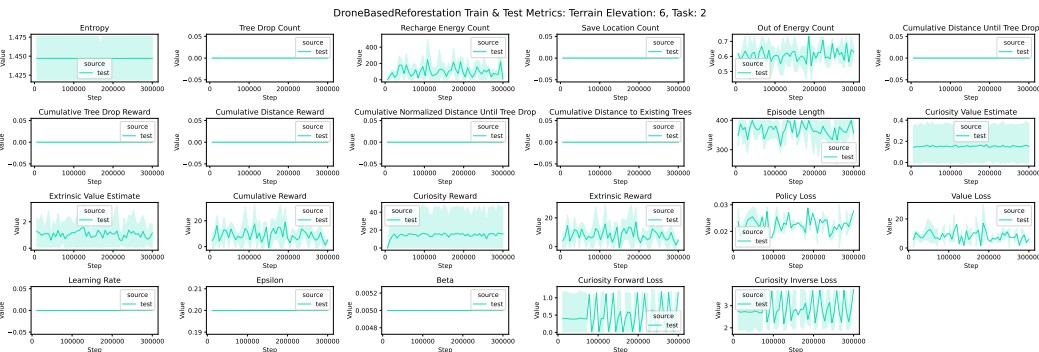

Figure 118: Drone-Based Reforestation: Train & Test Metrics: Terrain Elevation 6, Task 2.

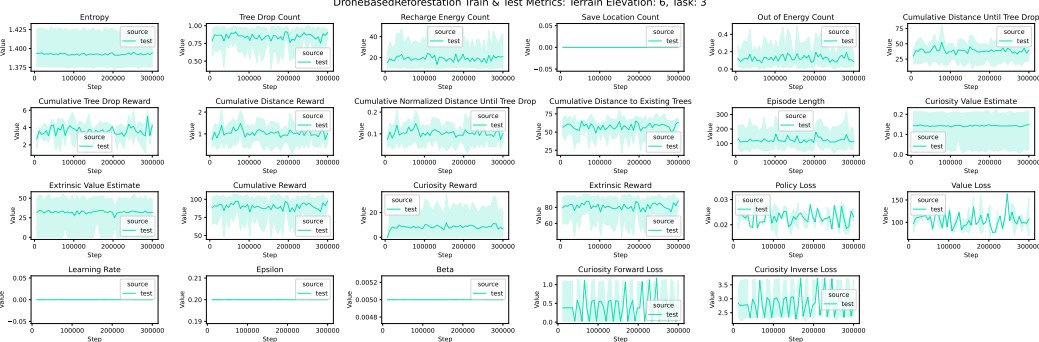

Figure 119: Drone-Based Reforestation: Train & Test Metrics: Terrain Elevation 6, Task 3.

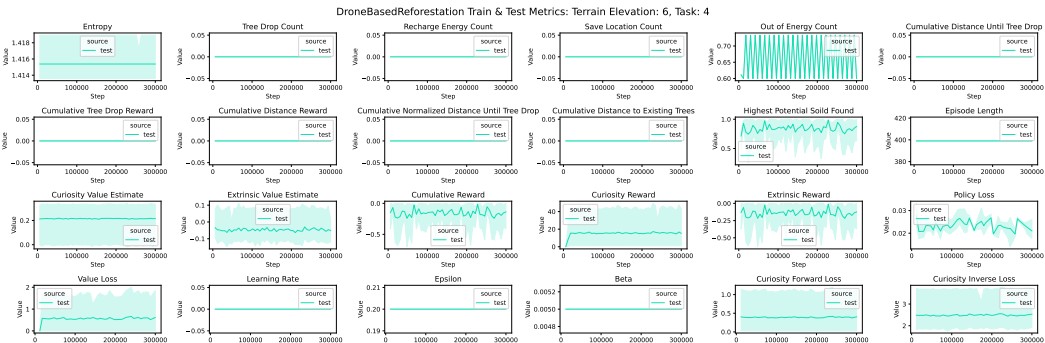

Figure 120: Drone-Based Reforestation: Train & Test Metrics: Terrain Elevation 6, Task 4.

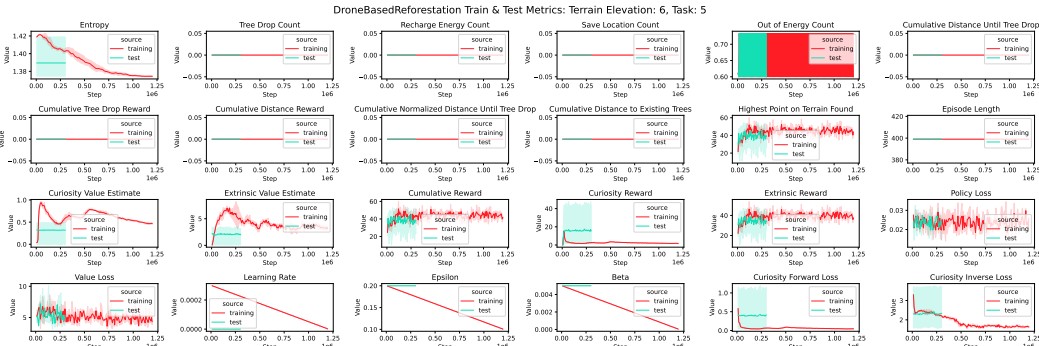

Figure 121: Drone-Based Reforestation: Train & Test Metrics: Terrain Elevation 6, Task 5.

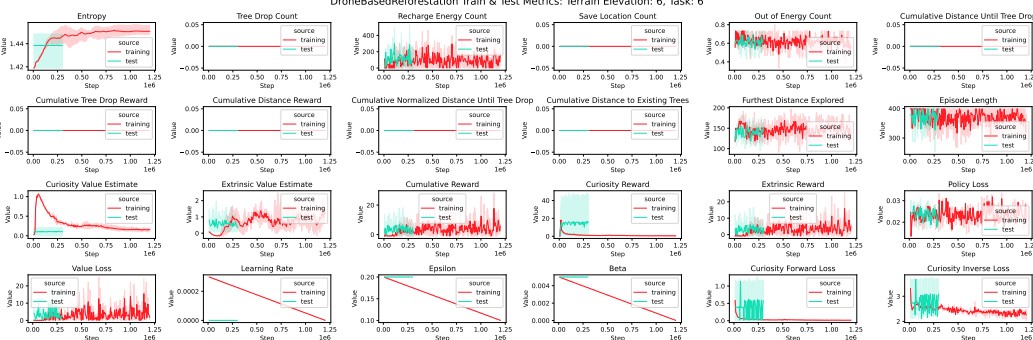

Figure 122: Drone-Based Reforestation: Train & Test Metrics: Terrain Elevation 6, Task 6.

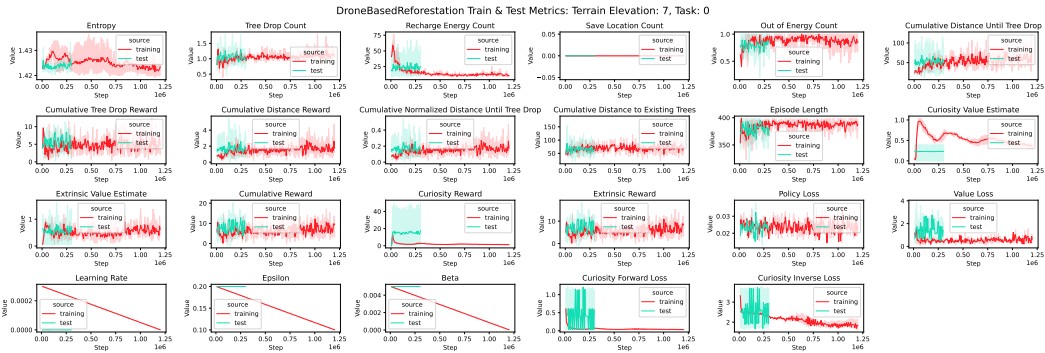

Figure 123: Drone-Based Reforestation: Train & Test Metrics: Terrain Elevation 7, Task 0.

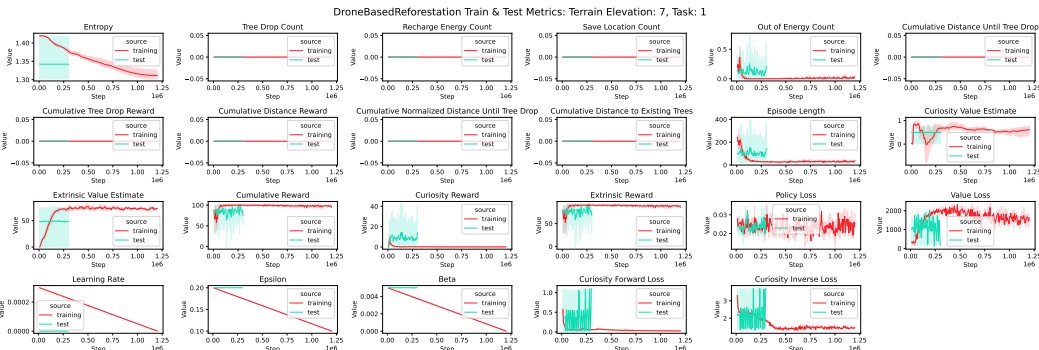

Figure 124: Drone-Based Reforestation: Train & Test Metrics: Terrain Elevation 7, Task 1.

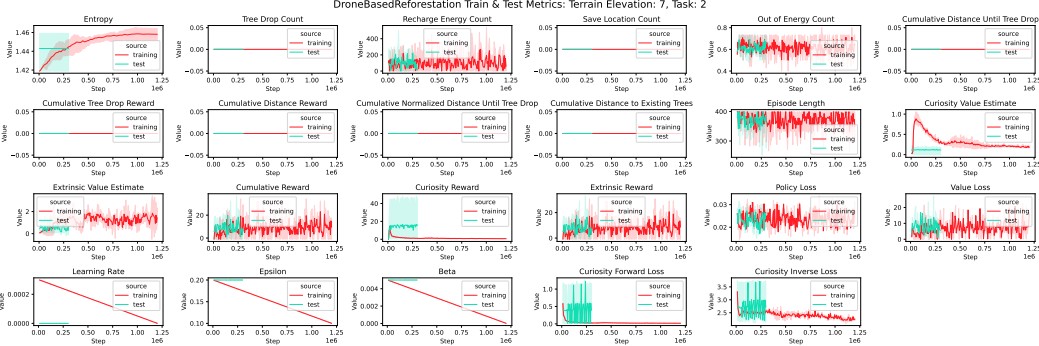

Figure 125: Drone-Based Reforestation: Train & Test Metrics: Terrain Elevation 7, Task 2.

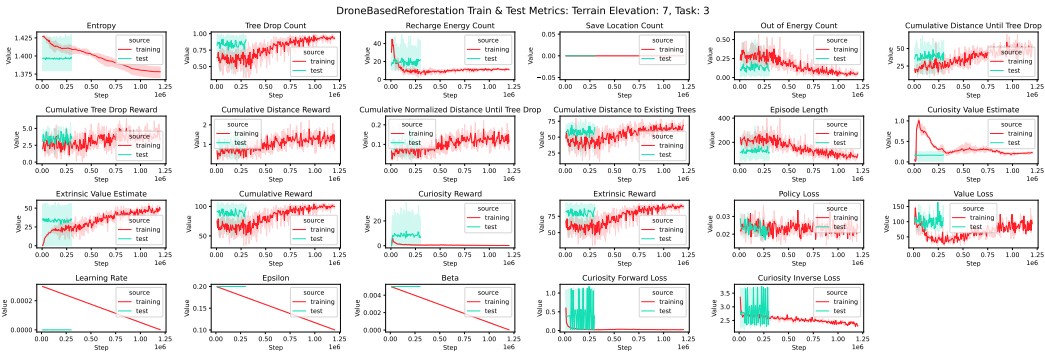

Figure 126: Drone-Based Reforestation: Train & Test Metrics: Terrain Elevation 7, Task 3.

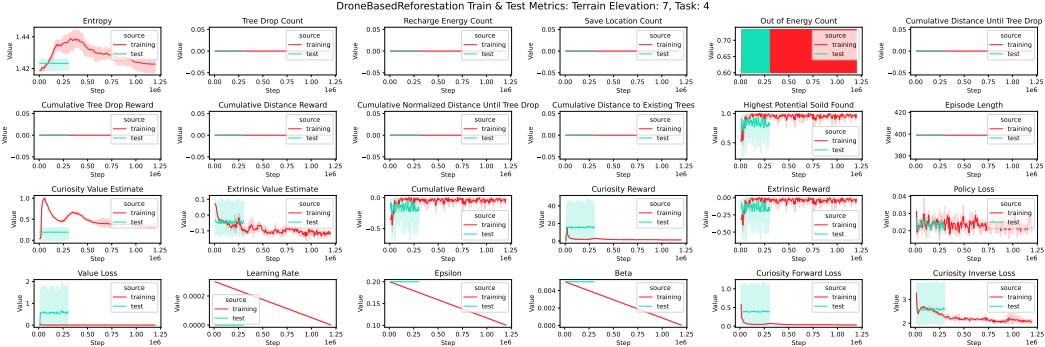

Figure 127: Drone-Based Reforestation: Train & Test Metrics: Terrain Elevation 7, Task 4.

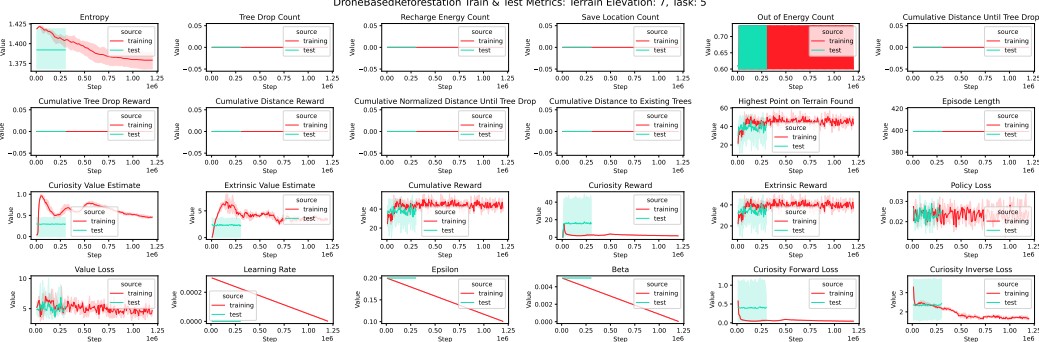

Figure 128: Drone-Based Reforestation: Train & Test Metrics: Terrain Elevation 7, Task 5.

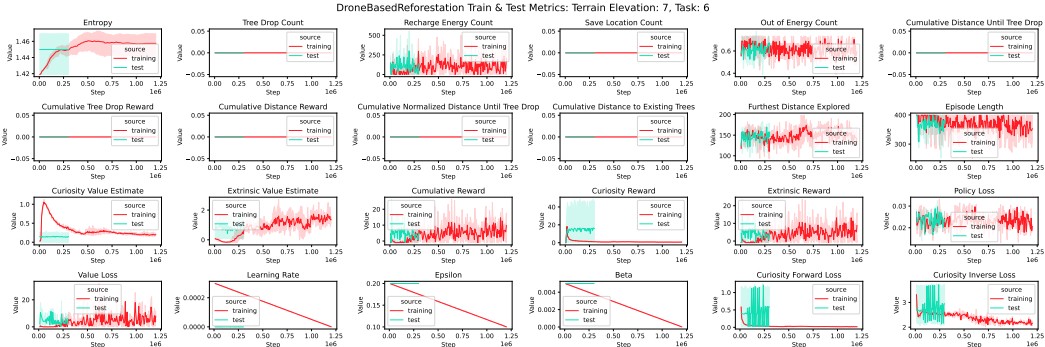

Figure 129: Drone-Based Reforestation: Train & Test Metrics: Terrain Elevation 7, Task 6.

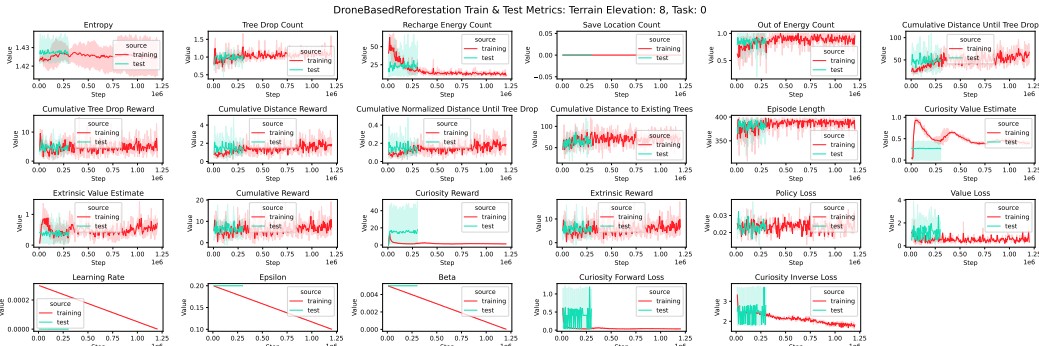

Figure 130: Drone-Based Reforestation: Train & Test Metrics: Terrain Elevation 8, Task 0.

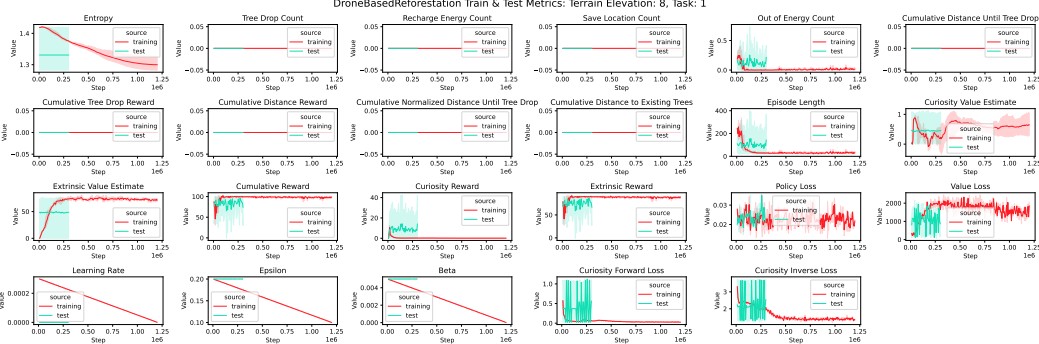

Figure 131: Drone-Based Reforestation: Train & Test Metrics: Terrain Elevation 8, Task 1.

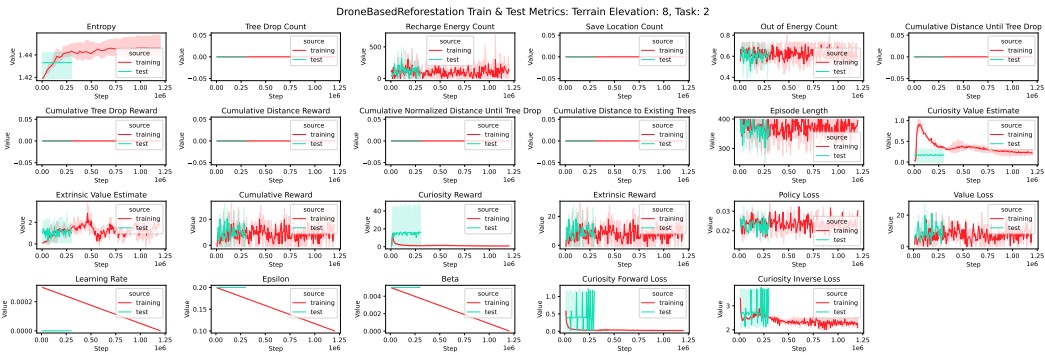

Figure 132: Drone-Based Reforestation: Train & Test Metrics: Terrain Elevation 8, Task 2.

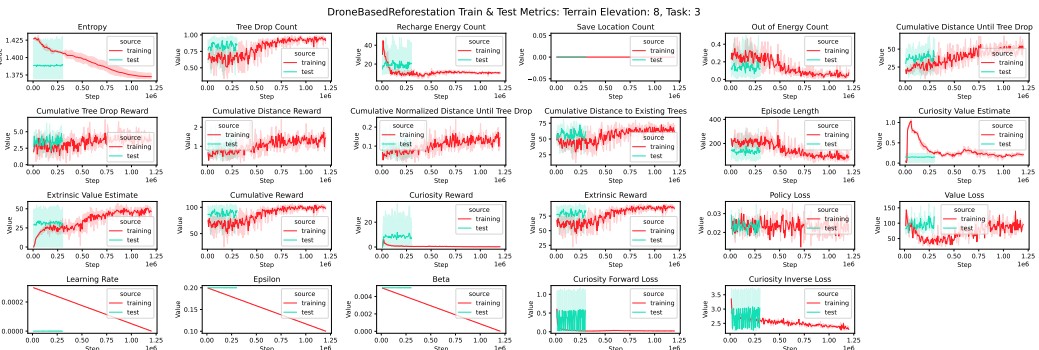

Figure 133: Drone-Based Reforestation: Train & Test Metrics: Terrain Elevation 8, Task 3.

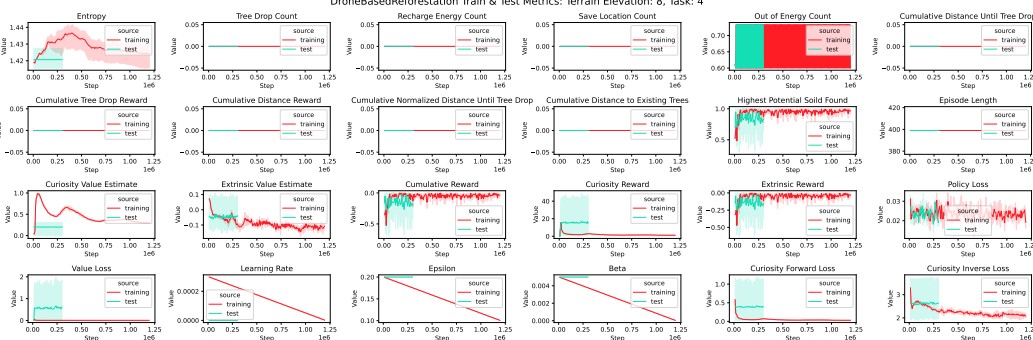

Figure 134: Drone-Based Reforestation: Train & Test Metrics: Terrain Elevation 8, Task 4.

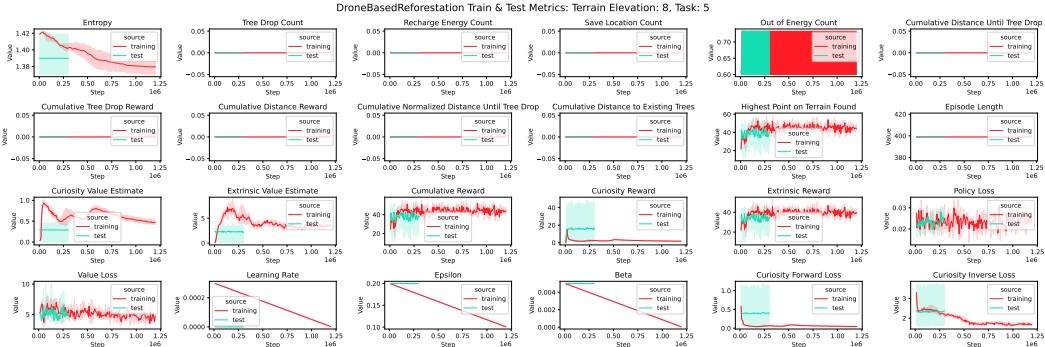

Figure 135: Drone-Based Reforestation: Train & Test Metrics: Terrain Elevation 8, Task 5.

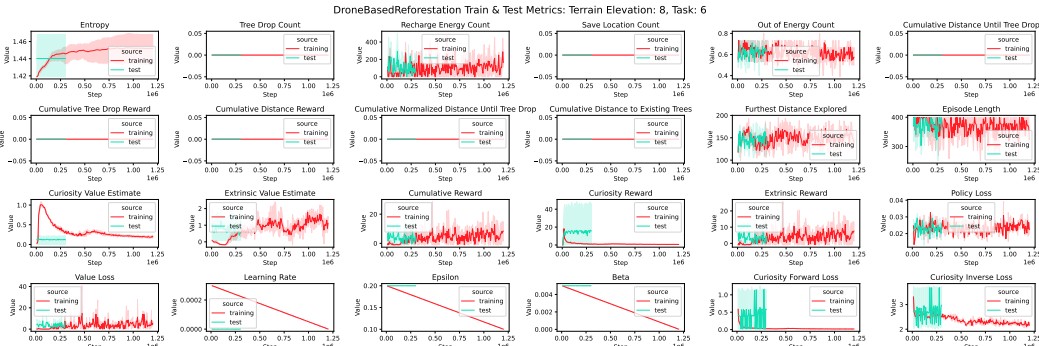

Figure 136: Drone-Based Reforestation: Train & Test Metrics: Terrain Elevation 8, Task 6.

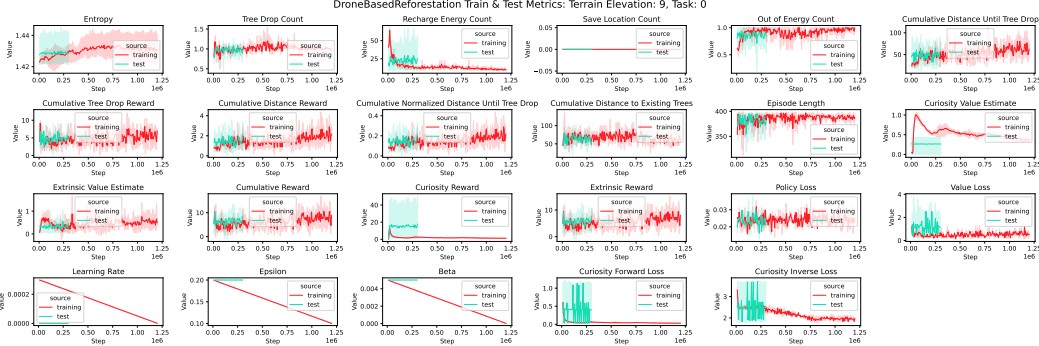

Figure 137: Drone-Based Reforestation: Train & Test Metrics: Terrain Elevation 9, Task 0.

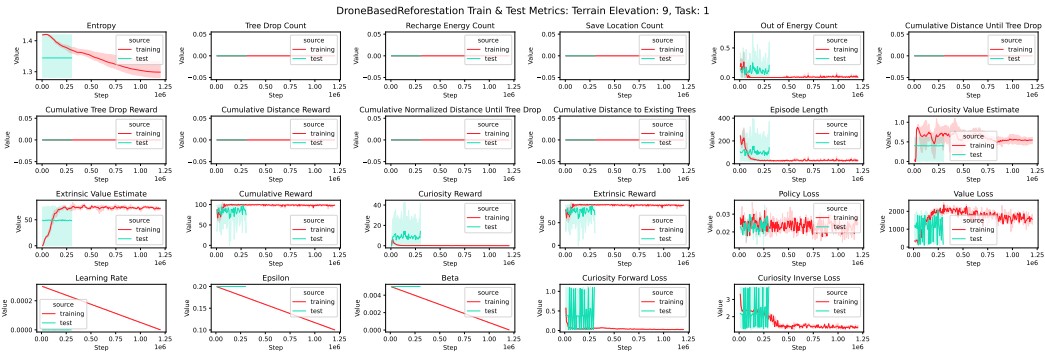

Figure 138: Drone-Based Reforestation: Train & Test Metrics: Terrain Elevation 9, Task 1.

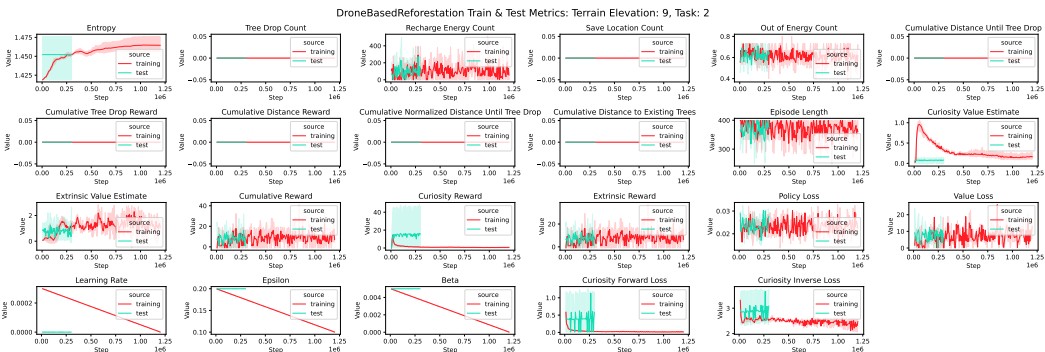

Figure 139: Drone-Based Reforestation: Train & Test Metrics: Terrain Elevation 9, Task 2.

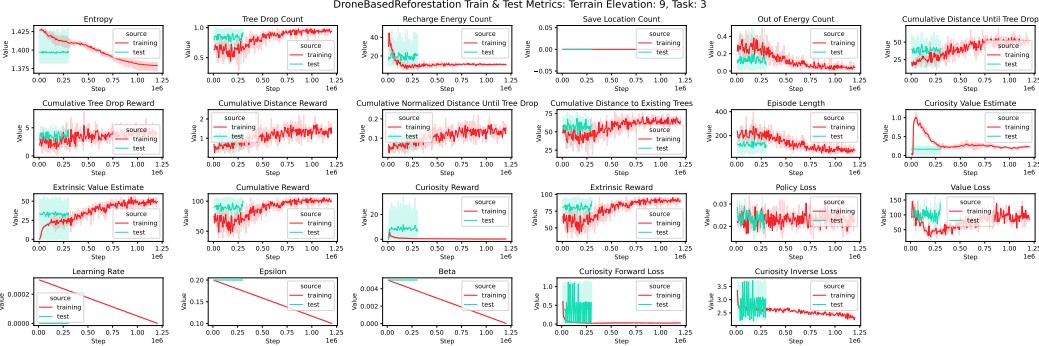

Figure 140: Drone-Based Reforestation: Train & Test Metrics: Terrain Elevation 9, Task 3.

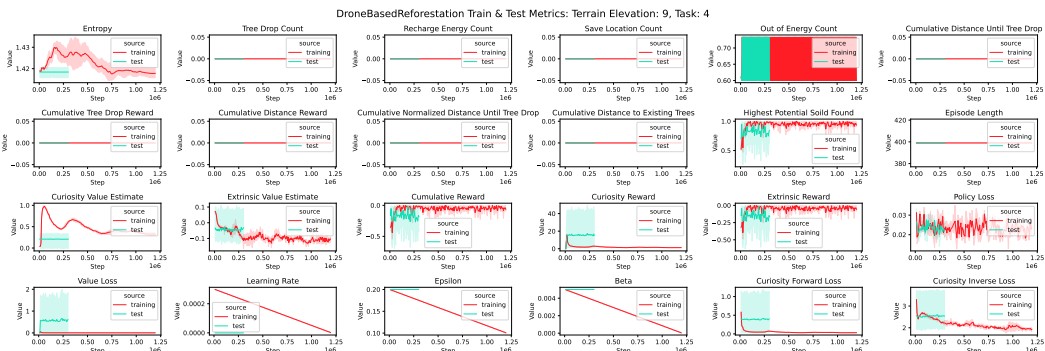

Figure 141: Drone-Based Reforestation: Train & Test Metrics: Terrain Elevation 9, Task 4.

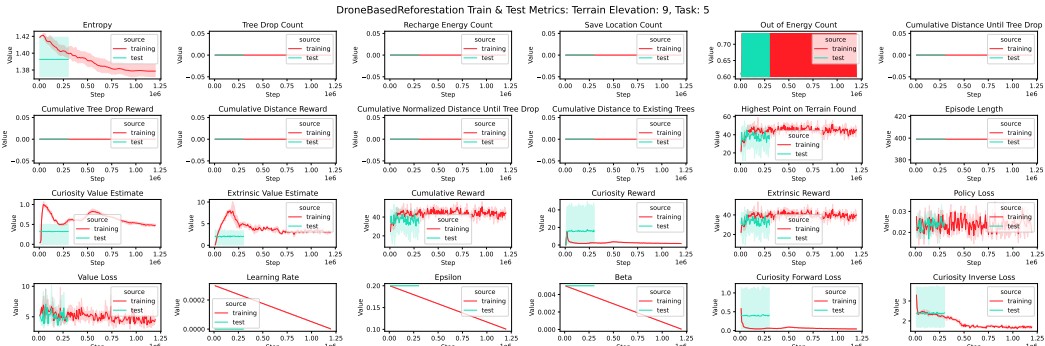

Figure 142: Drone-Based Reforestation: Train & Test Metrics: Terrain Elevation 9, Task 5.

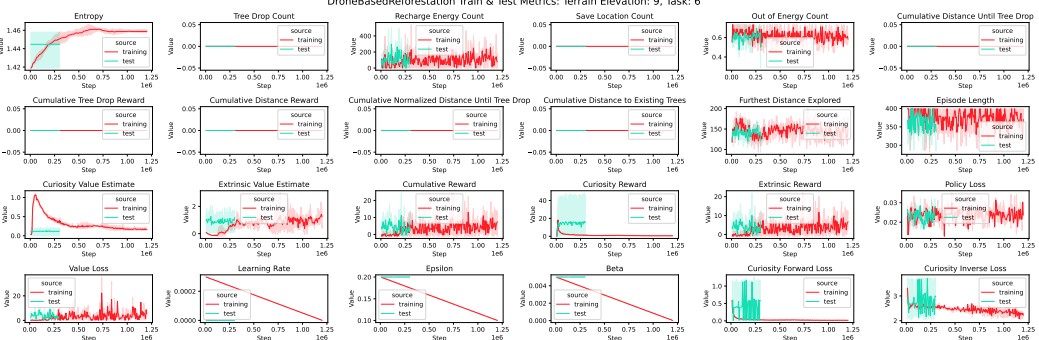

Figure 143: Drone-Based Reforestation: Train & Test Metrics: Terrain Elevation 9, Task 6.

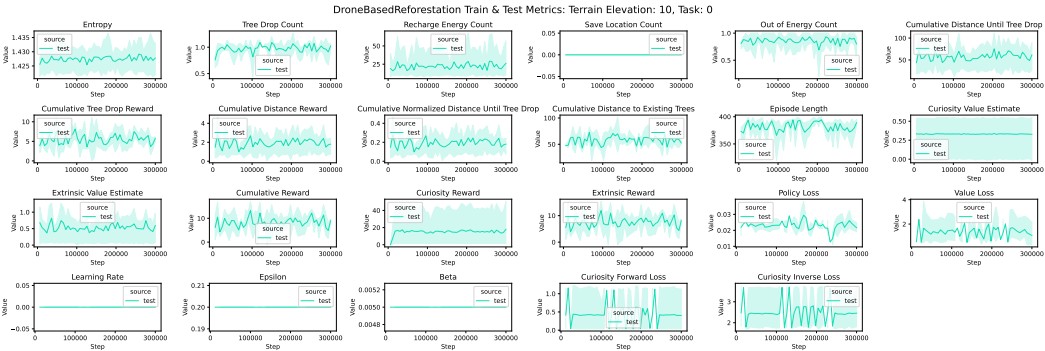

Figure 144: Drone-Based Reforestation: Train & Test Metrics: Terrain Elevation 10, Task 0.

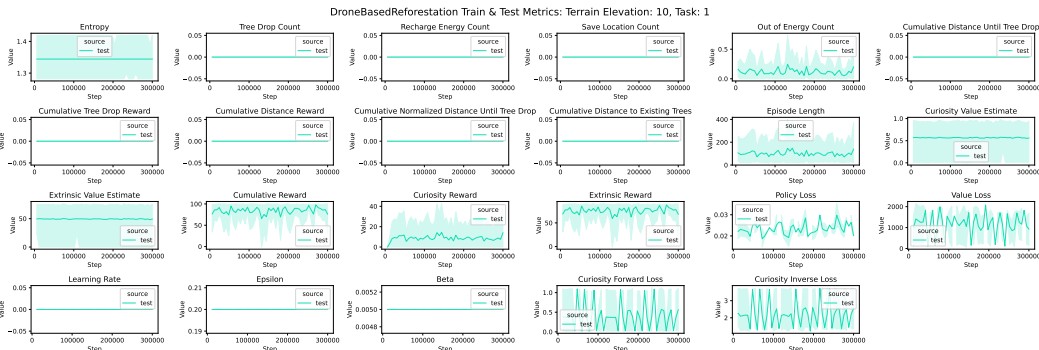

Figure 145: Drone-Based Reforestation: Train & Test Metrics: Terrain Elevation 10, Task 1.

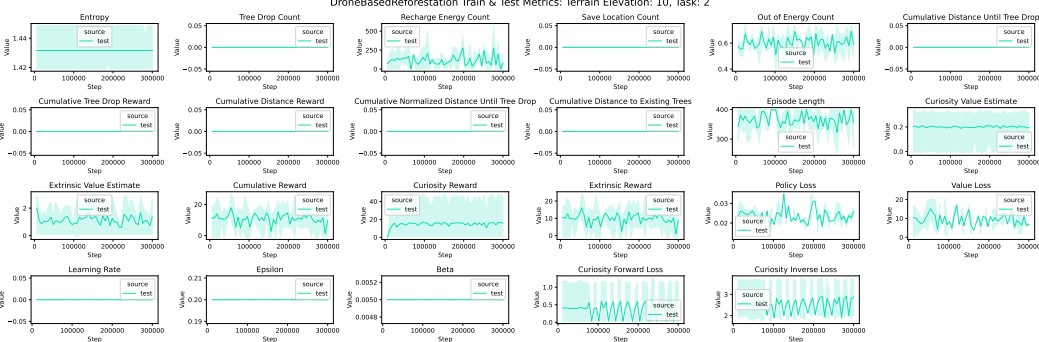

Figure 146: Drone-Based Reforestation: Train & Test Metrics: Terrain Elevation 10, Task 2.

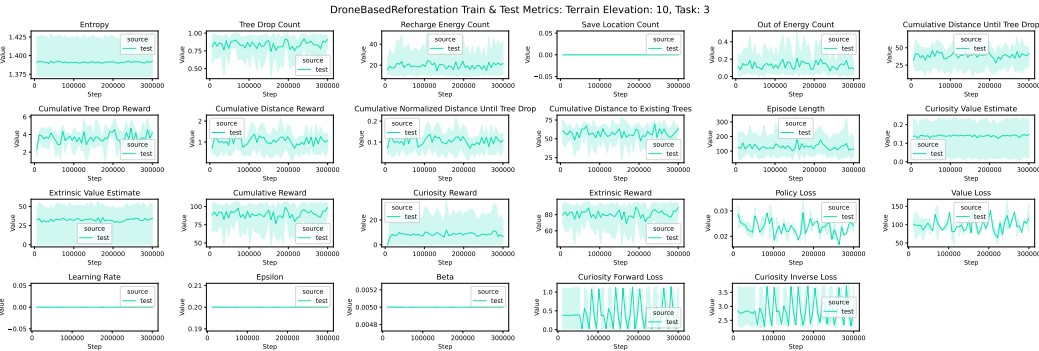

Figure 147: Drone-Based Reforestation: Train & Test Metrics: Terrain Elevation 10, Task 3.

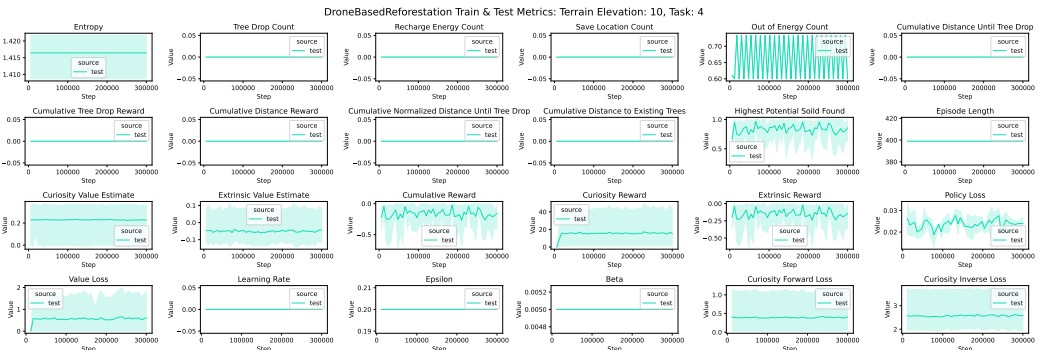

Figure 148: Drone-Based Reforestation: Train & Test Metrics: Terrain Elevation 10, Task 4.

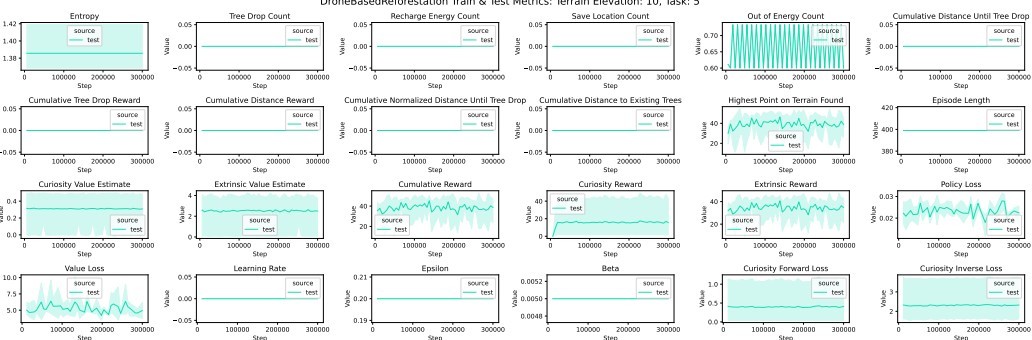

Figure 149: Drone-Based Reforestation: Train & Test Metrics: Terrain Elevation 10, Task 5.

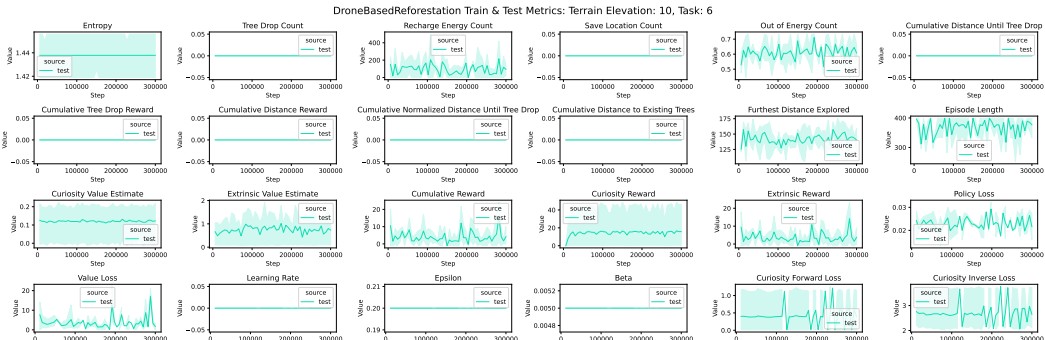

Figure 150: Drone-Based Reforestation: Train & Test Metrics: Terrain Elevation 10, Task 6.

## A.10.8 DRONE-BASED REFORESTATION: AVERAGE TEST METRIC - TASK VS PATTERN

Figure 151: Drone-Based Reforestation: Average Train & Test Metrics.

### A.10.9 AERIAL WILDFIRE SUPPRESSION: TRAIN & TEST METRICS

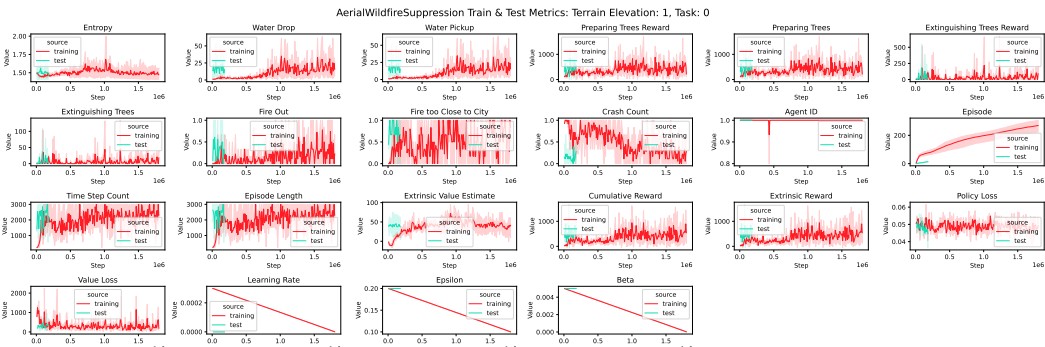

Figure 152: Aerial Wildfire Suppression: Train & Test Metrics: Terrain Elevation 1, Task 0.

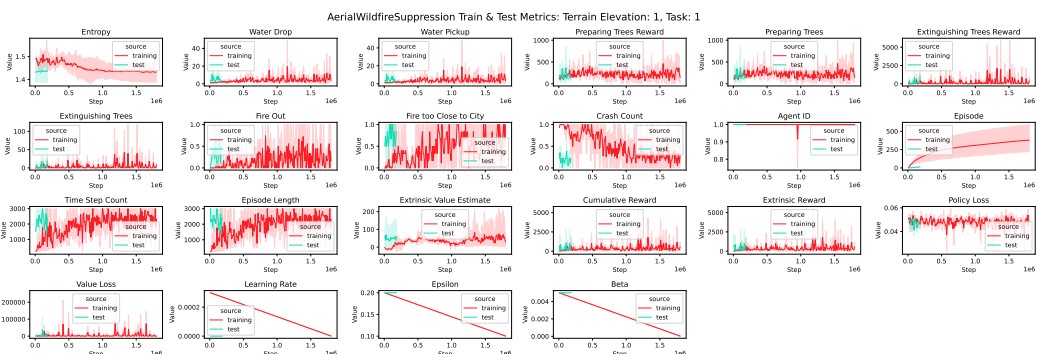

Figure 153: Aerial Wildfire Suppression: Train & Test Metrics: Terrain Elevation 1, Task 1.

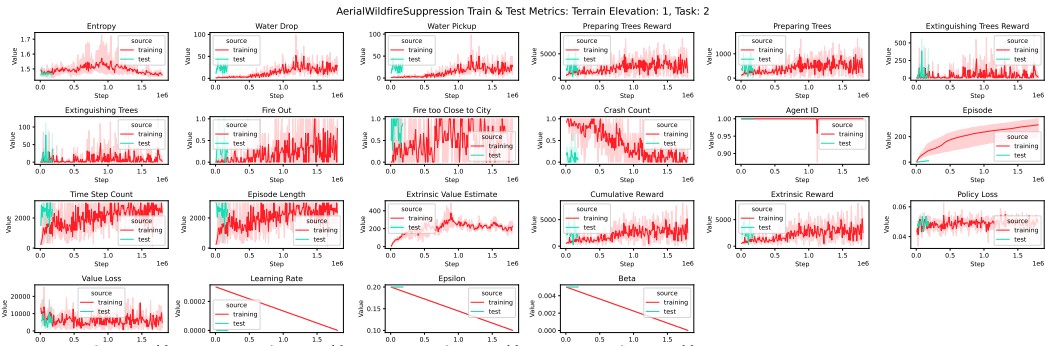

Figure 154: Aerial Wildfire Suppression: Train & Test Metrics: Terrain Elevation 1, Task 2.

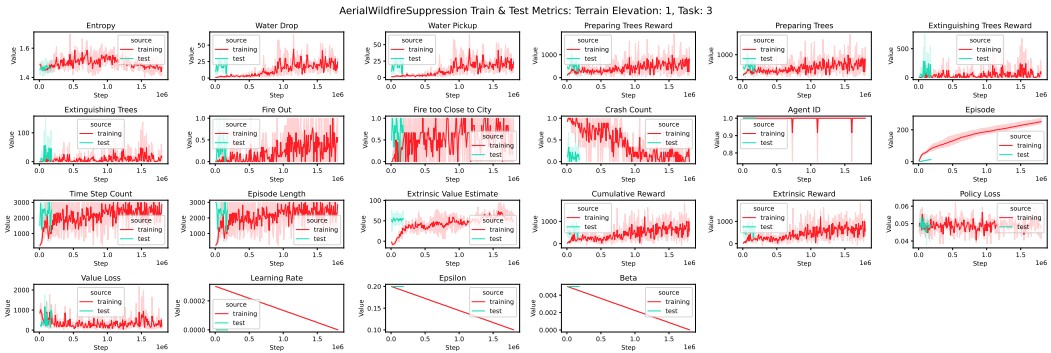

Figure 155: Aerial Wildfire Suppression: Train & Test Metrics: Terrain Elevation 1, Task 3.

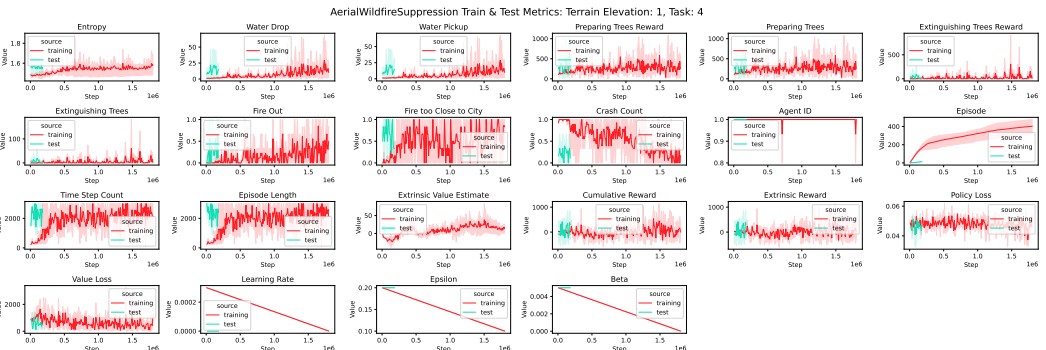

Figure 156: Aerial Wildfire Suppression: Train & Test Metrics: Terrain Elevation 1, Task 4.

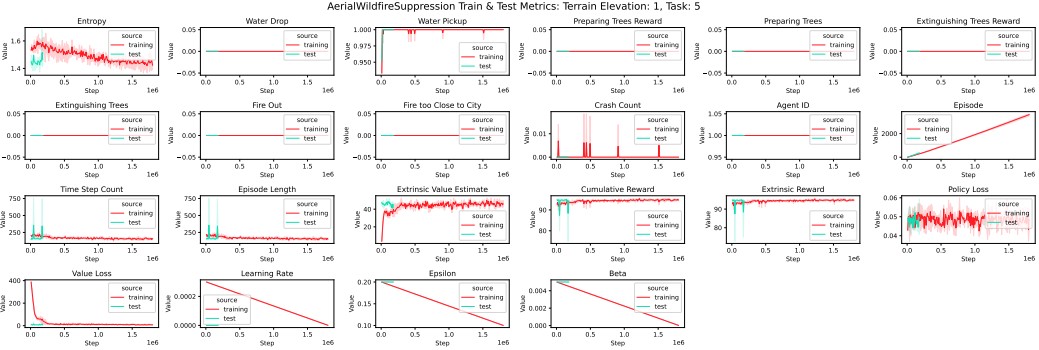

Figure 157: Aerial Wildfire Suppression: Train & Test Metrics: Terrain Elevation 1, Task 5.

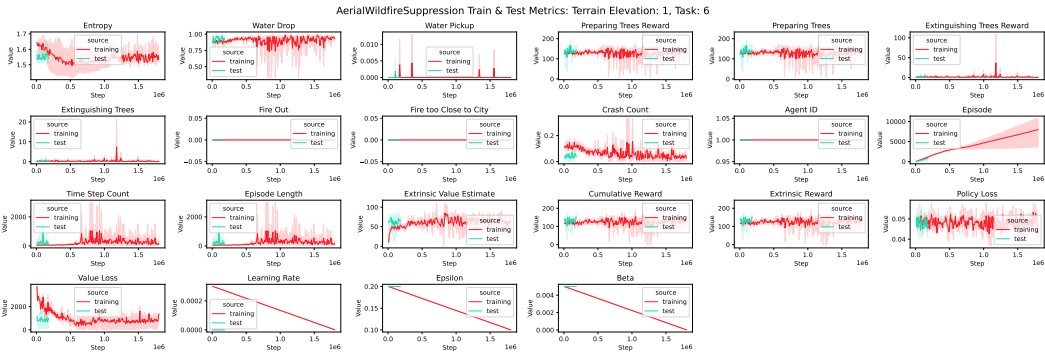

Figure 158: Aerial Wildfire Suppression: Train & Test Metrics: Terrain Elevation 1, Task 6.

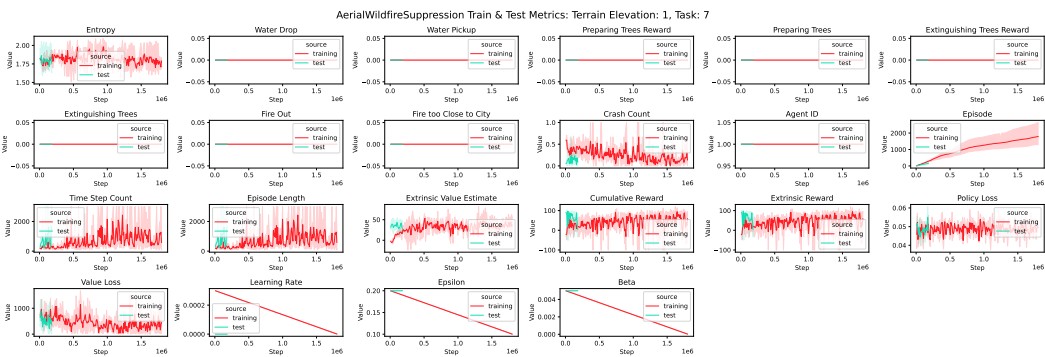

Figure 159: Aerial Wildfire Suppression: Train & Test Metrics: Terrain Elevation 1, Task 7.

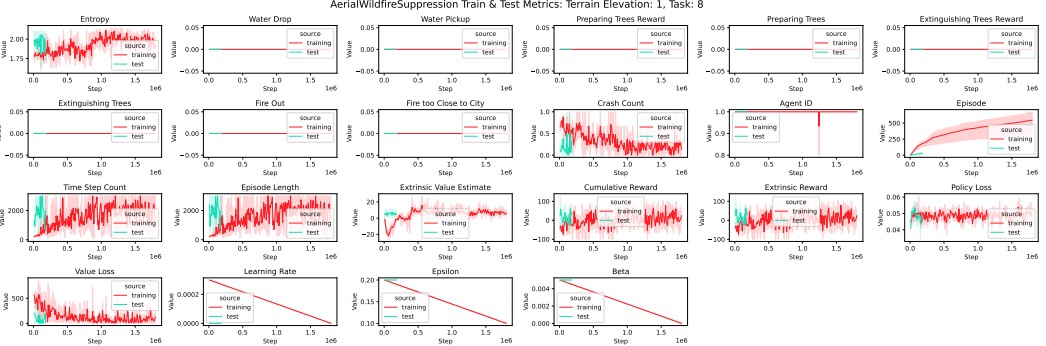

Figure 160: Aerial Wildfire Suppression: Train & Test Metrics: Terrain Elevation 1, Task 8.

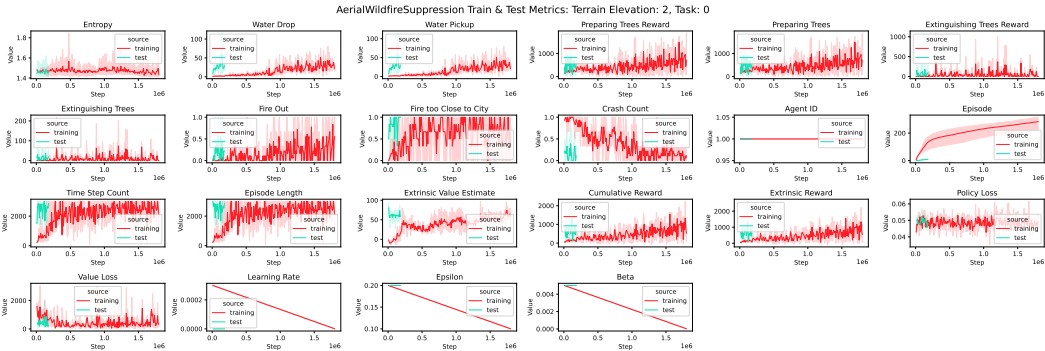

Figure 161: Aerial Wildfire Suppression: Train & Test Metrics: Terrain Elevation 2, Task 0.

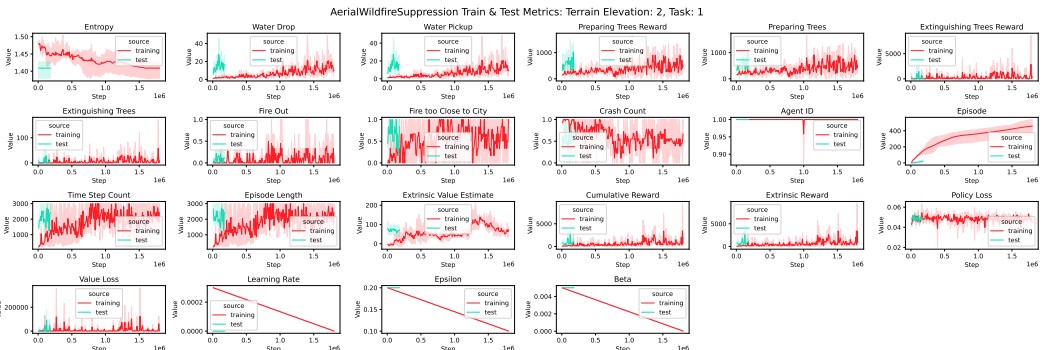

Figure 162: Aerial Wildfire Suppression: Train & Test Metrics: Terrain Elevation 2, Task 1.

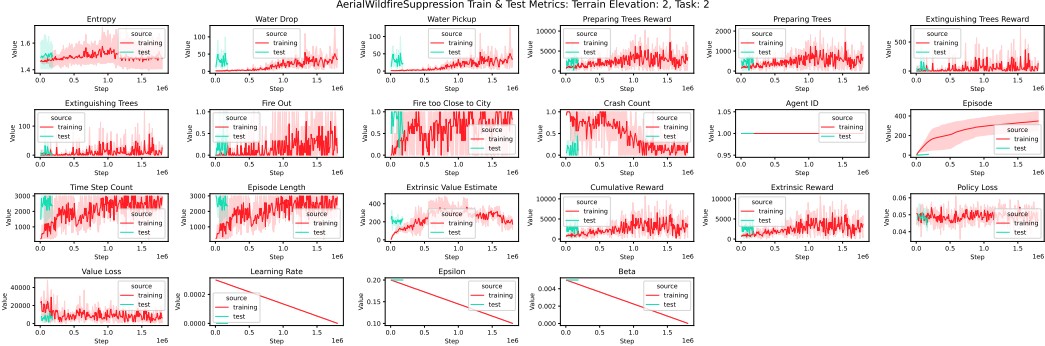

Figure 163: Aerial Wildfire Suppression: Train & Test Metrics: Terrain Elevation 2, Task 2.

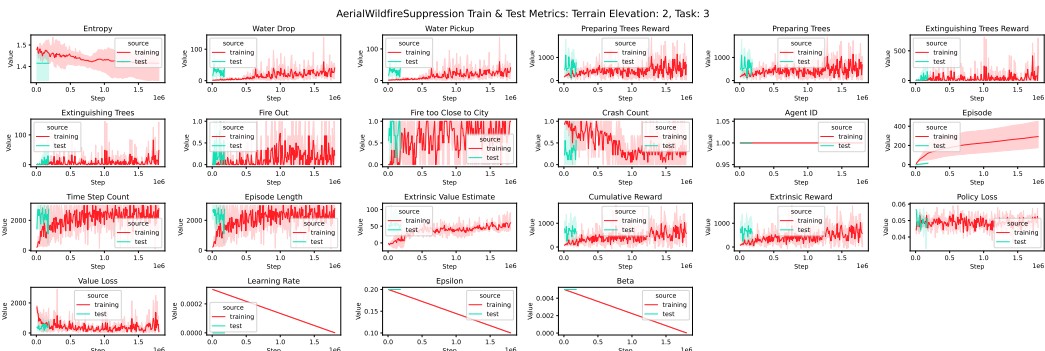

Figure 164: Aerial Wildfire Suppression: Train & Test Metrics: Terrain Elevation 2, Task 3.

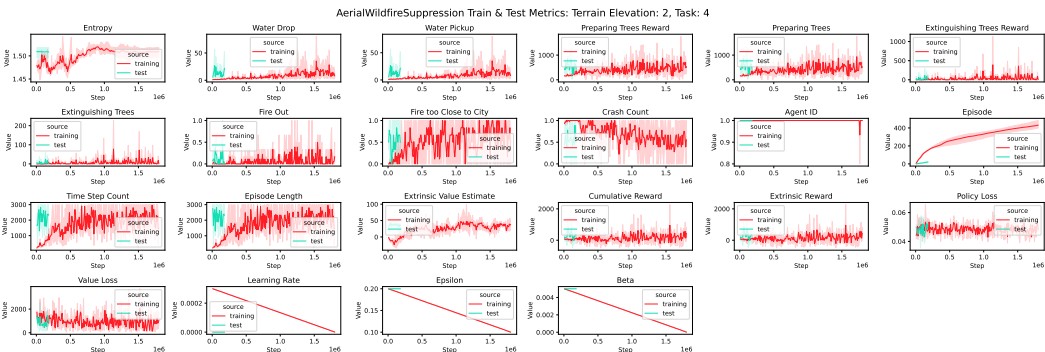

Figure 165: Aerial Wildfire Suppression: Train & Test Metrics: Terrain Elevation 2, Task 4.

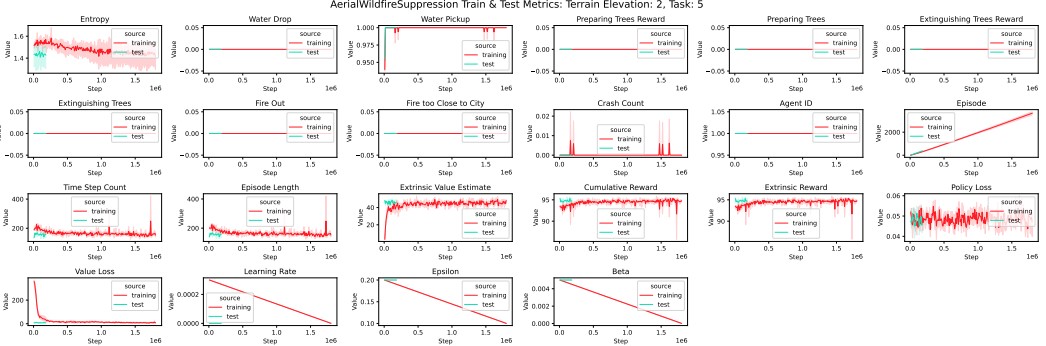

Figure 166: Aerial Wildfire Suppression: Train & Test Metrics: Terrain Elevation 2, Task 5.

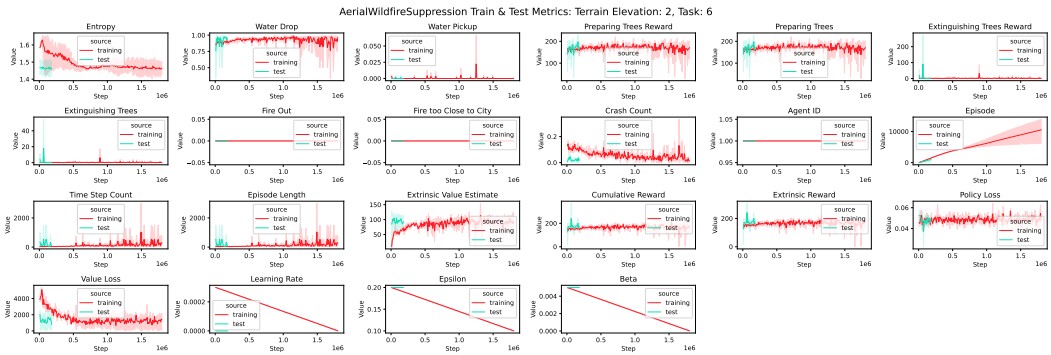

Figure 167: Aerial Wildfire Suppression: Train & Test Metrics: Terrain Elevation 2, Task 6.

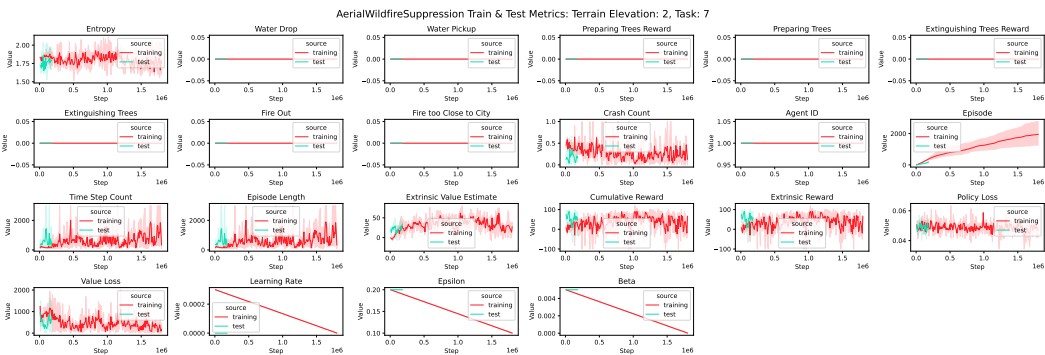

Figure 168: Aerial Wildfire Suppression: Train & Test Metrics: Terrain Elevation 2, Task 7.

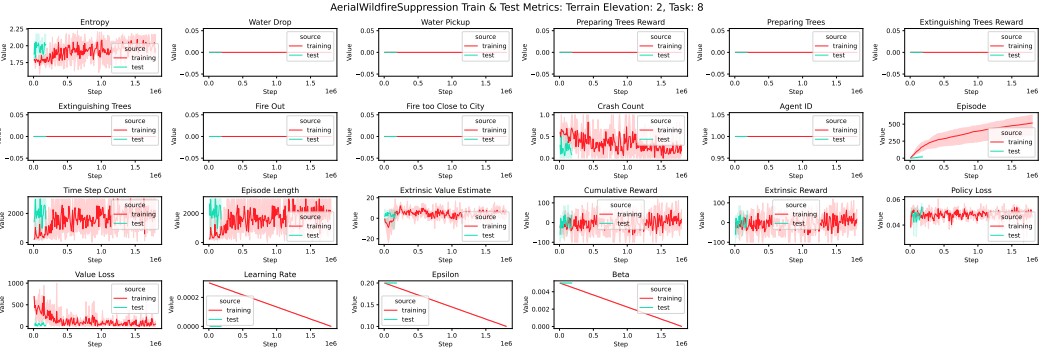

Figure 169: Aerial Wildfire Suppression: Train & Test Metrics: Terrain Elevation 2, Task 8.

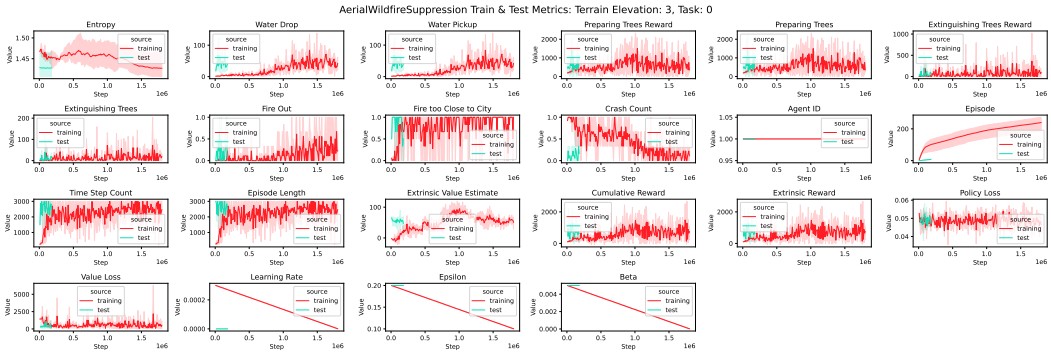

Figure 170: Aerial Wildfire Suppression: Train & Test Metrics: Terrain Elevation 3, Task 0.

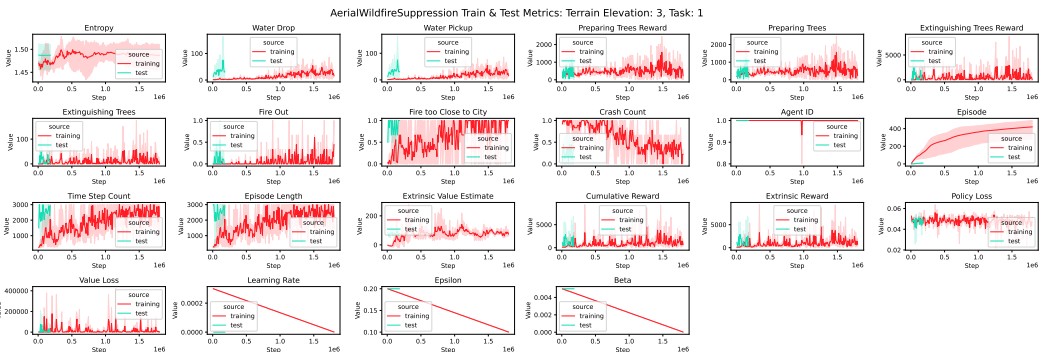

Figure 171: Aerial Wildfire Suppression: Train & Test Metrics: Terrain Elevation 3, Task 1.

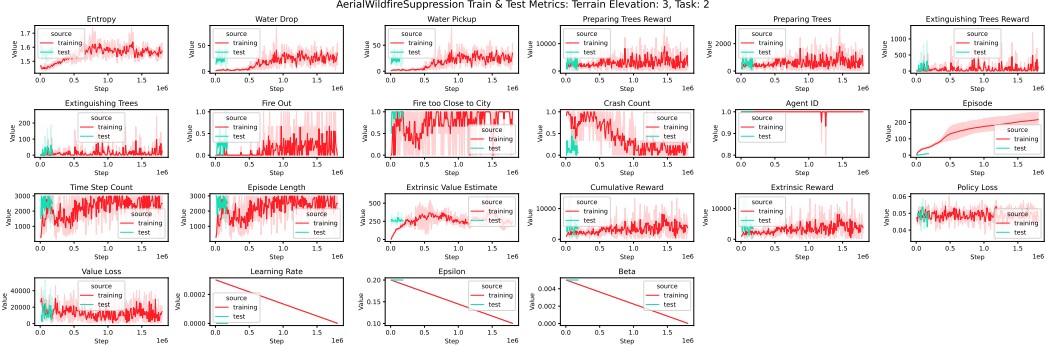

Figure 172: Aerial Wildfire Suppression: Train & Test Metrics: Terrain Elevation 3, Task 2.

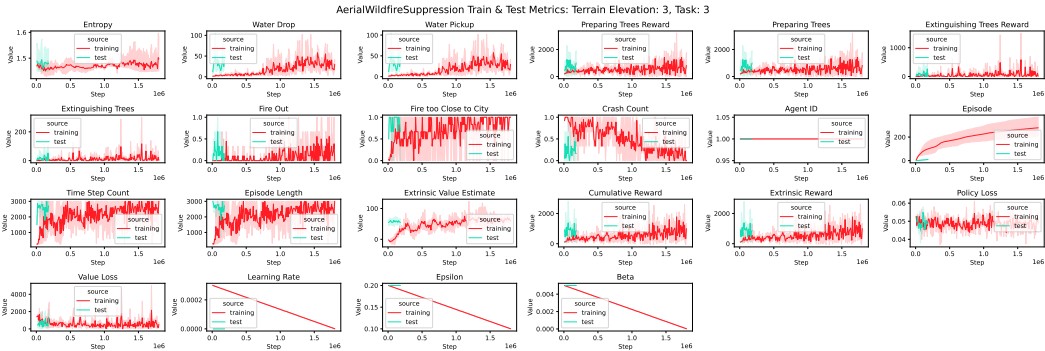

Figure 173: Aerial Wildfire Suppression: Train & Test Metrics: Terrain Elevation 3, Task 3.

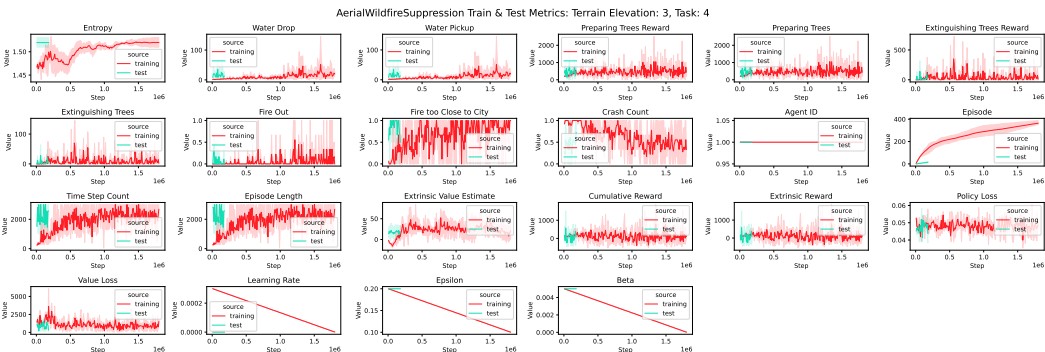

Figure 174: Aerial Wildfire Suppression: Train & Test Metrics: Terrain Elevation 3, Task 4.

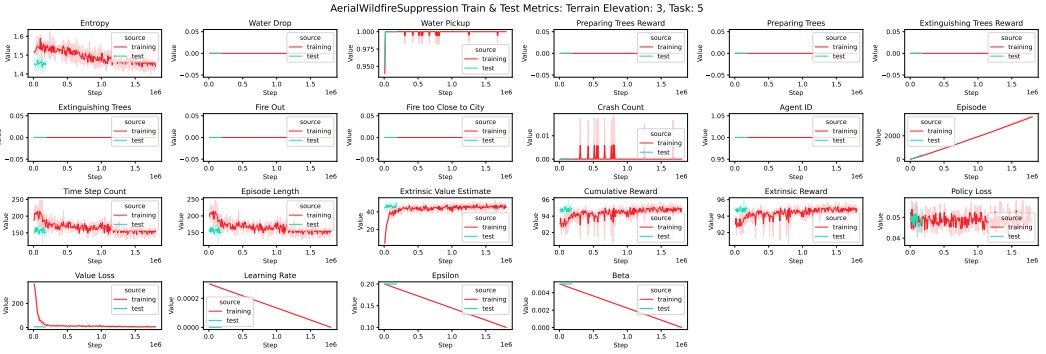

Figure 175: Aerial Wildfire Suppression: Train & Test Metrics: Terrain Elevation 3, Task 5.

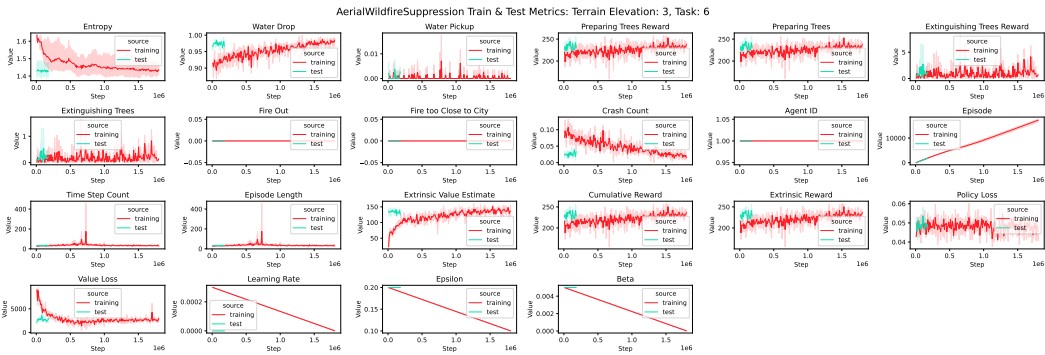

Figure 176: Aerial Wildfire Suppression: Train & Test Metrics: Terrain Elevation 3, Task 6.

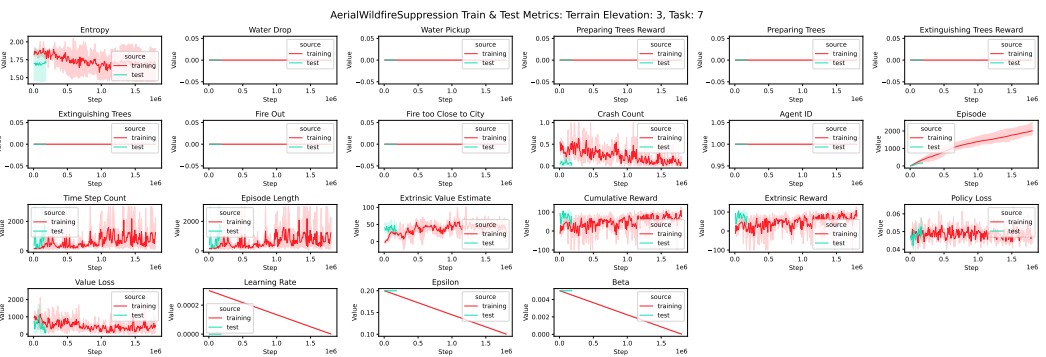

Figure 177: Aerial Wildfire Suppression: Train & Test Metrics: Terrain Elevation 3, Task 7.

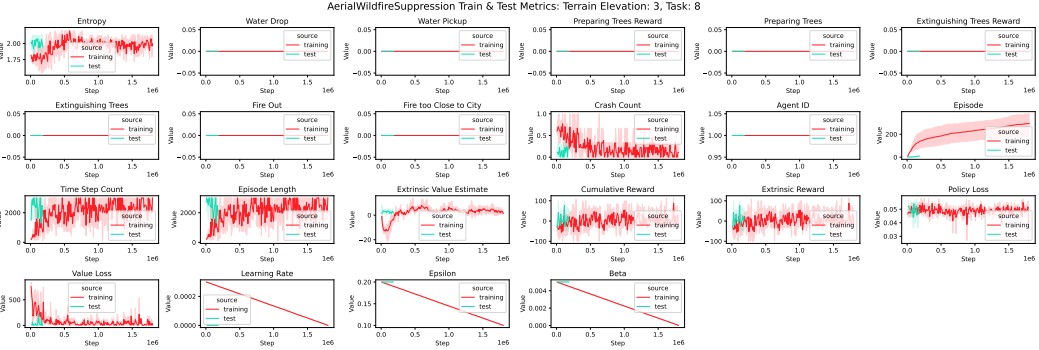

Figure 178: Aerial Wildfire Suppression: Train & Test Metrics: Terrain Elevation 3, Task 8.

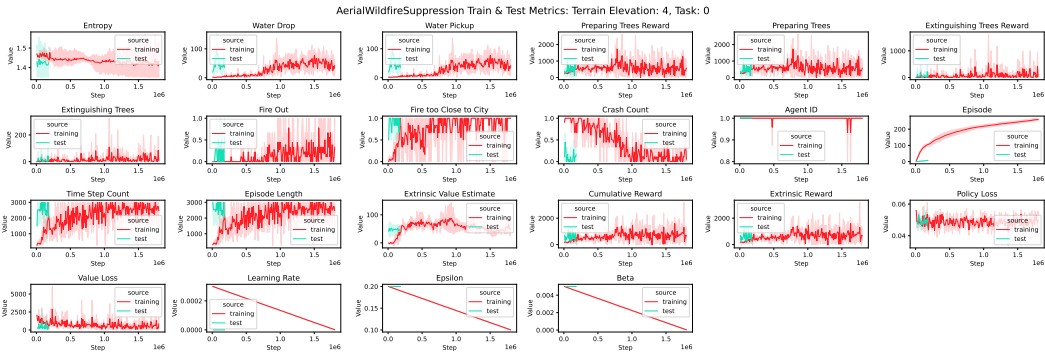

Figure 179: Aerial Wildfire Suppression: Train & Test Metrics: Terrain Elevation 4, Task 0.

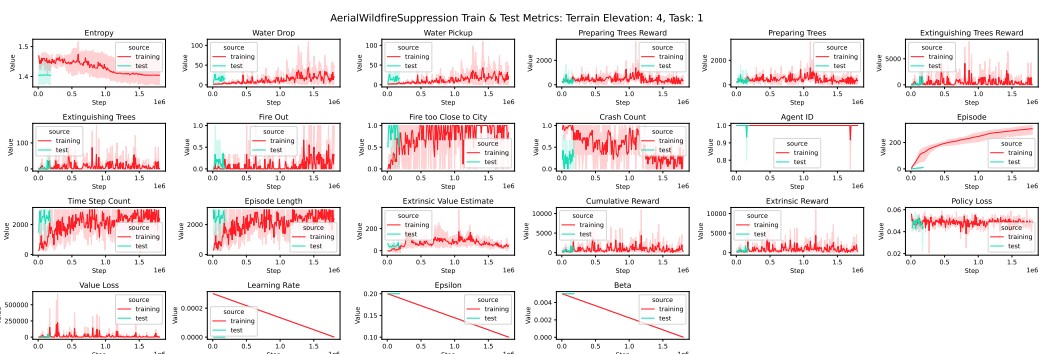

Figure 180: Aerial Wildfire Suppression: Train & Test Metrics: Terrain Elevation 4, Task 1.

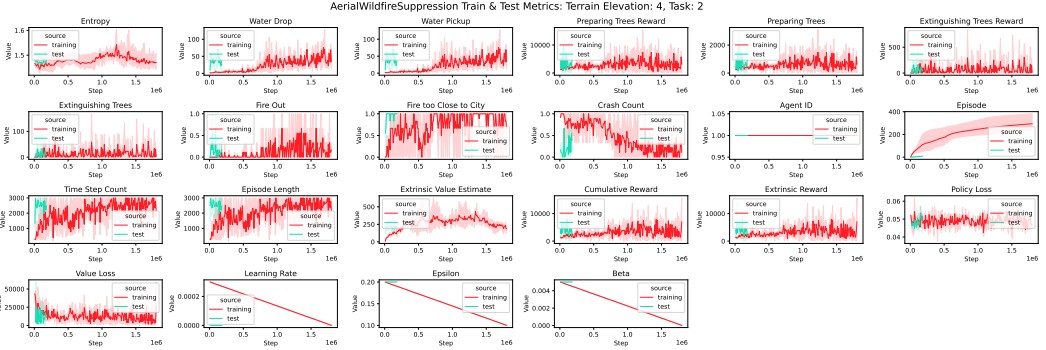

Figure 181: Aerial Wildfire Suppression: Train & Test Metrics: Terrain Elevation 4, Task 2.

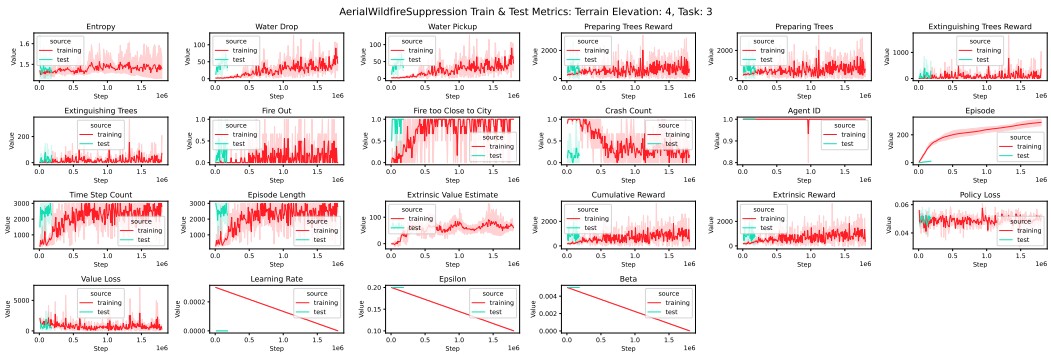

Figure 182: Aerial Wildfire Suppression: Train & Test Metrics: Terrain Elevation 4, Task 3.

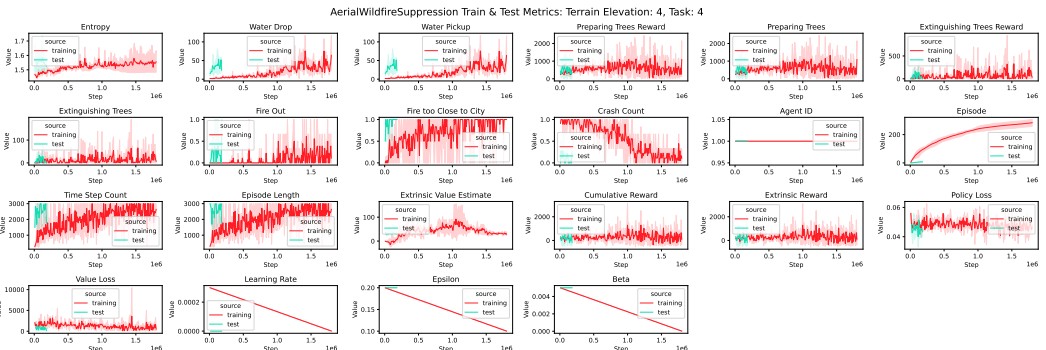

Figure 183: Aerial Wildfire Suppression: Train & Test Metrics: Terrain Elevation 4, Task 4.

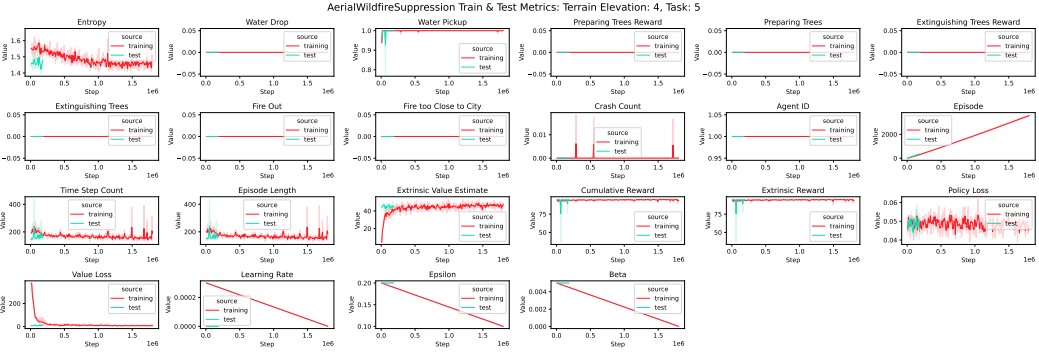

Figure 184: Aerial Wildfire Suppression: Train & Test Metrics: Terrain Elevation 4, Task 5.

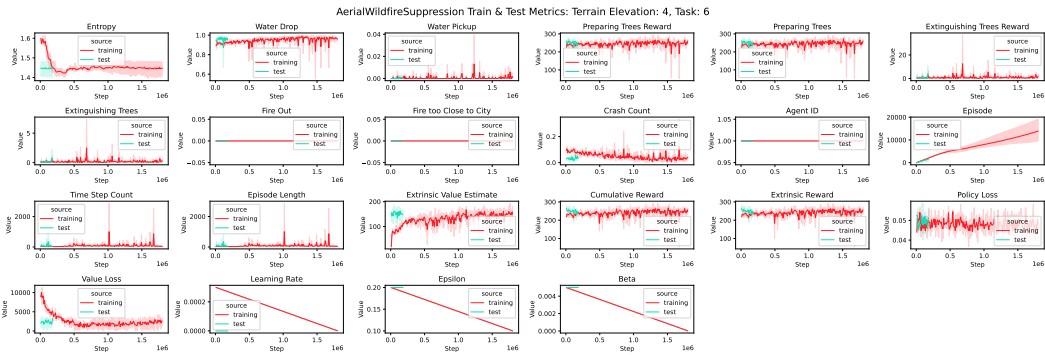

Figure 185: Aerial Wildfire Suppression: Train & Test Metrics: Terrain Elevation 4, Task 6.

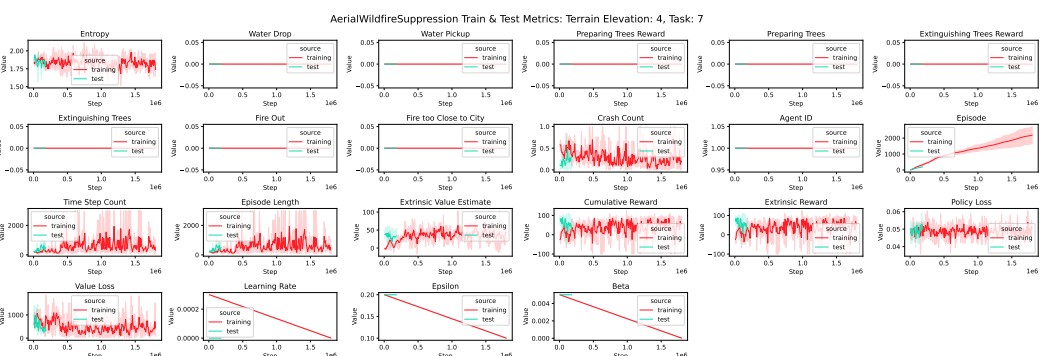

Figure 186: Aerial Wildfire Suppression: Train & Test Metrics: Terrain Elevation 4, Task 7.

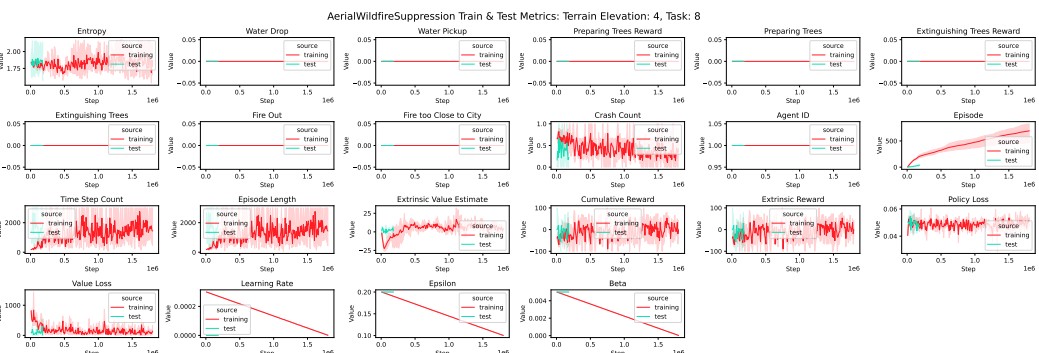

Figure 187: Aerial Wildfire Suppression: Train & Test Metrics: Terrain Elevation 4, Task 8.

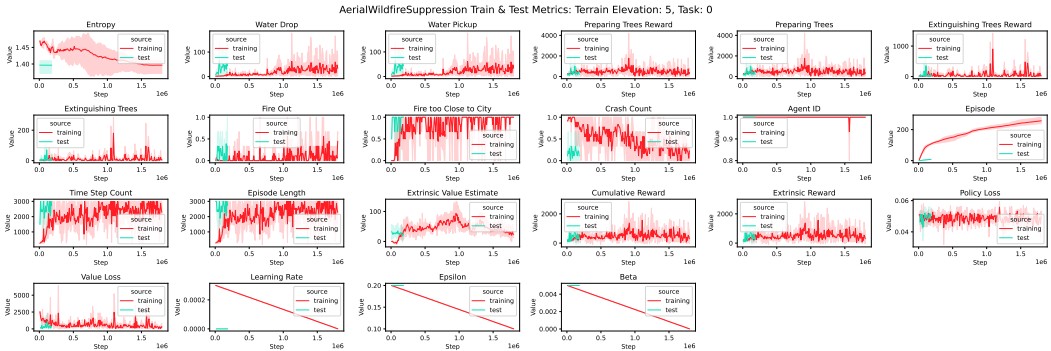

Figure 188: Aerial Wildfire Suppression: Train & Test Metrics: Terrain Elevation 5, Task 0.

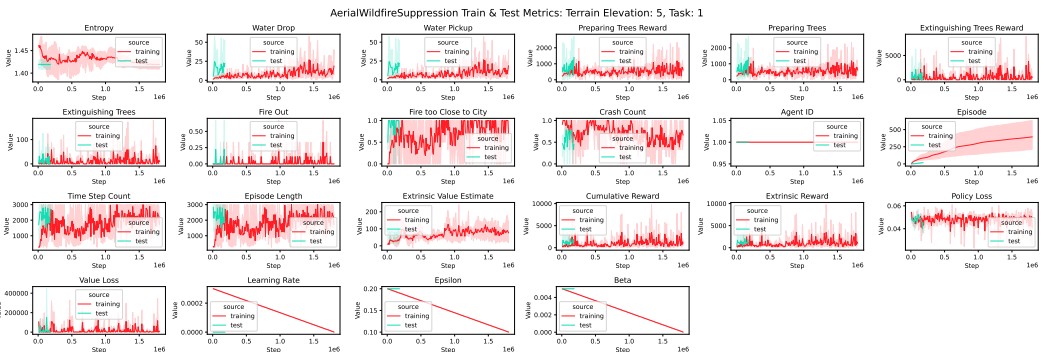

Figure 189: Aerial Wildfire Suppression: Train & Test Metrics: Terrain Elevation 5, Task 1.

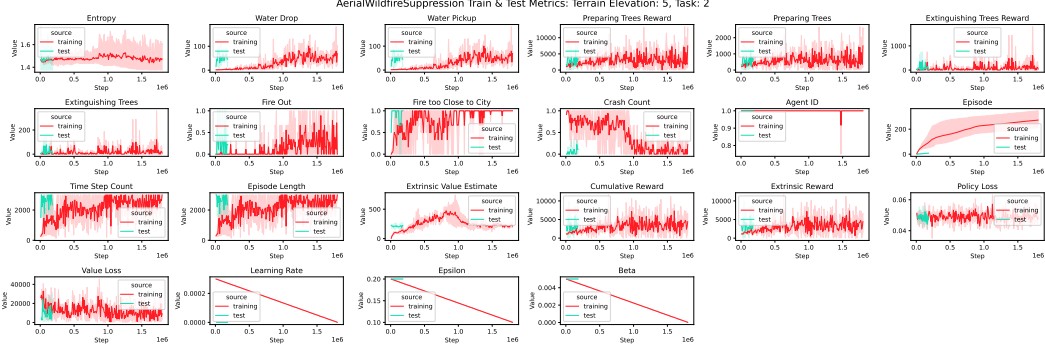

Figure 190: Aerial Wildfire Suppression: Train & Test Metrics: Terrain Elevation 5, Task 2.

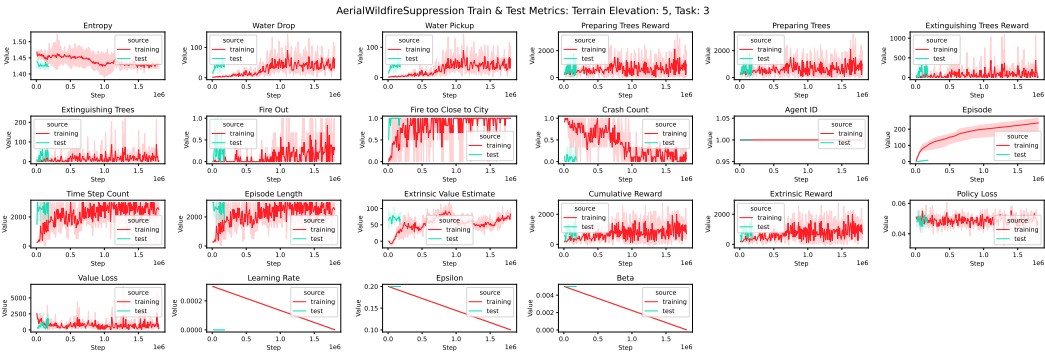

Figure 191: Aerial Wildfire Suppression: Train & Test Metrics: Terrain Elevation 5, Task 3.

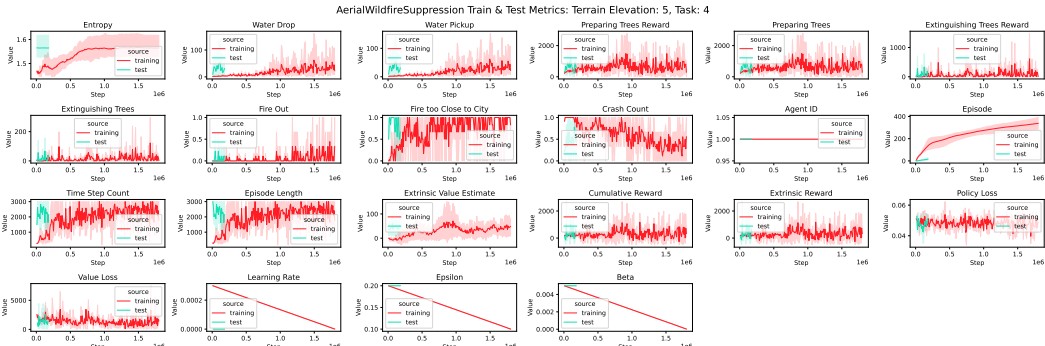

Figure 192: Aerial Wildfire Suppression: Train & Test Metrics: Terrain Elevation 5, Task 4.

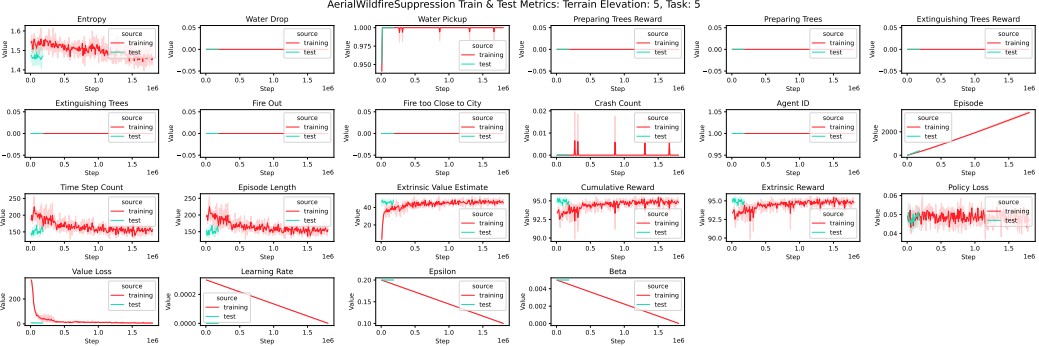

Figure 193: Aerial Wildfire Suppression: Train & Test Metrics: Terrain Elevation 5, Task 5.

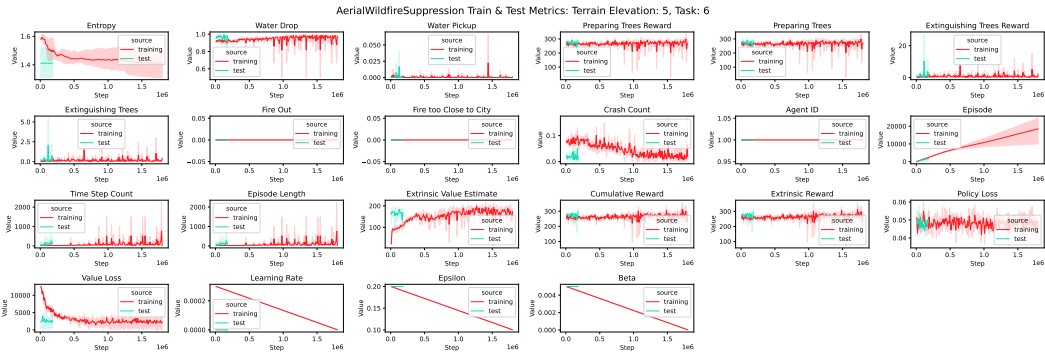

Figure 194: Aerial Wildfire Suppression: Train & Test Metrics: Terrain Elevation 5, Task 6.

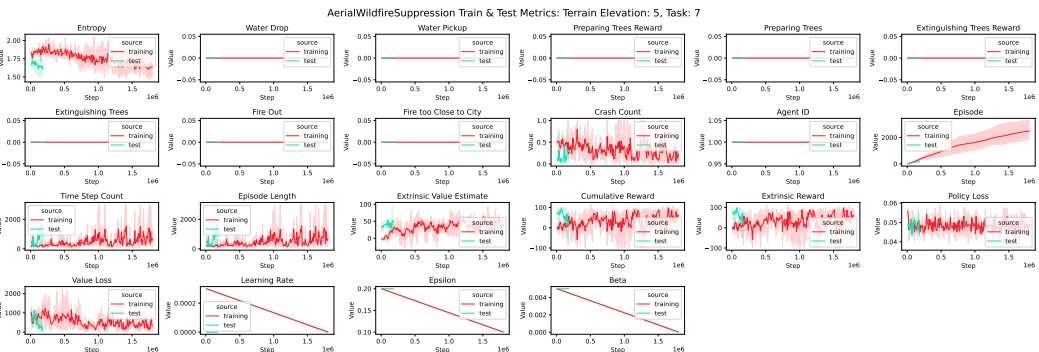

Figure 195: Aerial Wildfire Suppression: Train & Test Metrics: Terrain Elevation 5, Task 7.

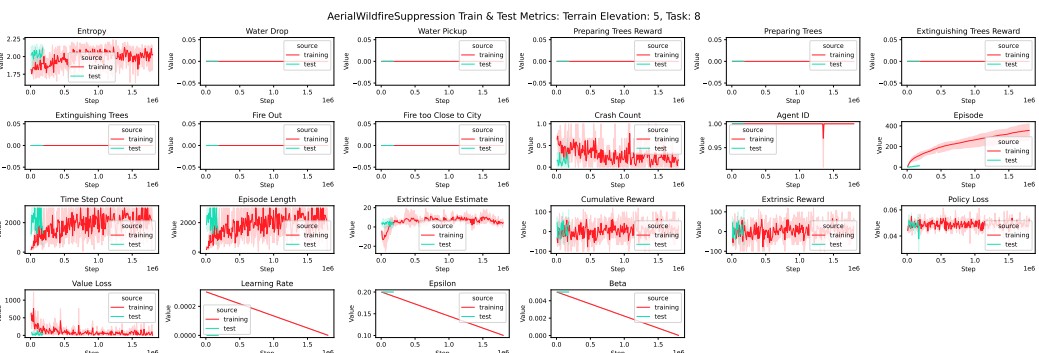

Figure 196: Aerial Wildfire Suppression: Train & Test Metrics: Terrain Elevation 5, Task 8.

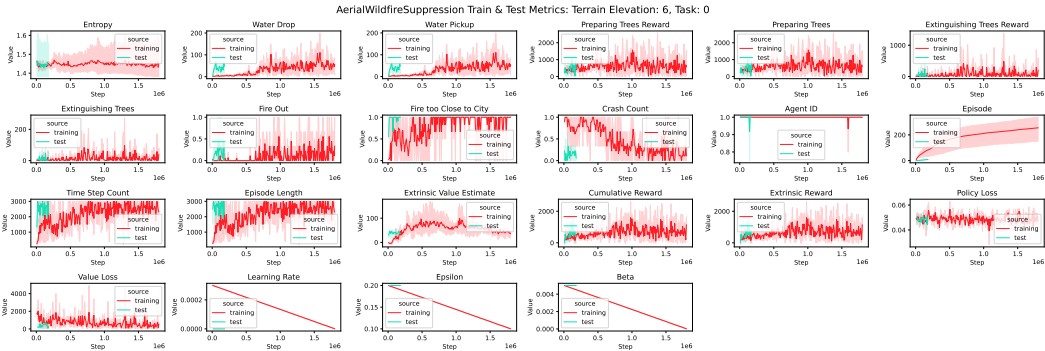

Figure 197: Aerial Wildfire Suppression: Train & Test Metrics: Terrain Elevation 6, Task 0.

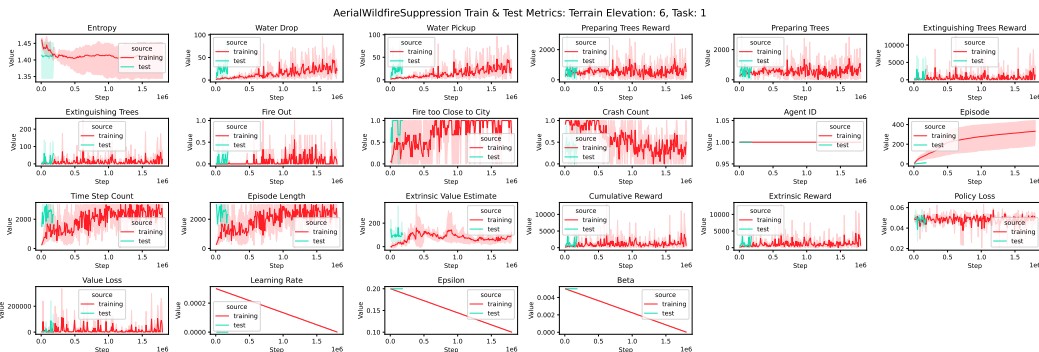

Figure 198: Aerial Wildfire Suppression: Train & Test Metrics: Terrain Elevation 6, Task 1.

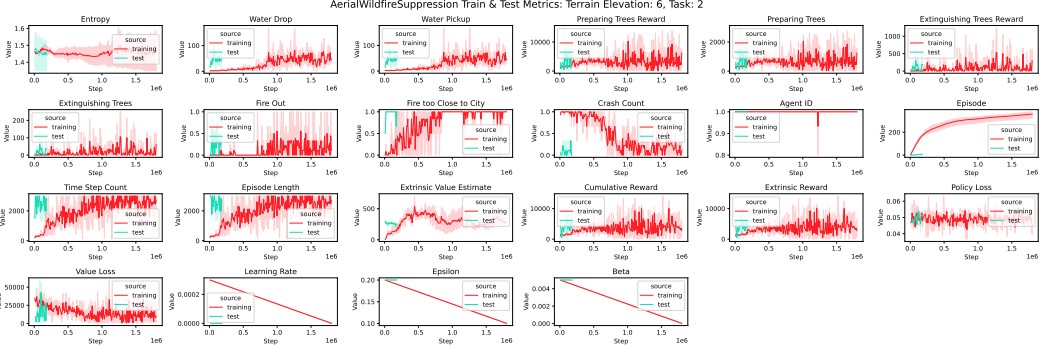

Figure 199: Aerial Wildfire Suppression: Train & Test Metrics: Terrain Elevation 6, Task 2.

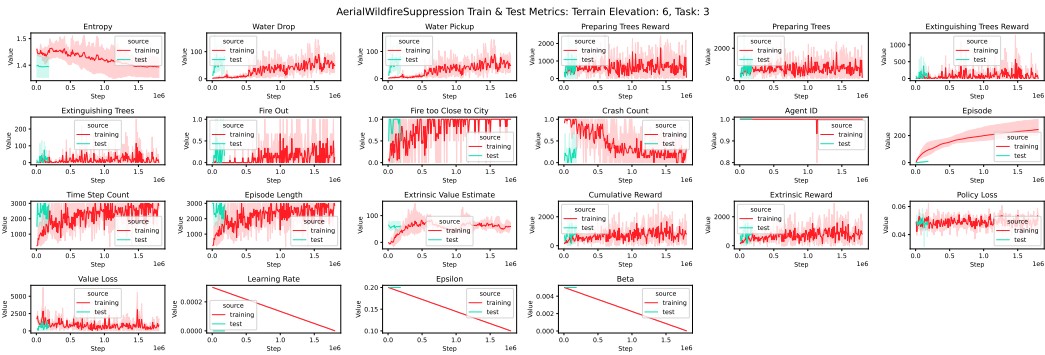

Figure 200: Aerial Wildfire Suppression: Train & Test Metrics: Terrain Elevation 6, Task 3.

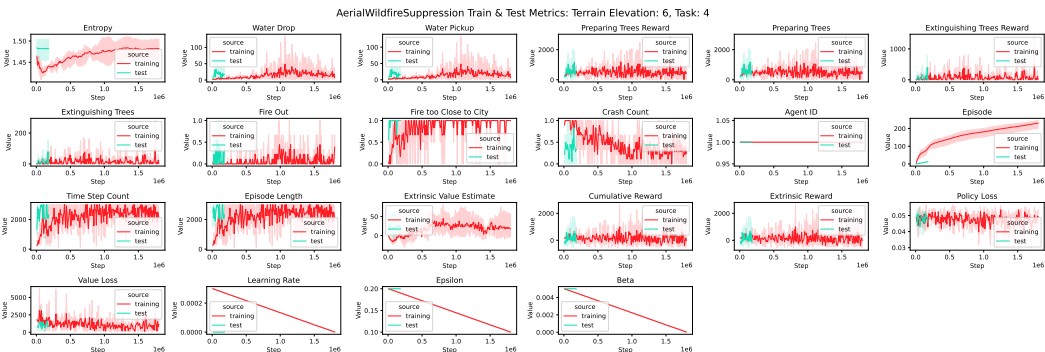

Figure 201: Aerial Wildfire Suppression: Train & Test Metrics: Terrain Elevation 6, Task 4.

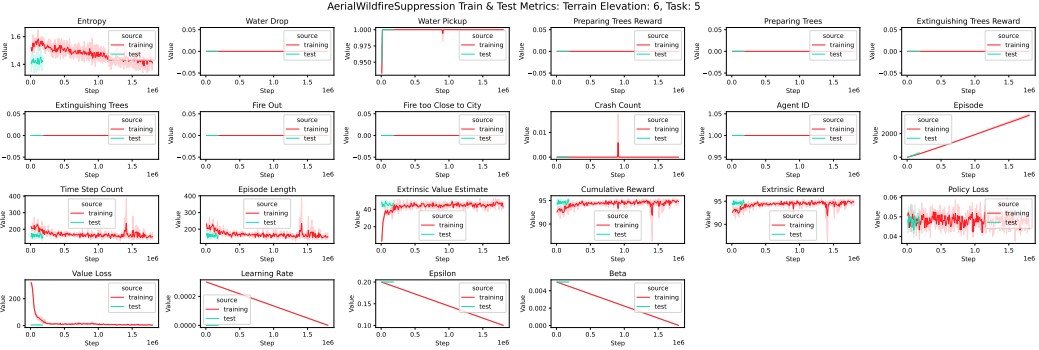

Figure 202: Aerial Wildfire Suppression: Train & Test Metrics: Terrain Elevation 6, Task 5.

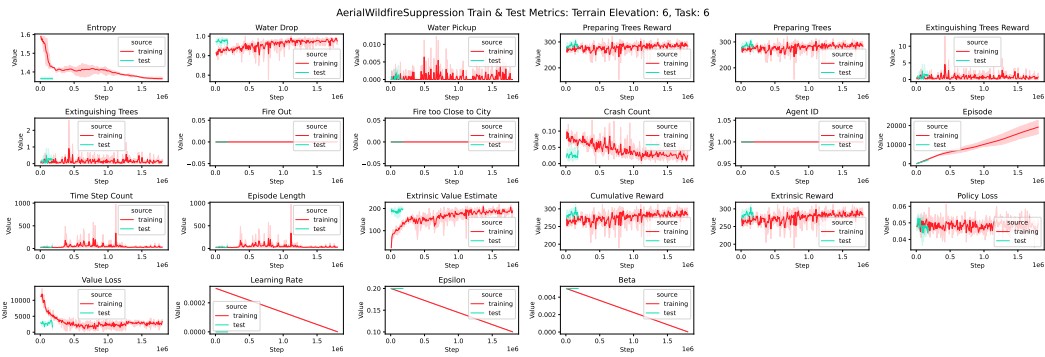

Figure 203: Aerial Wildfire Suppression: Train & Test Metrics: Terrain Elevation 6, Task 6.

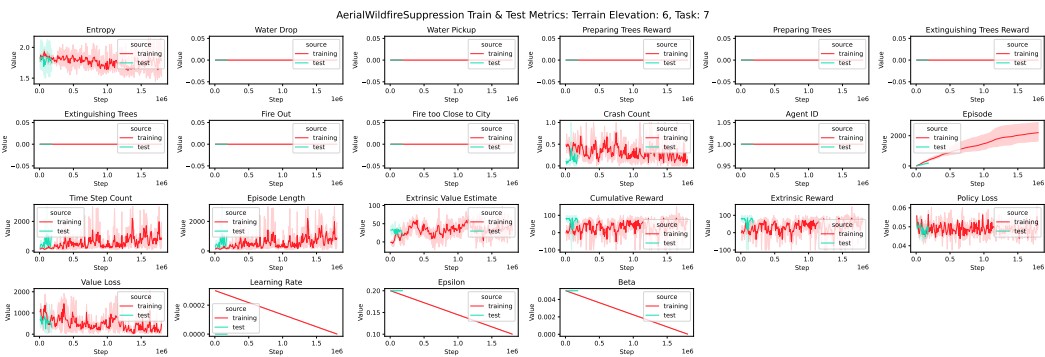

Figure 204: Aerial Wildfire Suppression: Train & Test Metrics: Terrain Elevation 6, Task 7.

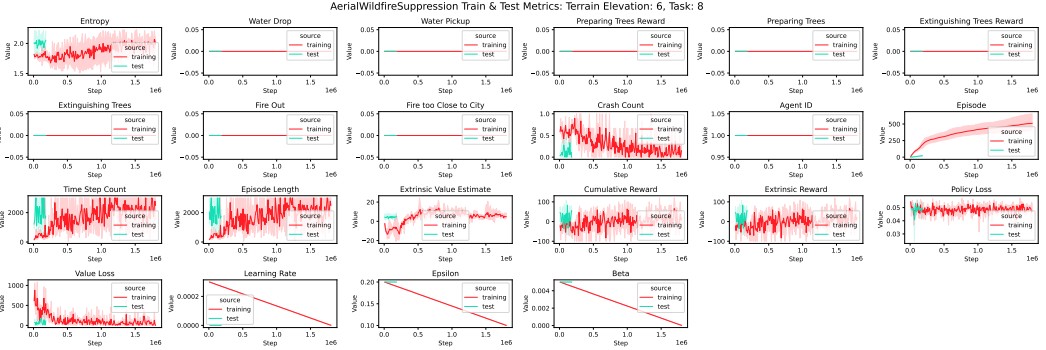

Figure 205: Aerial Wildfire Suppression: Train & Test Metrics: Terrain Elevation 6, Task 8.

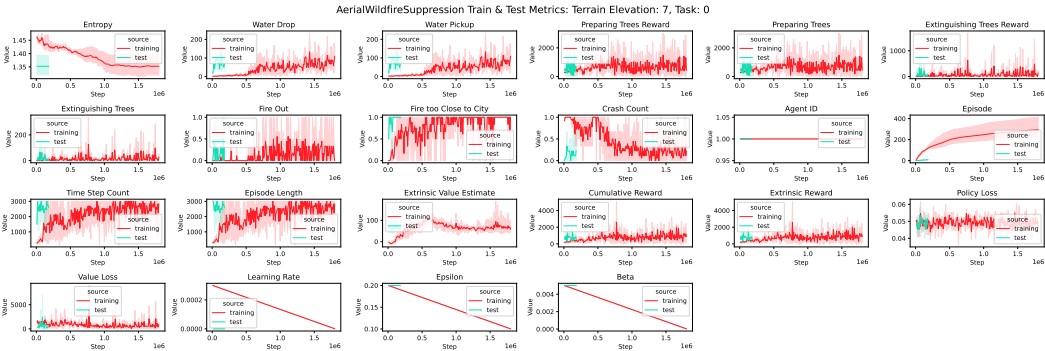

Figure 206: Aerial Wildfire Suppression: Train & Test Metrics: Terrain Elevation 7, Task 0.

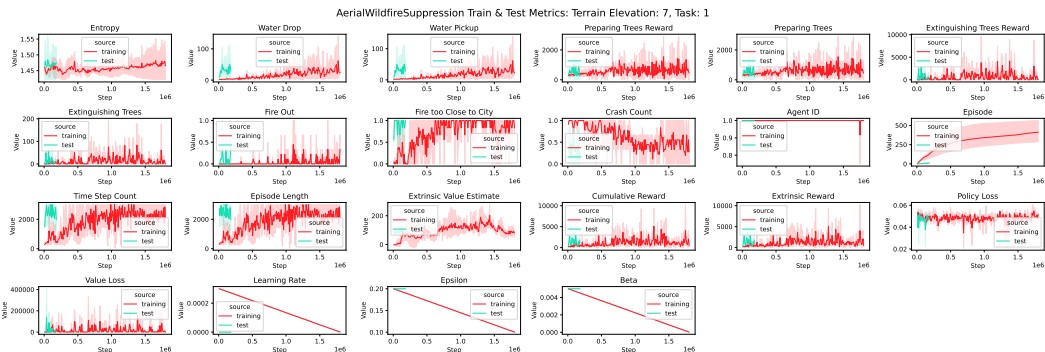

Figure 207: Aerial Wildfire Suppression: Train & Test Metrics: Terrain Elevation 7, Task 1.

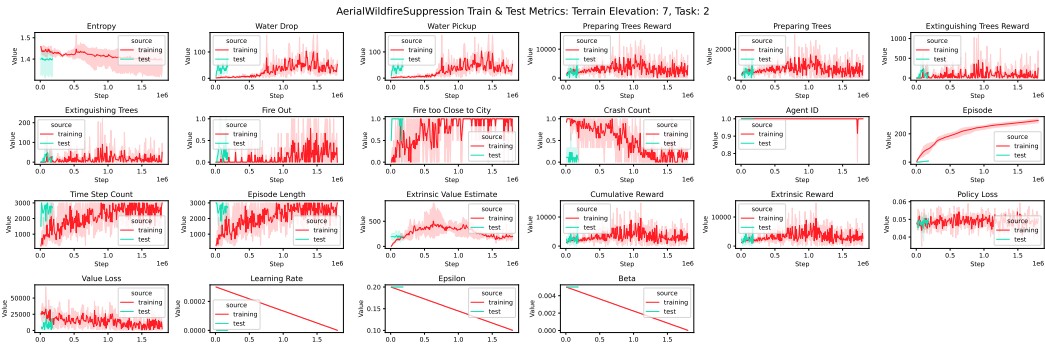

Figure 208: Aerial Wildfire Suppression: Train & Test Metrics: Terrain Elevation 7, Task 2.

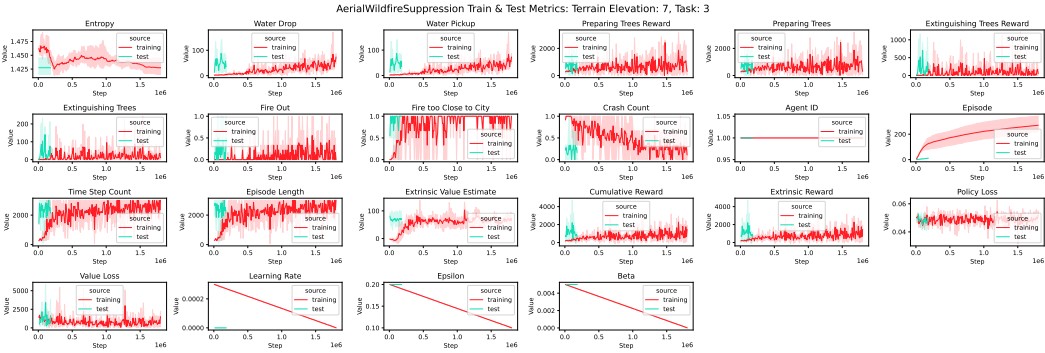

Figure 209: Aerial Wildfire Suppression: Train & Test Metrics: Terrain Elevation 7, Task 3.

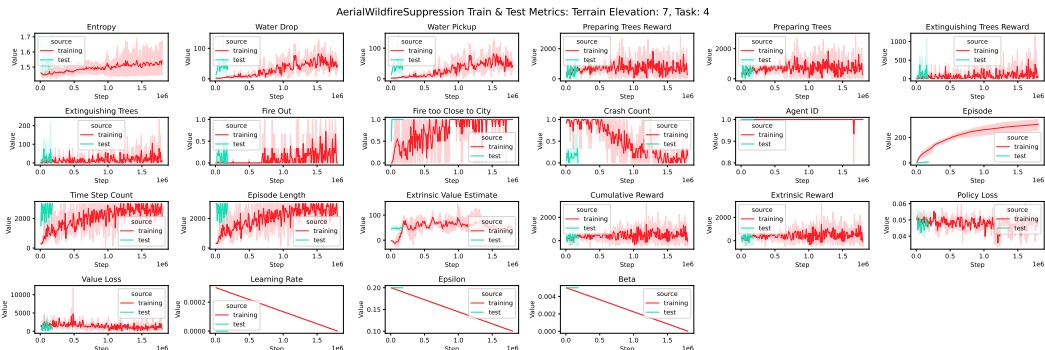

Figure 210: Aerial Wildfire Suppression: Train & Test Metrics: Terrain Elevation 7, Task 4.

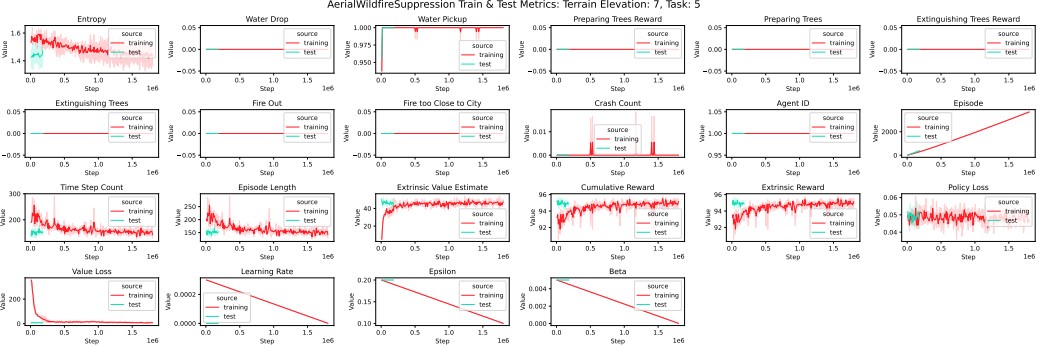

Figure 211: Aerial Wildfire Suppression: Train & Test Metrics: Terrain Elevation 7, Task 5.

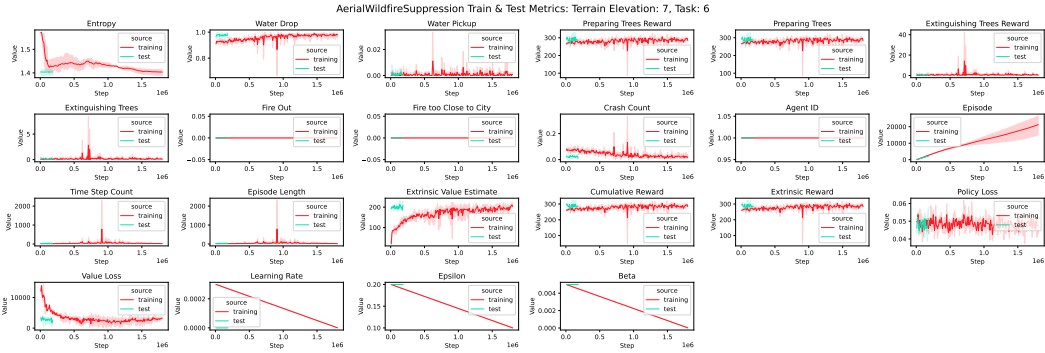

Figure 212: Aerial Wildfire Suppression: Train & Test Metrics: Terrain Elevation 7, Task 6.

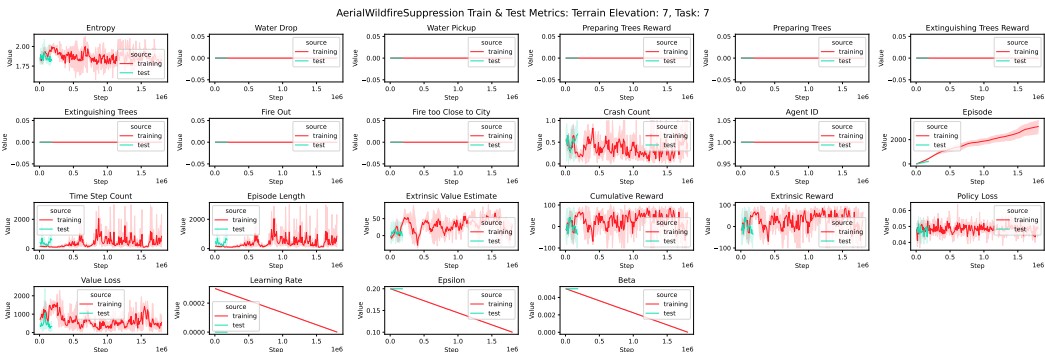

Figure 213: Aerial Wildfire Suppression: Train & Test Metrics: Terrain Elevation 7, Task 7.

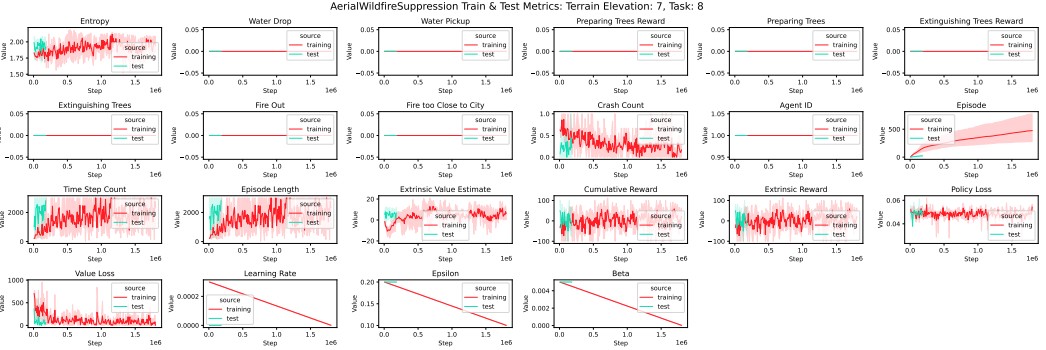

Figure 214: Aerial Wildfire Suppression: Train & Test Metrics: Terrain Elevation 7, Task 8.

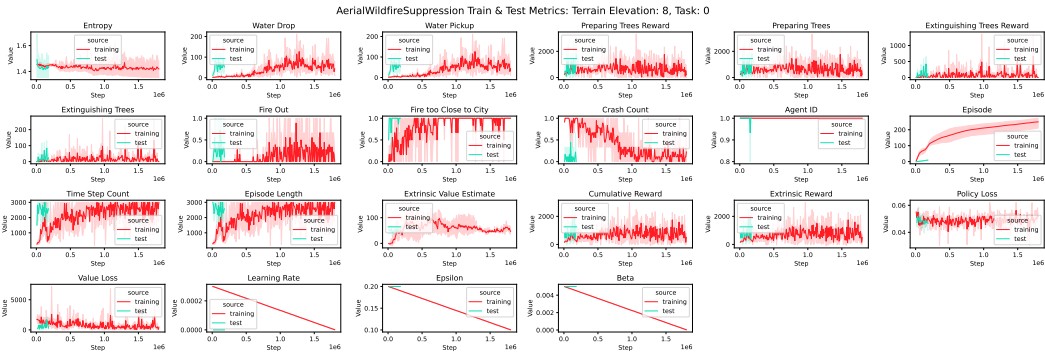

Figure 215: Aerial Wildfire Suppression: Train & Test Metrics: Terrain Elevation 8, Task 0.

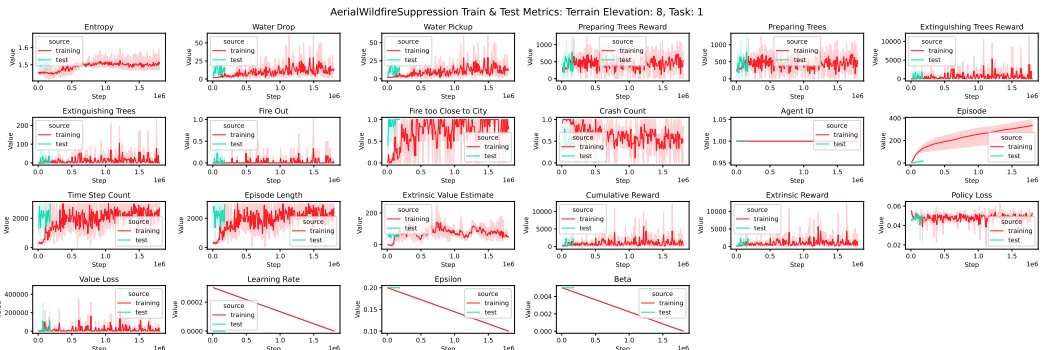

Figure 216: Aerial Wildfire Suppression: Train & Test Metrics: Terrain Elevation 8, Task 1.

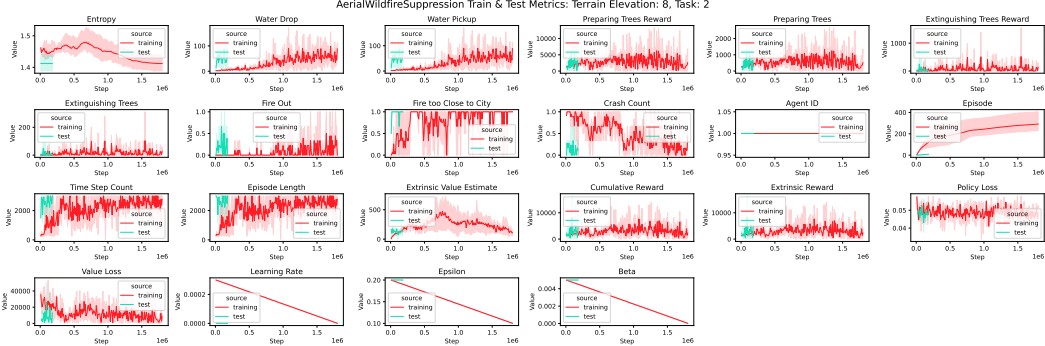

Figure 217: Aerial Wildfire Suppression: Train & Test Metrics: Terrain Elevation 8, Task 2.

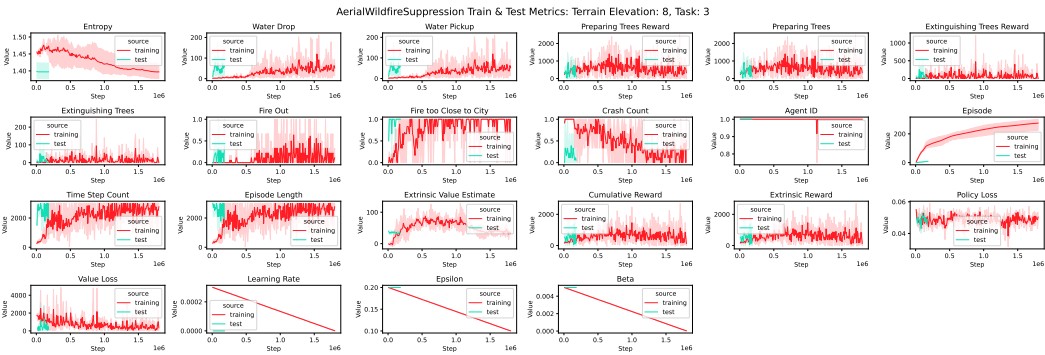

Figure 218: Aerial Wildfire Suppression: Train & Test Metrics: Terrain Elevation 8, Task 3.

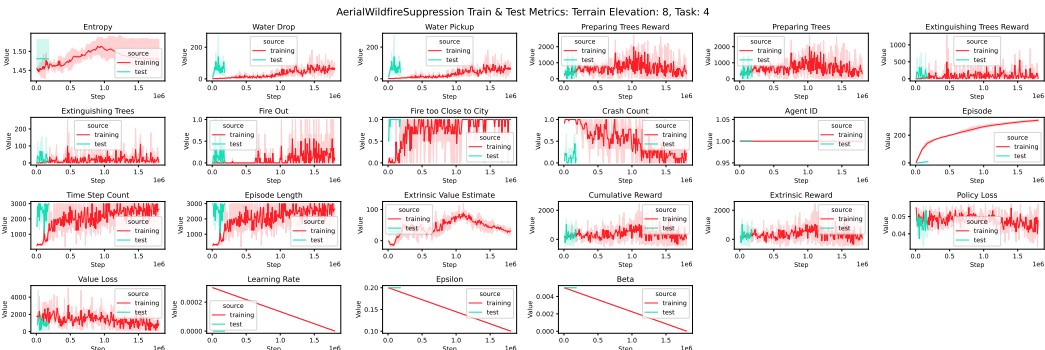

Figure 219: Aerial Wildfire Suppression: Train & Test Metrics: Terrain Elevation 8, Task 4.

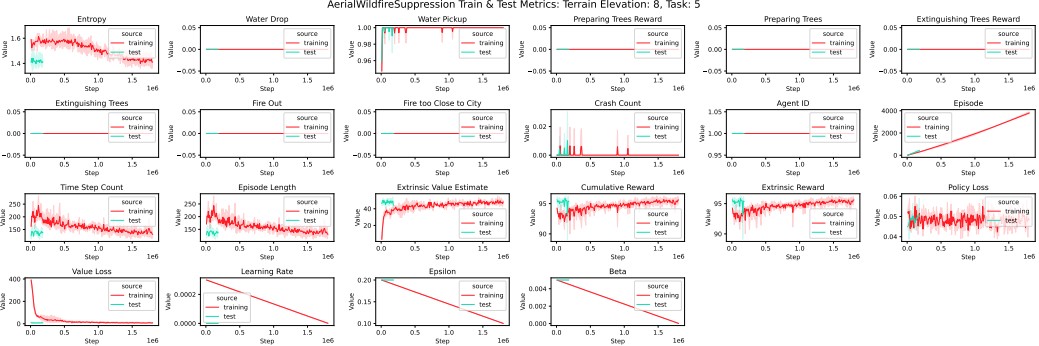

Figure 220: Aerial Wildfire Suppression: Train & Test Metrics: Terrain Elevation 8, Task 5.

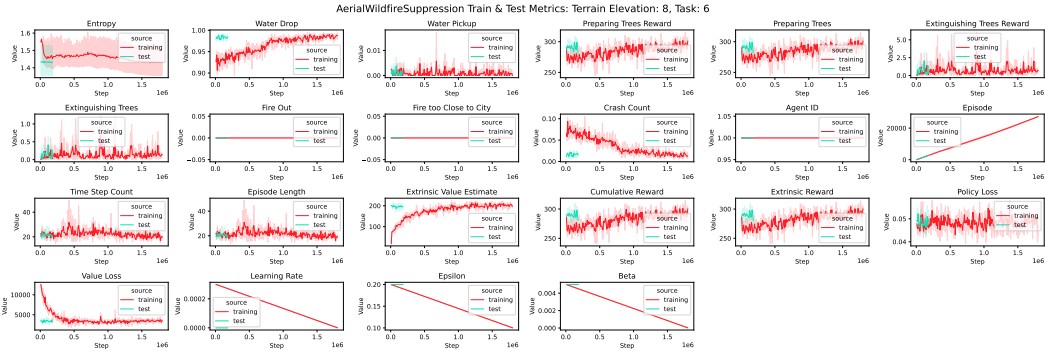

Figure 221: Aerial Wildfire Suppression: Train & Test Metrics: Terrain Elevation 8, Task 6.

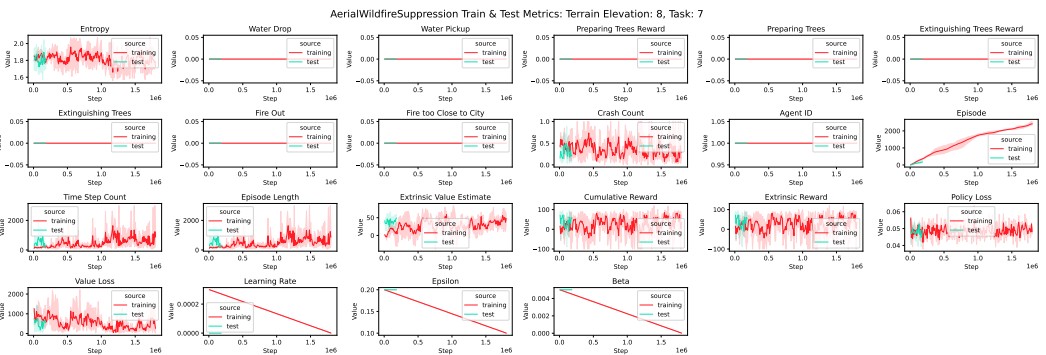

Figure 222: Aerial Wildfire Suppression: Train & Test Metrics: Terrain Elevation 8, Task 7.

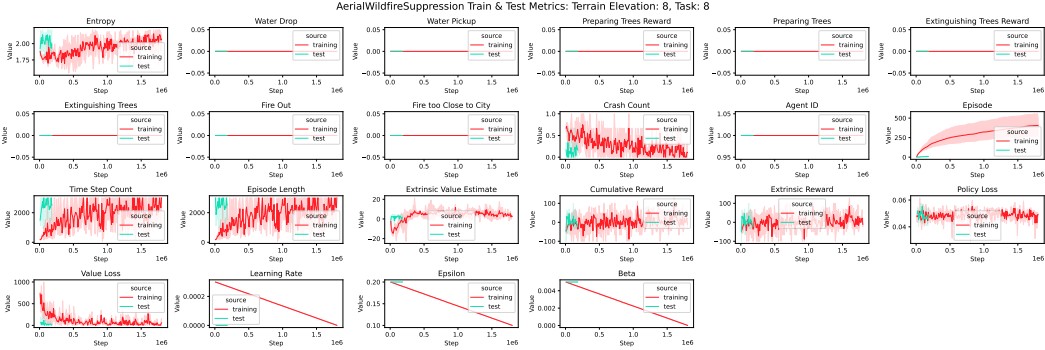

Figure 223: Aerial Wildfire Suppression: Train & Test Metrics: Terrain Elevation 8, Task 8.

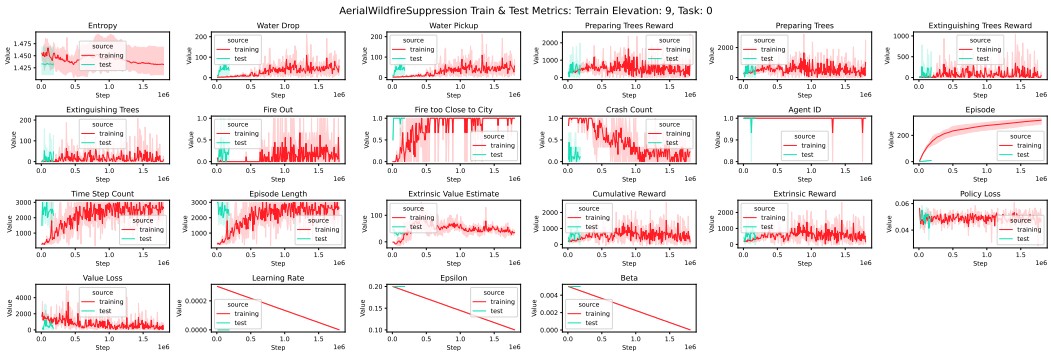

Figure 224: Aerial Wildfire Suppression: Train & Test Metrics: Terrain Elevation 9, Task 0.

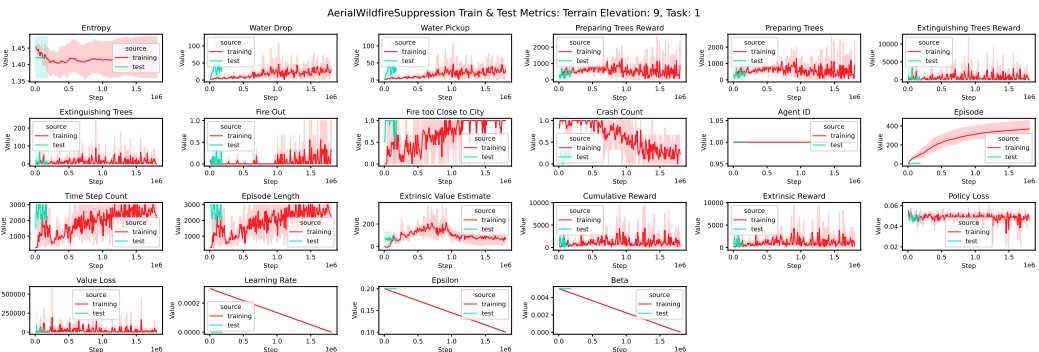

Figure 225: Aerial Wildfire Suppression: Train & Test Metrics: Terrain Elevation 9, Task 1.

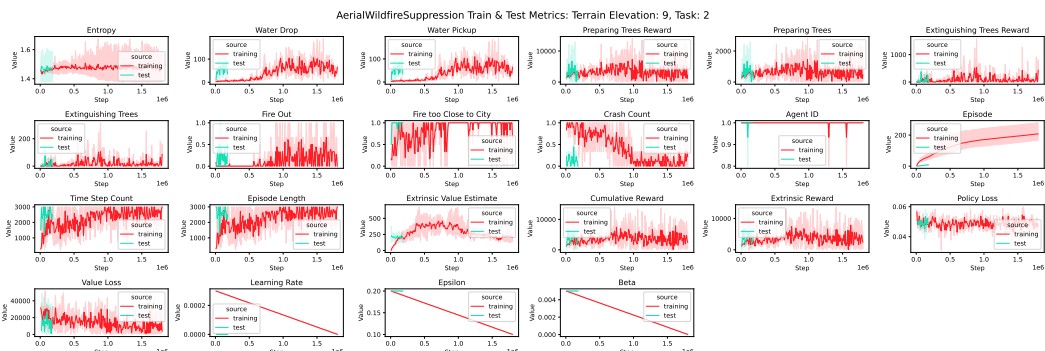

Figure 226: Aerial Wildfire Suppression: Train & Test Metrics: Terrain Elevation 9, Task 2.

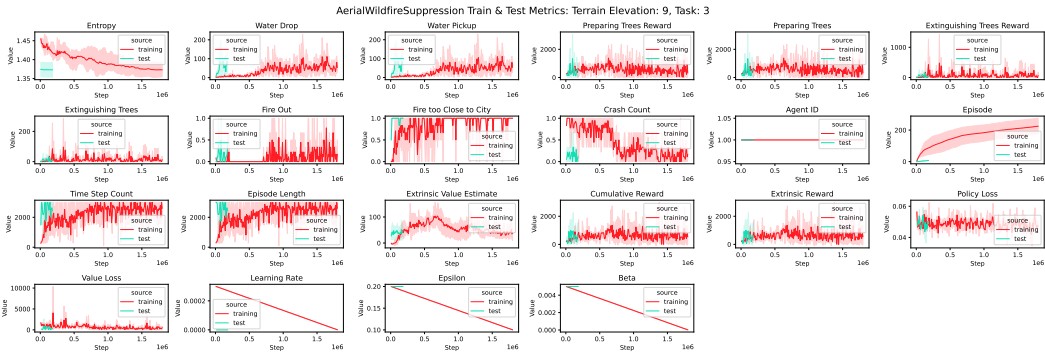

Figure 227: Aerial Wildfire Suppression: Train & Test Metrics: Terrain Elevation 9, Task 3.

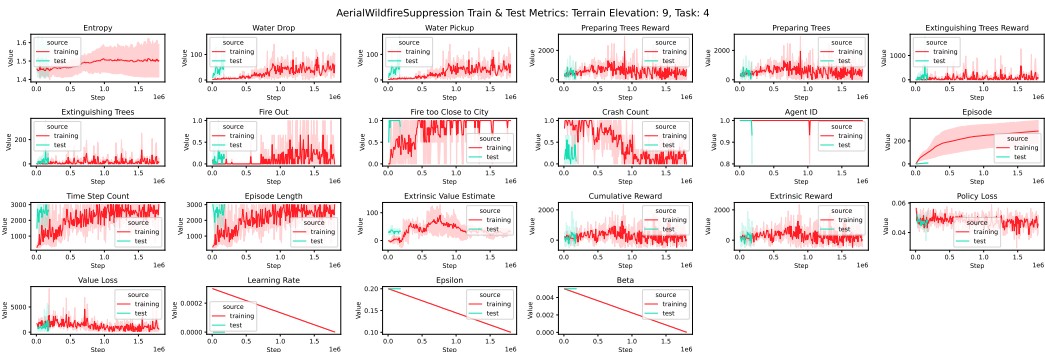

Figure 228: Aerial Wildfire Suppression: Train & Test Metrics: Terrain Elevation 9, Task 4.

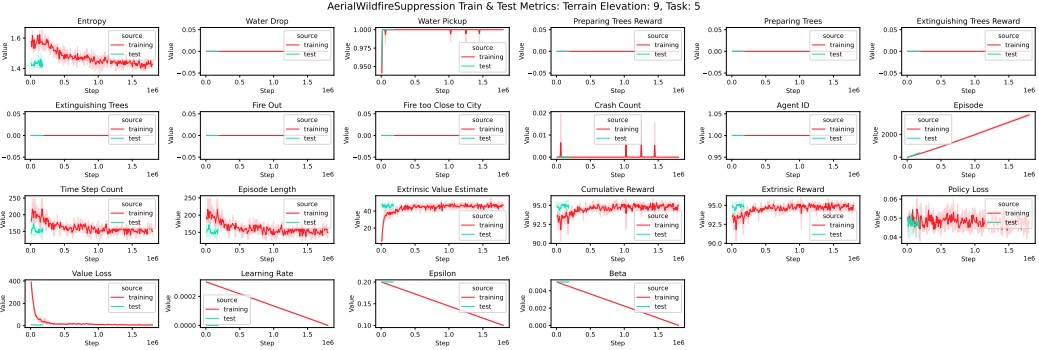

Figure 229: Aerial Wildfire Suppression: Train & Test Metrics: Terrain Elevation 9, Task 5.

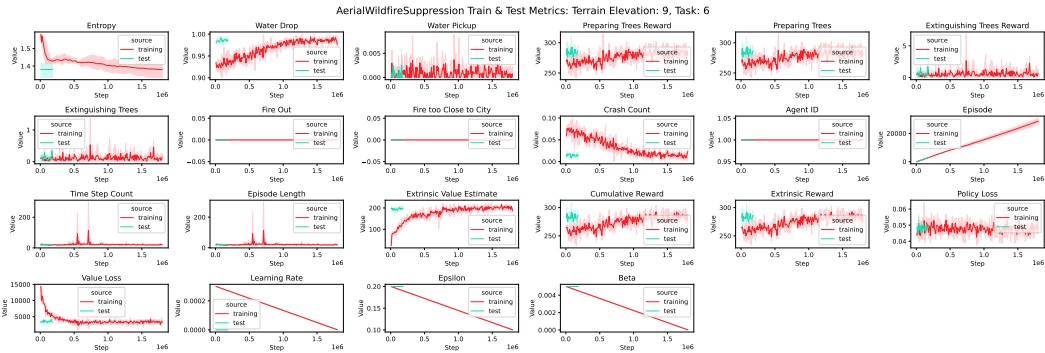

Figure 230: Aerial Wildfire Suppression: Train & Test Metrics: Terrain Elevation 9, Task 6.

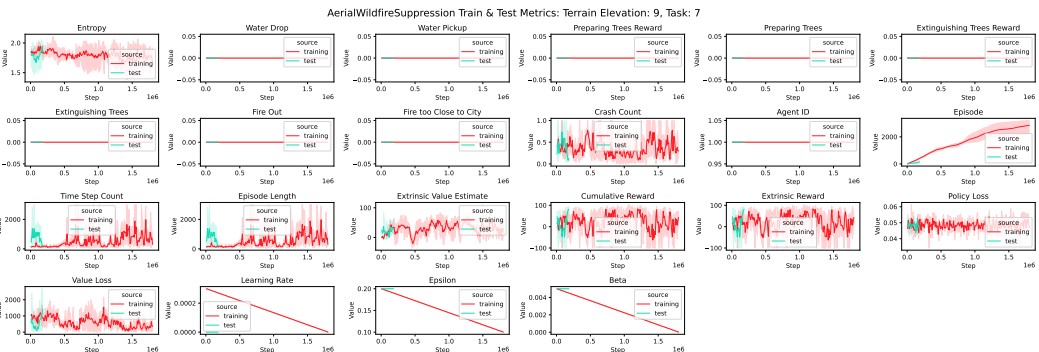

Figure 231: Aerial Wildfire Suppression: Train & Test Metrics: Terrain Elevation 9, Task 7.

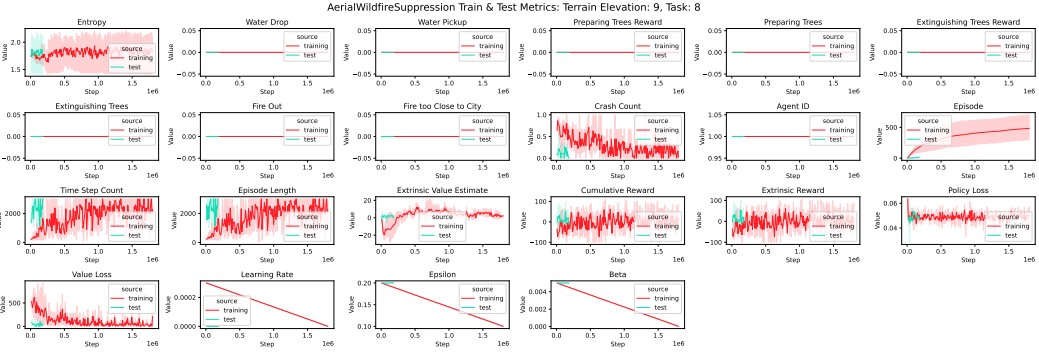

Figure 232: Aerial Wildfire Suppression: Train & Test Metrics: Terrain Elevation 9, Task 8.

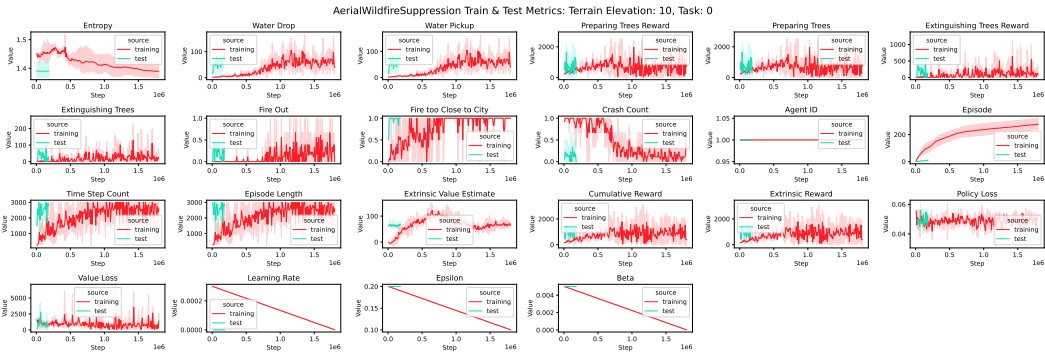

Figure 233: Aerial Wildfire Suppression: Train & Test Metrics: Terrain Elevation 10, Task 0.

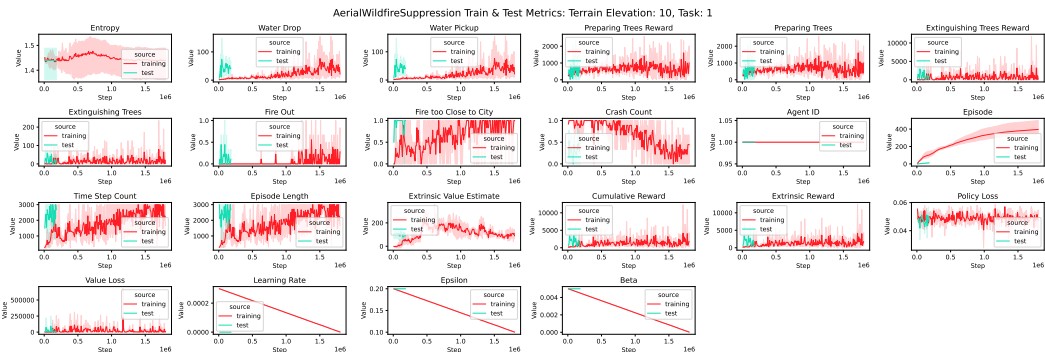

Figure 234: Aerial Wildfire Suppression: Train & Test Metrics: Terrain Elevation 10, Task 1.

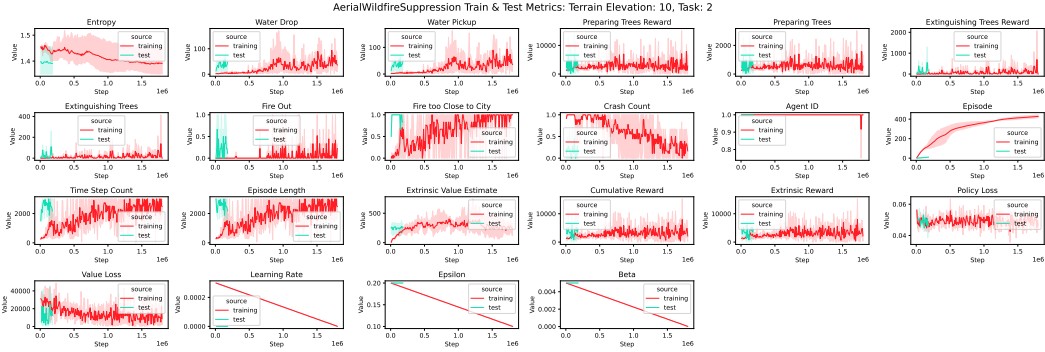

Figure 235: Aerial Wildfire Suppression: Train & Test Metrics: Terrain Elevation 10, Task 2.

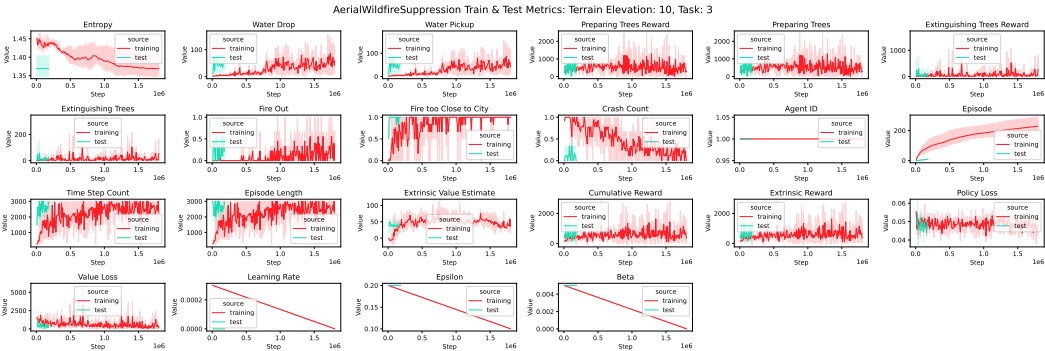

Figure 236: Aerial Wildfire Suppression: Train & Test Metrics: Terrain Elevation 10, Task 3.

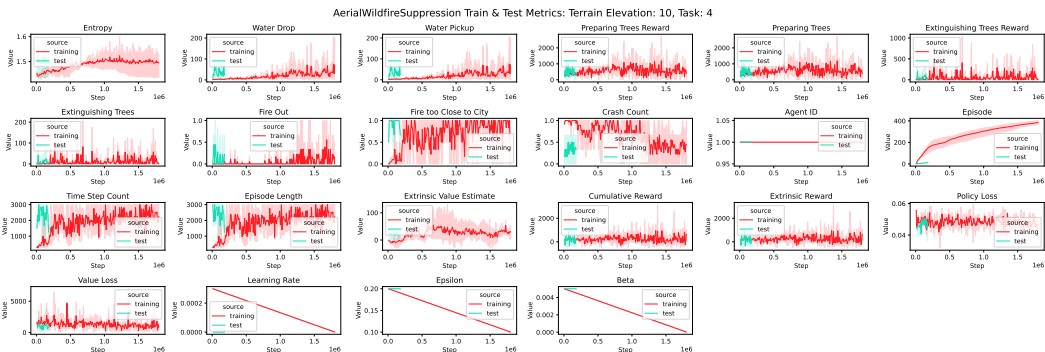

Figure 237: Aerial Wildfire Suppression: Train & Test Metrics: Terrain Elevation 10, Task 4.

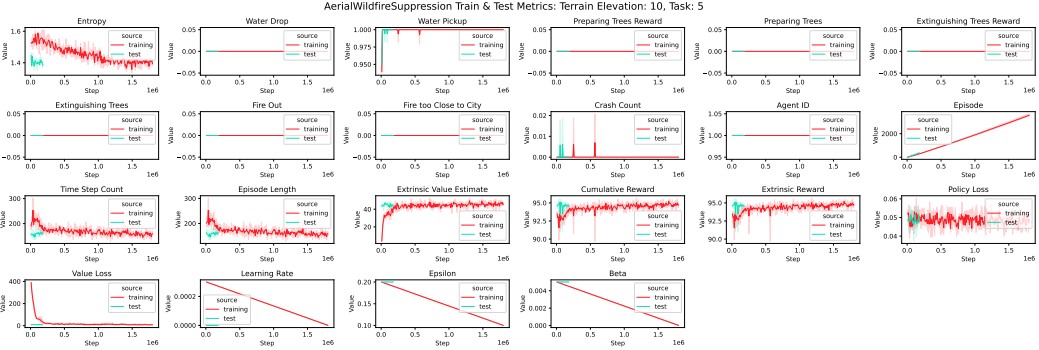

Figure 238: Aerial Wildfire Suppression: Train & Test Metrics: Terrain Elevation 10, Task 5.

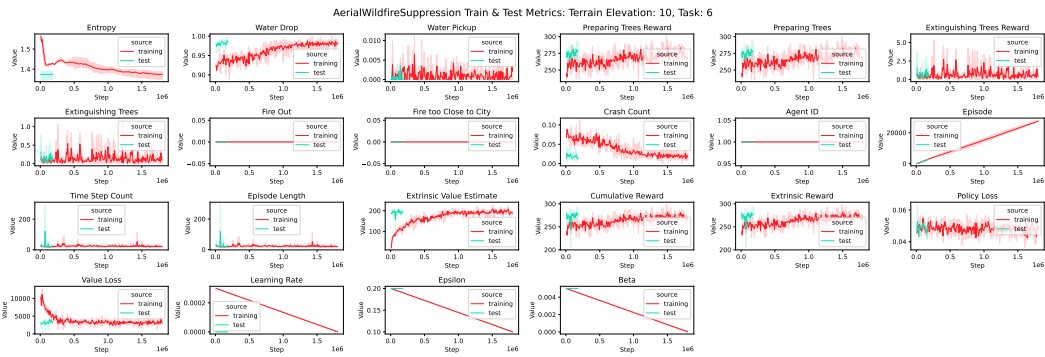

Figure 239: Aerial Wildfire Suppression: Train & Test Metrics: Terrain Elevation 10, Task 6.

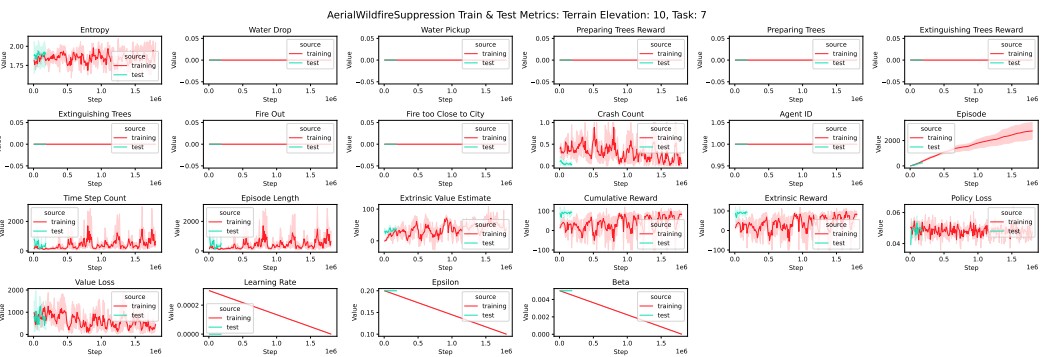

Figure 240: Aerial Wildfire Suppression: Train & Test Metrics: Terrain Elevation 10, Task 7.

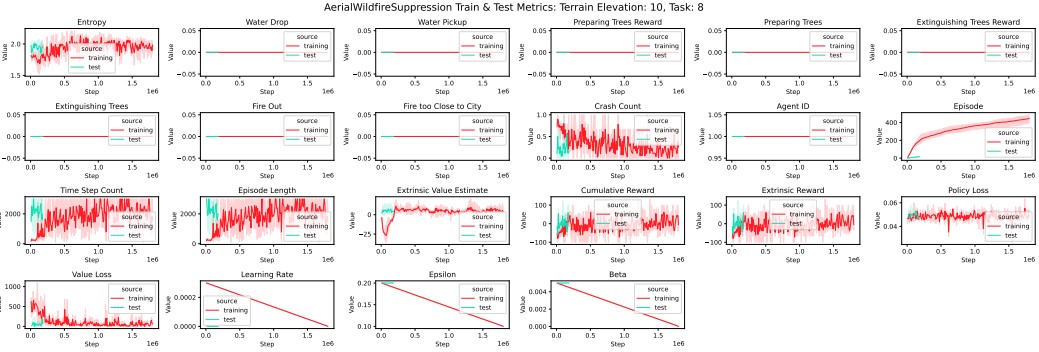

Figure 241: Aerial Wildfire Suppression: Train & Test Metrics: Terrain Elevation 10, Task 8.

## A.10.10  Aerial Wildfire Suppression: Average Test Metric - Task vs Pattern

Figure 242: Aerial Wildfire Suppression: Average Train & Test Metrics.

