# OpenReview forum: "HIVEX: A High-Impact Environment Suite for Multi-Agent Research"
_ICLR.cc/2025/Conference — Submitted to ICLR 2025_

### Official Review · Reviewer_KT7M · 2024-11-01

**Soundness:** 2
**Presentation:** 3
**Contribution:** 2
**Rating:** 3
**Confidence:** 4

**Summary:**

The paper proposes HIVEX which is an environment suite to benchmark multi-agent research focusing on ecological challenges. The environments include Wind Farm Control, Wildfire Resource Management, Drone-Based Reforestation, Ocean Plastic Collection, and Aerial Wildfire Suppression. Although Drone-Based Reforestation is an interesting cooperative MARL task, the other environments do not seem to be related to MARL.

**Strengths:**

- The paper is generally easy to follow and interesting to read.
- Prior work hasn't extensively studies multi agent research on ecological challenges.
- Drone-Based Reforestation is an interesting cooperative MARL task. The agents need to pick up seeds and recharge at the drone station, explore fertile ground near existing trees, and drop seeds while ensuring sufficient battery charge to return to the station.

**Weaknesses:**

- Wind Farm Control: In the Wind Farm Control environment, the agents' primary task is to adjust wind turbines to positions aligned against the wind direction to maximize energy generation, receiving rewards based on each turbine's performance. However, there is no mention of direct interactions or mutual influence among the multiple agents, nor is it clearly stated whether cooperation or competition exists between them.
- Wildfire Resource Management: Is "neighbouring watchtowers" referring to other agents? If so, why is there only 3 neighbors per agent? If not, then this environment is unrelated to MARL, as there is no competition or cooperation involved.
- Ocean Plastic Collection: What is the advantage of implementing this environment in Unity. Is it merely to make the demonstration of the environment look more visually impressive.

**Questions:**

- From Figure 26 onwards, the epsilon parameter was set to 0.2 in all experimental tests. However, according to the standard procedure in the PPO algorithm, epsilon should generally be set to 0 during testing, as this phase is intended solely to evaluate the performance of the trained policy without any further updates. Setting epsilon to a non-zero value may introduce unnecessary policy perturbations, potentially biasing the test results. I recommend that the authors revisit this configuration and set epsilon to 0 in testing to ensure the accuracy and consistency of the experimental results.
- In Figure 144, the caption is too close to the figure, and other figures have similar problems.
- "Crossing the environment's boundary (a 1500x1500 square surrounding a 1200x1200 island) results in a negative reward of −100." However, in Table 6, from task 1 to task 9, the reward for crossing the border is 1, which does not match the description in this paper. For example, in task 8, if the agents' goal is to find fire, then the reward 'Crossed Border' should be set to 0.

---

> ### Author Response · Authors · 2024-11-19
>
> We sincerely thank the reviewer for their encouraging feedback. We are pleased that the paper was found easy to follow and engaging to read. Your recognition of the novelty of studying multi-agent research in the context of ecological challenges is greatly appreciated. We are also glad that the Drone-Based Reforestation task stood out as an interesting cooperative MARL challenge, and that the complex agent interactions, such as seed collection, recharging, exploration, and resource management, were acknowledged. Your comments affirm the uniqueness and relevance of our work.
>
> **Weakness**
>
> "Wind Farm Control: In the Wind Farm Control environment..."
>
> Answer: You’re absolutely correct that there is no direct interaction between the agents in this environment. In this simplest example of our provided environments, the multi-agent (MA) aspect is primarily reflected in parameter sharing. However, we have developed a graph-based communication network that enables more advanced communication and collaboration capabilities. We chose to hold this for a follow-up study, as we felt incorporating it into this submission might overcomplicate the paper and risk reducing its clarity. We truly appreciate your understanding here.
>
> "Wildfire Resource Management: Is "neighbouring watchtowers"..."
>
> Answer: You are correct that "watchtowers" and "agents" are used interchangeably in this context. The number of neighboring agents is set to three, based on our reasoning that triangulation is a practical and reasonable approach for defining such neighboring structures/networks. This decision was inspired by real-world communication and reception towers, which often operate on similar principles. Thank you for pointing this out.
>
> "Ocean Plastic Collection: What is the advantage of implementing..."
>
> Answer: One of the main reasons for developing our environments in Unity is the visual richness it offers, which we believe can help spark interest from researchers in other domains. More importantly, Unity provides a highly advanced physics engine “for free” (as highlighted by DeepMind: https://deepmind.google/discover/blog/using-unity-to-help-solve-intelligence/). This enables us to simulate realistic scenarios efficiently, which we see as a significant advantage.
>
> **Questions**
>
> "From Figure 26 onwards, the epsilon parameter was set to 0.2..."
>
> Answer: Thank you for this detailed observation! To clarify, the epsilon parameter in PPO is only relevant during training, where it serves as part of the clipping mechanism to constrain policy updates. During testing, there are no gradient updates or policy optimizations; the policy acts solely based on its learned parameters. We have set the learning-rate to 0 and the learning-rate schedule to linear, which should eleminate the effect of epsilon.
> This means the epsilon parameter has no functional impact on the policy’s behavior or performance during testing. However, we see how this could cause confusion, so we will ensure this distinction is made clear in the revised manuscript. We deeply appreciate your attentiveness to this detail and your suggestion to improve clarity.
>
> "In Figure 144, the caption is too close to the figure..."
>
> Answer: Thank you for catching that! We’ve corrected the issue.
>
> "Crossing the environment's boundary..."
>
> Answer: Thank you for pointing this out—you are absolutely correct. This was an error on our part, and it has now been rectified. We greatly appreciate your attention to detail!

---

> > ### Author Response · Authors · 2024-11-24
> >
> > Dear Reviewer KT7M, we hope this message finds you well. We would like to inform you that we have updated our submission and incorporated your suggestions as much as possible within the given timeframe. Specifically, we have adjusted the distances in the figure captions within the additional results section to ensure they no longer overlap.
> >
> > Additionally, we would like to clarify the reward scale table. The reward for crossing the boundary is -100. The table in question refers to the Reward Scale, meaning the rewards described in the **The HIVEX Environment Suite** and **Reward Description and Calculation** sections are scaled according to the table, depending on the task. Please note that a single task can have multiple negative and positive rewards.
> >
> > Thank you once again for your valuable feedback. Should you have any further questions or suggestions, please do not hesitate to let us know.

---

> > > ### Author Response · Authors · 2024-11-26
> > >
> > > We would greatly appreciate your thoughts on the responses we have provided and whether there is anything further we can clarify or expand upon. Our goal is to enhance our work as much as possible by incorporating your valuable feedback and identifying actionable steps. Any additional insights you could share would mean a great deal to us. Thank you again for your time and support.

---

### Official Review · Reviewer_eEt4 · 2024-11-01

**Soundness:** 3
**Presentation:** 3
**Contribution:** 3
**Rating:** 6
**Confidence:** 3

**Summary:**

This paper introduces a novel benchmark designed for multi-agent control systems, specifically tailored to address the challenges posed by climate change. This paper opens by emphasizing the critical nature of climate change and underscores the significance of climate research, thereby establishing the relevance and importance of the proposed benchmark. It proceeds to offer succinct introductions to each environment within the benchmark, detailing their contents and associated tasks via visually engaging interfaces and foundational setup descriptions. Subsequently, the paper employs classical PPO algorithms to execute comprehensive experiments across all environments and varying levels of complexity within the benchmark, effectively illustrating its exploratory potential and validity. Furthermore, it furnishes baseline performance metrics, facilitating future research endeavors on the benchmark. Overall, this work presents an innovative multi-agent cooperation benchmark within the climate domain, addressing the deficiency of suitable experimental platforms in this area and offering robust support for subsequent investigations.

**Strengths:**

1. Each environment is accompanied by lucid diagrams, aiding the reader in comprehending the numerous influencing factors and configurations present. The visually appealing interface also intuitively conveys the rendering capabilities of the environment.
2. The RELATED WORK section is thorough and informative, contrasting a multitude of existing multi-agent benchmarks, thereby enabling readers to grasp the current landscape of benchmark development in the multi-agent field and the underlying rationale for this study.
3. The experimental section includes assessments conducted at different difficulty levels for each environment, indicating substantial untapped potential for foundational reinforcement learning algorithms. This offers researchers objective comparative data to understand the essential characteristics of each environment and its respective challenges.

**Weaknesses:**

environments or tasks? Highlighting the challenges or immediate needs for simulation environments within this domain could aid readers new to the field in appreciating the necessity of developing such a simulation environment.
2. The descriptions of the environment setups appear fragmented, detracting from the overall coherence of the manuscript. Consolidating these details into a unified format, perhaps through a tabulated summary of ENVIRONMENT SPECIFICATIONS alongside brief overviews of key points, would enhance clarity and conciseness.
3. Despite being a pioneering effort in proposing a climate-focused benchmark, the paper would benefit from a tabular comparison highlighting the attributes of the proposed benchmark relative to other multi-agent environments.

**Questions:**

1. Does the framework offer a user-friendly interface that allows developers to modify parameters such as the reward function and the number of agents?

---

> ### Author Response · Authors · 2024-11-19
>
> We sincerely thank the reviewer for their thoughtful feedback. We are glad that the lucid diagrams and intuitive interface effectively conveyed the environment's complexity and rendering capabilities. Your recognition of the thorough and informative RELATED WORK section, which situates our study within the broader multi-agent benchmark landscape, is greatly appreciated. Lastly, we are pleased that the experimental assessments across varying difficulty levels were found valuable for highlighting the potential of reinforcement learning algorithms and providing comparative data for future research.
>
> **Weakness**
>
> "environments or tasks?"
>
> Answer: Environments > Tasks > [Difficulty, Layout, None]
>
> "Highlighting the challenges or immediate needs for simulation environments..."
>
> Answer: Incorporating this, thank you.
>
> **Questions**
>
> 1. Answer: We currently do not provide a dedicated interface for customizing the reward function. However, users have the flexibility to shape rewards as part of their own learning frameworks. On the other hand, we have made all environment-specific parameters—such as the number of agents, task configurations, layout, and difficulty—easily accessible. These can be modified effortlessly, as demonstrated in the training and testing configurations provided in the appendix. We hope this flexibility supports users in tailoring the environments to their specific needs.

---

> > ### Author Response · Authors · 2024-11-24
> >
> > Dear Reviewer eEt4, we would like to kindly inform you that we have updated our submission. Specifically, we have addressed the **weaknesses** you mentioned and ensured consistent tables for the environment specifications across all environments, as per your suggestion. Thank you for bringing this to our attention.
> >
> > Please do not hesitate to reach out if you have any further questions or additional suggestions for us to consider. We greatly appreciate your time and valuable feedback.

---

### Official Review · Reviewer_BSb6 · 2024-11-04

**Soundness:** 3
**Presentation:** 3
**Contribution:** 2
**Rating:** 5
**Confidence:** 4

**Summary:**

This paper proposes a benchmark for multiagent research. The benchmark provides environments about ecological challenges including wind farm control, wildfire resource management, drone-based reforestation, ocean plastic collection, and aerial wildfire suppression. This paper claims that the proposed benchmark presents more realistic environments than existing benchmarks, so the findings in the proposed benchmark have more potential to be equipped for real-world problems.

**Strengths:**

1. This work uses the Unity engine to create multiple environments representing ecological challenges and allows each environment to have different difficulties. Moreover, each environment provides both vector observations and visual observations, which is more realistic than many existing environments

2. Some environments allow their features to be procedurally generated and the test-time evaluation can use environments that have never been seen.

3. This work provides evaluation results for each environment which can serve as the baselines for future works.

**Weaknesses:**

1. This paper claims that the proposed environments are more realistic than existing environments. However, according to the description of the paper, this is done only by adding visual representations and low-level action/state space. It is unclear how accurately the environments can simulate the real-world environment, for example, the uncertainty of state transition. Therefore, the sim-to-real gap can still be considerably large. I know that the authors have discussed this in the limitation section, but it is indeed a problem that weakens the contribution of this work.

2. This paper proposes environments for multiagent research. However, when generating baselines for evaluation, this work only tests the performance of the PPO algorithm. Why not try some multiagent RL algorithms as the environments are for multiagent research?

3. The authors claim that Wildfire Resource Management and Ocean Plastic Collection are excluded from scalability tests because of fixed layout, agent count, and amount of plastic. Why these are fixed? The environments should allow for configuration like other environments.

**Questions:**

Please see the Weaknesses above.

---

> ### Author Response · Authors · 2024-11-19
>
> We sincerely thank the reviewer for highlighting the strengths of our work. We are delighted that you found value in our use of the Unity engine to create diverse environments with varying levels of difficulty and realistic observation types. Your recognition of the procedurally generated features and the ability to evaluate performance in unseen environments is especially great. Additionally, we appreciate your acknowledgment of the evaluation results as useful baselines for future research. Your encouraging comments reinforce our contributions and motivate us to continue improving this work - thank you!
>
> **Weaknesses**
>
> 1. Lack of Comparable Open-Source MAS Environments: Thank you for raising this point. To the best of our knowledge, there are currently no open-source MAS environments focused specifically on real-world applications. Most existing open-source MAS environments appear to fall under toy or game scenarios. However, if you are aware of relevant references or examples that we may have missed, we would greatly appreciate it if you could share them. Including such comparisons would strengthen our paper and allow us to better contrast our contributions.
>
> 2. Testing Additional Methods: We truly appreciate your suggestion to test additional methods. As much as we would like to, running full training and testing across all environments, tasks, and difficulty levels is quite resource-intensive, taking approximately three months to complete on our current setup (a single 3090 GPU). Despite these constraints, we are optimistic that the community interested in using our MAS evaluation suite will contribute additional results over time.
>
> 3. For the “Wildfire Resource Management” environment, you are absolutely correct that fixed locations for the watchtowers were designed to ensure complete coverage of the island. However, as you pointed out, introducing multiple layout configurations (similar to the approach used in Wind Farm Control) would add further diversity. We will prioritize this enhancement in future iterations.
> For the “Ocean Plastic Collection” environment, increasing the agent count does present challenges, as it would reduce the number of plastic pebbles available per agent, thereby impacting individual agent performance. This tradeoff is an important consideration, and we thank you for highlighting it.

---

> > ### Author Response · Authors · 2024-11-24
> >
> > Dear Reviewer BSb6, we hope this message finds you well. We would like to kindly inform you that we have updated our submission. If you have any follow-up questions or require further information, please do not hesitate to let us know.
> >
> > Thank you once again for your time and effort in reviewing our work. We greatly appreciate your thoughtful input.

---

> > > ### Comment · Reviewer_BSb6 · 2024-11-25
> > >
> > > I thank the authors for the rebuttal and revision. As the weaknesses 2 and 3 are still unresolved, I choose to maintain my current score.

---

> > > > ### Author Response · Authors · 2024-11-26
> > > >
> > > > Thank you for your thoughtful review and the time you've dedicated to improving our work. We sincerely appreciate your feedback. Could you kindly help us better understand your concerns regarding **Weakness 3** and clarify how our previous response fell short of addressing them to your satisfaction?
> > > >
> > > > Thank you very much

---

> > > > > ### Comment · Reviewer_BSb6 · 2024-11-30
> > > > >
> > > > > Regarding weakness 3, what I expect is that the Wildfire Resource Management and Ocean Plastic Collection should also be included in the scalability tests by using dynamic layouts, agent counts, etc.

---

> > > > > > ### Author Response · Authors · 2024-11-30
> > > > > >
> > > > > > That is a fantastic idea; thank you very much. We will include this in our manuscript. Thank you for bringing this up and helping us improve our work.

---

### Official Review · Reviewer_Ujp7 · 2024-11-05

**Soundness:** 1
**Presentation:** 2
**Contribution:** 1
**Rating:** 3
**Confidence:** 4

**Summary:**

This paper introduces HIVEX, a benchmark for multi-agent reinforcement learning that addresses critical ecological challenges. It includes five environments with varying levels of difficulty and supports over ten agents, accepting both vector and image inputs. Episodes can last up to 5000 steps. The benchmark provides PPO as a baseline, running through different environments and presenting the results.

**Strengths:**

1. It provides comprehensive code for both evaluation and training.
2. The task design is well-structured, and the benchmark’s support for image input is both rare and crucial for advancing multi-agent reinforcement learning (MARL) development.

**Weaknesses:**

1. The methods tested are quite limited, with only PPO evaluated. Have you considered testing additional methods like MAPPO or MAA2C?
2. The reward structure doesn’t clearly reflect the difficulty levels of the tasks. What is the best performance achievable with an optimal policy?
3. The role of image input is unclear. How does your method utilize visual inputs? If tasks are still achievable without image input, what is the intended value of including them?
4. The advantages of HIVEX compared to benchmarks like Melting Pot and Neural MMO remain unclear.
5. How can this benchmark be expanded? Is it easy for users to create custom tasks or modify existing ones within your benchmark?
6. Lack of innovation in new methods. How could the training pipeline or algorithm be modified to achieve better results?

**Questions:**

Please see the Weakness.

---

> ### Author Response · Authors · 2024-11-19
>
> Thank you for taking the time to thoroughly review our paper. We truly appreciate your insightful feedback, as well as your positive comments on the provided code, task design structure, and the diversity of observation spaces we included, spanning vector, visual, continuous, and discrete formats.
>
> **Weaknesses**
>
> 1. Answer: Testing Additional Methods: We greatly appreciate your suggestion to include results for additional methods. However, performing full training and testing across all environments, tasks, and difficulty levels is quite resource-intensive, taking approximately three months to complete with our current setup (a single 3090 GPU). While our compute resources are limited, we are optimistic that the community interested in benchmarking with our MAS evaluation suite will contribute additional results over time. Your comment highlights an important area, and we hope this will foster collaborative engagement.
>
> 2. Answer: Clarifying the "Difficulty" Term: Thank you for pointing this out. We fully agree that the term "difficulty" requires refinement. As you noted, it is currently tied to the maximum height of the terrain amplitude, but this doesn’t directly correlate with performance outcomes. We will revise this terminology in the paper to ensure it better reflects the intended meaning and avoids ambiguity.
>
> 3. Answer: Visual Input in Environments: You raised a valid question regarding the role of visual inputs. To clarify, the Drone-Based Reforestation and Aerial Wildfire Suppression environments utilize visual observations, and the Ocean Plastic Collection environment includes grid-based observations, which can also be treated as images depending on the user’s preference. A core goal of our design is to offer environments with combinations of observation types (e.g., vector and visual) and formats (e.g., continuous and discrete), allowing for diverse testing scenarios.
>
> 4. Answer: Comparison to Other Frameworks: Thank you for bringing up comparisons to Melting Pot and Neural MMO. Our suite emphasizes diverse and larger observation spaces, alongside varied action spaces, to simulate scenarios closer to real-world complexity rather than grid-worlds focusing primarily on social behavior (as in Melting Pot). While Neural MMO focuses on scalability and performance with many agents, our focus is on diverse, realistic environments that challenge algorithms in ways relevant to real-world applications.
>
> 5. Answer: Environment Modifiability: We appreciate your feedback on making the benchmarking suite more customizable. While our primary goal for this submission was to provide a variety of fixed environments for standardized testing and comparability, we absolutely agree with the value of open-sourcing. We are committed to releasing the full environment source code so that others can modify and expand upon it in the future. Thank you for this thoughtful suggestion!
>
> 6. Answer: Focus of the Submission: The primary aim of this work is to introduce a new benchmarking and evaluation suite for MAS research rather than to propose new algorithms or methods. We hope this suite will provide a valuable platform for evaluating and comparing existing and future approaches in the field.

---

> > ### Author Response · Authors · 2024-11-24
> >
> > Dear Reviewer Ujp7, we sincerely thank you for your insightful feedback. In response to your comments, we have revised our submission and replaced the term "difficulty" with "terrain elevation" throughout the figures and text, as this term better reflects the intended meaning.
> > We deeply appreciate your time and effort in evaluating our work and hope the revisions align with your suggestions. Thank you once again for your thoughtful review.

---

> > > ### Comment · Reviewer_Ujp7 · 2024-11-25
> > >
> > > I thank the authors for the rebuttal and revision. As the weaknesses 1,2,3,5,6 are still unresolved, I choose to maintain my current score.

---

> ### Author Response · Authors · 2024-11-26
>
> Thank you for considering our additional changes, please allow us to iterate over the weaknesses that you have mentioned:
>
> **Weakness 2.** The reward structure doesn’t clearly reflect the difficulty levels of the tasks. What is the best performance achievable with an optimal policy?
>
> > Answer: We have eliminated the difficulty term and changed this to what it exactly is: Terrain Elevation, so we were hoping this weakness was solved.
>
> **Weakness 3.** The role of image input is unclear. How does your method utilize visual inputs? If tasks are still achievable without image input, what is the intended value of including them?
>
> > Answer: The visual input is relevant for the following environments and is part of the agents individual observation space:
> > - Ocean Plastic Collection: capturing the surrounding area of the agent-controlled vessel
> > - Drone-Based Reforestation: there is a virtual camera attached to each drone pointing downward
> > - Aerial Wildfire Suppression: also here, there is a virtual camera attached to each aeroplane
>
> > Details for the visual observation in the respective environments can be found in the Appendix A.6
>
> **Weakness 5.** How can this benchmark be expanded? Is it easy for users to create custom tasks or modify existing ones within your benchmark?
> > Answer: Currently, we are not supporting modifications to the tasks other than reward-shaping.
>
> **Weakness 6.** Lack of innovation in new methods. How could the training pipeline or algorithm be modified to achieve better results?
> > Answer: We hope the community will develop and test new methods and algorithms on our environment suite.
>
> ---
>
> We dearly hope this answers your questions and weaknesses. Please let us know if you think otherwise and why. This would help us tremendously to identify **action-items** to improve and incorporate your valuable feedback.
>
> Thank you very much for your time

---

### Meta-Review · Area_Chair_t6XQ · 2024-12-29

**Metareview:**

The paper proposes HIVEX, multi-agent benchmark focused on ecological challenges aiming to provide environments that mimic real-world problems and increase the complexity of environments to advance the state of multi-agent research.

The reviews of the HIVEX paper are mixed, with reviewers identifying both strengths and weaknesses. The reviewers acknowledge that the suite addresses critical ecological challenges and provides a comprehensive code for evaluation and training. The use of the Unity engine and the inclusion of both vector and visual observations are also considered strengths. However, reviewers raise concerns about the limited methods tested, with only PPO being evaluated. Some reviewers also point out that the reward structure does not clearly reflect the difficulty levels of the tasks and ask for clarification on the role of image inputs. Some reviewers also note that the sim-to-real gap may be large and that it is unclear how the environments simulate real-world uncertainties.

**Additional Comments On Reviewer Discussion:**

While the reviewers and myself appreciate the author's responses and the updates to the paper, the changes made were insufficient to tip the decisions. Most notably, the reviewers maintain, and I concur, that the paper needs more comprehensive baselines and more convincing realism argument, and Wildfire Resource Management and Ocean Plastic Collection environments to be included in scalability tests.

I also appreciate the authors' note, and have personally double checked the paper, reviewers, and made the decision on the paper, taking my option of the current state of the paper.

---

### Decision · Program_Chairs · 2025-01-22

Reject